# PINFDiT: Energy-Based Physics-Informed Diffusion Transformers for General-purpose Time Series Tasks

**Defu Cao**,* **Wen Ye**,* **Yizhou Zhang, Sam Griesemer, Yan Liu**
University of Southern California
`{defucao, yewen, zhangyiz, samgriesemer, yanliu.cs}@usc.edu`

## Abstract

Time series analysis underpins scientific advances. While specialized models have advanced various time series tasks, scientific domains face unique challenges: limited samples with complex physical dynamics, missing observations, multi-resolution sampling, and requirements for physical consistency. With the increasing demands on generative modeling capabilities, we introduce PINFDiT, a diffusion transformer-based model with physics injection during inference. Our approach combines a transformer backbone for capturing temporal dependencies with a comprehensive masking strategy that addresses imperfect data. The diffusion framework enables high-quality sample generation with inherent generative capability. In addition, our model-free physics-guided correction steers generated samples toward physically consistent solutions using calibrated Langevin dynamics, which balances distribution fidelity and physical law adherence without architectural modifications or retraining. Our evaluation demonstrates PINFDiT's effectiveness across multivariate forecasting with imperfect data, physics knowledge incorporation in data-limited scenarios, zero-shot and fine-tuning performance across diverse domains, establishing that it bridges the gap between general-purpose and domain-specific models.

## 1 Introduction

Time series analysis serves as a cornerstone methodology underpinning scientific advances across natural science, sustainability research, and healthcare innovation(Ye et al., 2024; Cuomo et al., 2022; Burger et al., 2024). While specialized models like TCNs (Franceschi et al., 2019), LSTMs (Siami-Namini et al., 2019), GNNs (Wu et al., 2020), and Transformers (Woo et al., 2024a) have advanced in time series tasks including forecasting, imputation, anomaly detection, and data generation, each task presents unique requirements that are difficult to generalize across different applications. Scientific domains introduce additional layers of complexity beyond these general challenges. Experimental data often suffers from limited sample sizes due to the prohibitive costs of data collection in fields like high-energy physics, climate science, and biomedical research. Scientific time series frequently exhibit complex, non-stationary dynamics governed by underlying physical laws that simple statistical approaches fail to capture. Beyond that, generative modeling capabilities prove essential for scientific time series analysis, offering researchers powerful tools to simulate counterfactual scenarios, supplement sparse datasets, conduct hypothesis testing, and produce physically consistent synthetic data that accurately reflects the complex dynamics underlying natural phenomena.

These challenges are exacerbated by missing observations during critical phenomena, multi-resolution sampling intervals, and the need to incorporate multivariate data streams capturing different aspects of the same underlying process. Specifically, *missing values* (Kollovieh et al., 2023) fragment temporal continuity and create representational gaps that models must learn to bridge or impute, requiring sophisticated handling to preserve semantic meaning without introducing artifacts. In addition, *multi-resolution sampling* (Niu et al., 2023) introduces inconsistent information density across different variables and time periods, distorting cross-variable relationships that might otherwise

---

*Equal contribution alphabetical order.

reveal important dependencies and forcing models to reconcile information at different granularities. Similarly, *irregular temporal intervals* (Cao et al., 2023a) undermine the equidistant assumptions underlying many architectures, compromising temporal dependency learning and requiring models to adapt to varying time scales simultaneously while maintaining causal consistency. Furthermore, while channel-independent solutions (Nie et al., 2023; Ansari et al., 2024) benefit from temporal relationships, they overlook the correlations between *multivariate* inputs, resulting in impractical outcomes. Despite these prevalent challenges in scientific applications, current benchmark datasets (Li et al., 2018; Zhou et al., 2021; Alexandrov et al., 2020) often present idealized scenarios that inadequately capture the inherent complexities, creating a critical gap between controlled evaluation environments and scientific deployment contexts.

Furthermore, scientific applications demand *physical consistency* and interpretability beyond what black-box models can deliver, as predictions that faithfully adhere to established scientific principles are required for validation and acceptance (Meng et al., 2022b). For example, in climate science, conservation laws must be preserved despite sparse observations, while in healthcare, physiological constraints like heart rate bounds and circadian rhythms prevent biologically implausible predictions. Incorporating such physics knowledge significantly enhances model performance while reducing computational overhead, especially in data-scarce domains where physical laws provide crucial prior knowledge. Properly modeling such systems requires likelihood-based approaches that quantify uncertainty and provide theoretical guarantees (Simons et al., 2023). However, computing exact likelihoods for complex time series distributions becomes intractable Beal (2003). Traditional simulation-based inference methods bypass the need for tractable likelihoods but require prohibitively large numbers of simulations and struggle to scale efficiently for time series data, particularly when physical constraints must be maintained across long-term time sequences (Gloeckler et al., 2024).

In this paper, we introduce PINFDiT, a diffusion transformer-based generative model with physics injection during the inference stage to address the aforementioned challenges. The transformer backbone provides natural advantages in capturing complex temporal dependencies across varied time scales, while the diffusion framework enables high-quality sample generation with inherent probabilistic results. To achieve uniform input, we design a comprehensive masking strategy including position, stride, and block masks, which addresses the practical realities of imperfect time series data encountered across domains. Furthermore, we incorporate a model-free physics-guided correction that steers the generated samples toward physically consistent solutions during the sampling stage. Different from simulation-based inference (Geffner et al., 2023; Xiong et al., 2025), this correction directly addresses likelihood intractability through an ELBO approximation and leverages our key theoretical insight that the optimal physics-informed distribution takes a closed-form Boltzmann form. Therefore, we employ calibrated Langevin dynamics with guaranteed convergence that balances distribution fidelity and physical law adherence without requiring architectural modifications or retraining. Our experimental framework features a comprehensive four-part evaluation strategy: First, we evaluate the effectiveness of physics knowledge incorporation within our framework, where the physics knowledge is known but the data is limited. Second, we assess multivariate time series forecasting capabilities on real-world applications featuring missing values and multi-resolution data. Later, we demonstrate PINFDiT's effectiveness as a generative model through comprehensive tasking, including synthetic generation, imputation, and anomaly detection. Fourth, we showcase how our PINFDiT serves as a proto-foundation model with the zero-shot setting.

The main contribution of our work can be summarized as the following threefold:

- A Unified Framework for Imperfect Real-World Data: We propose PINFDiT to address the real-world scientific challenges of missing values, multi-resolution and multivariate data. With a systematically designed masking mechanism, PINFDiT jointly leverages the transformer's temporal modeling and diffusion's generative sampling capabilities in an end-to-end training pipeline, creating a robust framework specifically tailored for complex scientific time series analysis.

- A Generalist-to-Specialist Inference Paradigm: We build a model-editing-free physics knowledge injection framework under the lightweight Langevin-correction mechanism and the large-scale pre-trained model. This approach treats the model as a statistical *generalist* and applies a lightweight Langevin correction step to act as a domain *specialist* during inference. The implementation maintains theoretical convergence guarantees while ensuring generated time series adhere to fundamental physical laws without retraining the model.

- We validate PINFDiT through comprehensive experiments aligned with our foundational principles, evaluating its performance across physics-guided forecasting, practical forecasting with miss value and multi-resolution sampling, synthetic generation, anomaly detection and imputation task. Moreover, we evaluate PINFDiT's ability in zero-shot settings. The results demonstrate state-of-the-art performance, exceptional cross-task adaptability, and seamless integration of scientific domain knowledge, confirming the model's effectiveness for addressing complex temporal challenges in scientific applications.

## 2 RELATED WORK

**Physics-Informed Machine Learning.** Physics has traditionally been injected into machine learning systems *during training* by augmenting the loss with PDE residuals, as in physics-informed neural networks (PINNs) (Raissi et al., 2019), operator learners such as DeepONet (Lu et al., 2021) and FNO (Li et al., 2021), or more recent physics-aware transformers (Zhao et al., 2024). NeuralODEs (Chen et al., 2018; Beck et al., 2024) and their extension, NeuralCDEs Kidger et al. (2020b), explicitly model latent dynamics as ordinary or controlled differential equations, thereby embedding continuous-time evolution into the network design. When the vector field is further regularised to respect conservation laws or Hamiltonian structure (Greydanus et al., 2019; Cranmer et al., 2020), the resulting trajectories satisfy first-principle constraints throughout training, akin to PINNs and operator learners. In contrast, we follow the emerging line of *inference-time debiasing* that corrects a frozen surrogate without retraining; Simulation-Calibrated SML (Fan et al., 2025) and Monte-Carlo importance pinning (Lin et al., 2024) exemplify this idea but have so far been limited to low-dimensional surrogates. Diffusion-based generators have also been coupled to physical laws, e.g. SDEdit-PDE (Meng et al., 2022a) and FluidDiffusion (Qiu et al., 2024), yet these works focus on single snapshots or images rather than long, irregular time series. Orthogonally, simulation-based inference (SBI) methods such as SNPE (Lueckmann et al., 2017), CSBI (Gloeckler et al., 2024) and LFBC (Xiong et al., 2025) exploit mechanistic simulators to learn parameter posteriors. Finally, hybrid forecasters that blend neural predictors with numerical weather models or other filters (Das et al., 2024; Shi et al., 2023) illustrate the benefits of combining data-driven and physics-based cues. Distinct from all prior art, PINFDiT marries a large-context transformer with a diffusion prior and a *plug-and-play Langevin corrector*: physics enters only through an energy term, giving a model-agnostic, zero-retraining route to enforce conservation laws while preserving the generative flexibility of foundation models.

**Diffusion models for Time Series.** Despite growing interest in diffusion models across various scenarios (Li et al., 2022a; Lu et al., 2024; Sui et al., 2024a;b), their application in time series analysis remains less explored compared to pre-trained language models (Zhang et al., 2024; Gruver et al., 2024; Jin et al., 2023; Pan et al., 2024; Ekambaram et al., 2024). Most existing studies also focus solely on forecasting and the choice of backbone model also varies among VAE(Li et al., 2022b), RNN(Rasul et al., 2021), and transformers. Recently, CSDI (Tashiro et al., 2021) first utilizes a diffusion model for time series imputation with a self-supervised approach. SSSD (Alcaraz & Strodthoff, 2023) combines the structured state space model with the diffusion model for imputation. ImDiffusion (Chen et al., 2023) leverages diffusion models as time series imputers to achieve accurate anomaly detection. $D^3VAE$ (Li et al., 2022b) proposes a generative time series forecasting method on top of VAE equipped with the diffusion model. Meanwhile, DiffusionTS (Yuan & Qiao, 2024) incorporates decomposition into the diffusion model to improve interoperability. Although TSDiff (Kollovieh et al., 2023) build a diffusion pipeline for multiple tasks with refinement, they still train different models for each task. Based on our knowledge, no unified diffusion transformer model has yet been explored for a comprehensive set of time series tasks. For a thorough literature review on diffusion models in time series analysis, please refer to (Yang et al., 2024).

## 3 METHODOLOGY

In this section, we begin by establishing a uniform problem setting for multiple downstream tasks, followed by a comprehensive examination of our model architecture. We then explore the training pipeline with specialized mask strategies and observation conditions that enable effective self-supervised learning for time series data. Next, we demonstrate how to incorporate physics information during inference to generate samples that better align with real-world constraints. Building on this foundation, we introduce our plug-and-play physics injection component and provide theoretical

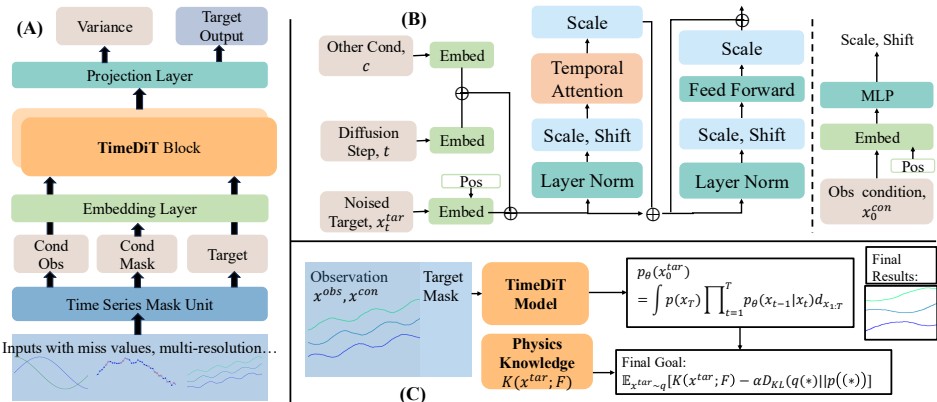

Figure 1: PINFDiT Architecture. (A): PINFDiT framework with diverse multivariate time series from different domains with multi-resolution or missing values; (B): Detailed structure of PINFDiT block; (C): Physics Injection: We employ physics knowledge as a plugin mechanism, injecting known physical residuals $F$ to refine predictions without requiring architectural modifications or retraining.

adjustments showing its adaptability across various models. These extensions highlight the flexibility of our proposed approach, establishing it as a powerful model suitable for diverse time series.

## 3.1 TIME SERIES DIFFUSION TRANSFORMERS

**Overview.** Figure 1 shows the overall framework of PINFDiT. Firstly, we establish $\mathbf{M_{obs}}$ and $\mathbf{x}_0^{obs}$ based on inputs with varying shapes, missing values, and multi-resolution data. Then, the unified time series mask unit constructs $\mathbf{M}$ and adapts to diverse scenarios, generating $\mathbf{x}_0^{con}$ and $\mathbf{x}_0^{tar}$ with shape $\mathbb{R}^{B \times L \times K}$, where $B$ is the batch size. Different from the channel-independent patching token approach (Nie et al., 2023; Liu et al., 2024; Shi et al., 2024) or vector quantization (Li et al., 2024), we adopt a "What You See Is What You Get" (WYSIWYG) design philosophy in our embedding layer, which directly maps tokens as direct, contiguous arrays of $\mathbf{x}_0^{con}$ and the noised $\mathbf{x}_0^{tar}$ to embedding vectors. After that, the output of the embedding layer is fed into PINFDiT's diffusion pipeline, where $\mathbf{x}_0^{con}$ is integrated to guide samples toward regions of high classifier likelihood, substantially improving output quality. Please refer to Section B for the mathematical preliminaries of the conditional diffusion framework and the derivation of the training objective.

**Time Series Mask Unit.** Our Time Series Mask Unit (TSMU) provides a powerful masking mechanism generating four specialized mask types: random mask $\mathbf{M}^R$, block mask $\mathbf{M}^B$, stride mask $\mathbf{M}^S$, and reconstruction mask $\mathbf{M}^{Rec}$. Leveraging diffusion models' inherent capability to progressively denoise targeted sequence regions with precise conditions, the TSMU enables selective focus on critical temporal patterns while maintaining contextual integrity. Furthermore, the unified masking approach eliminates the need for task-specific architectural modifications, enhancing the model's adaptability to diverse time series challenges.

**Transformer-based Condition Injection.** PINFDiT processes multivariate time series through a transformer architecture that receives embedded noisy targets $\mathbf{x}_t^{tar}$ and conditional observations $\mathbf{x}_t^{con}$. Improving upon previous approaches (Peebles & Xie, 2022; Lu et al., 2024), we directly inject the diffusion time step $t$ into the target noise representation as it provides unit temporal information across the entire series. For observation conditioning, rather than using simple concatenation (Rombach et al., 2022), we implement adaptive layer normalization (AdaLN):

$$\text{AdaLN}(h, c) = c_{scale}\text{LayerNorm}(h) + c_{shift}, \tag{1}$$

where $h$ represents the hidden state output from the previous layer. The parameters $c_{\text{scale}}$ and $c_{\text{shift}}$ are derived from $\mathbf{x}_0^{con}$ (Figure 1(B)) and are essential for preserving temporal continuity and progression in the reconstructed series.

---

**Algorithm 1** Physics-Informed Energy-based Sampling

---

1: $\boldsymbol{x}_T \sim \mathcal{N}(\mathbf{0}, \boldsymbol{I})$
2: **for** $t = T, \ldots, 1$ **do**
3:   $\boldsymbol{z} \sim \mathcal{N}(\mathbf{0}, \boldsymbol{I})$ if $t > 1$, else $\boldsymbol{z} = \mathbf{0}$
4:   $\boldsymbol{x}_{t-1} = \frac{1}{\sqrt{\alpha_t}} \left( \boldsymbol{x}_t - \frac{1-\alpha_t}{\sqrt{1-\bar{\alpha}_t}} \boldsymbol{\epsilon}_\theta(\boldsymbol{x}_t, t) \right) + \sigma_t \boldsymbol{z}$
5: **end for**
6: **for** $j = 0, 1, \ldots, k-1$ **do**
7:   $\boldsymbol{x}_{j+1}^{tar} = \boldsymbol{x}_j^{tar} + \epsilon \nabla K(\boldsymbol{x}_j^{tar}; \boldsymbol{x}^{obs}) + \alpha\epsilon \nabla \log p(\boldsymbol{x}_j^{tar}|\boldsymbol{x}^{obs}) + \sqrt{2\epsilon}\sigma, \sigma \sim \mathcal{N}(0,1)$
8: **end for**
9: **return** $\boldsymbol{x}_k^{tar}$

---

## 3.2 Physics-Informed Sampling

Physics principles fundamentally govern the evolution of temporal signals in scientific phenomena like climate patterns and oceanographic data. In this section, we develop a decoding method ensuring that $\mathbf{x}^{\text{tar}}$ generated by PINFDiT satisfies physical law constraints. Unlike the previous SBI solution (Xiong et al., 2025; Linhart et al., 2024), which requires access to a faithful, differentiable simulator, we target the state-trajectory distribution directly by incorporating physics knowledge as an energy-based prior during inference. This follows the paradigm of separating general model learning from targeted physical refinement at inference time, thus can be treated as a plugin mechanism requiring no architectural modifications or retraining.

**Generalized Goal of Physics Injection for Time Series:** We first start with a brief introduction to physical laws and PDE. A generic form of a physical law represented as a PDE that describes the evolution of a continuous temporal signal $\mathbf{x}(\mathbf{u}, \tau)$ over a spatial coordinate $\mathbf{u}$ and time $\tau$ is given by:

$$\frac{\partial \mathbf{x}}{\partial \tau} = F(\tau, \mathbf{x}, \mathbf{u}, \frac{\partial \mathbf{x}}{\partial \mathbf{u}_i}, \frac{\partial^2 \mathbf{x}}{\partial \mathbf{u}_i \partial \mathbf{u}_j}, \ldots) \tag{2}$$

Based on this PDE representation of physical knowledge, the consistency between the predicted time series $\mathbf{x}^{\text{tar}}$ and the physics knowledge can be quantified using the following squared residual function:

$$K(\mathbf{x}^{\text{tar}}; F) = -||\frac{\partial \mathbf{x}^{\text{tar}}}{\partial \tau} - F(\tau, \mathbf{x}^{\text{tar}}, \mathbf{u}, \frac{\partial \mathbf{x}^{\text{tar}}}{\partial \mathbf{u}_i}, \frac{\partial^2 \mathbf{x}^{\text{tar}}}{\partial \mathbf{u}_i \partial \mathbf{u}_j}, \ldots)||_2^2 \tag{3}$$

This function reaches its maximum when the predicted time series is perfectly consistent with the physical model, resulting in a residual of 0. Using this metric $K$, physics knowledge can be integrated into a probabilistic time series foundation model $p(\mathbf{x}^{\text{tar}}|\mathbf{x}^{\text{con}})$ as an explicit regularization by solving the following optimization problem to obtain a refined model $q(\mathbf{x}^{\text{tar}}|\mathbf{x}^{\text{con}})$:

$$q(\mathbf{x}^{\text{tar}}|\mathbf{x}^{\text{con}}) = \arg\max_q \left[ \mathbb{E}_{\mathbf{x}^{\text{tar}} \sim q} K(\mathbf{x}^{\text{tar}}; F) - \alpha D_{KL}(q(\mathbf{x}^{\text{tar}}|\mathbf{x}^{\text{con}})||p(\mathbf{x}^{\text{tar}}|\mathbf{x}^{\text{con}})) \right] \tag{4}$$

where the first term represents the aforementioned physics knowledge metric, and the second term controls the divergence between $q(\mathbf{x}^{\text{tar}}|\mathbf{x}^{\text{con}})$ and $p(\mathbf{x}^{\text{tar}}|\mathbf{x}^{\text{con}})$.

**Closed-Form Langevin Correction via Boltzmann Energy Distribution:** However, directly updating the model parameters to optimize the above function is resource-consuming. To solve this issue, we derive the closed-form solution, which does not need to update the model parameters. The above optimization problem has a closed-form solution as provided by the following theorem:

**Theorem 3.1** (Boltzmann Energy Distribution). *The optimal $q(\mathbf{x}^{tar}|\mathbf{x}^{con})$ in Eq.4 is the Boltzmann distribution defined on the following energy function: $E(\mathbf{x}^{tar}; \mathbf{x}^{con}) = K(\mathbf{x}^{tar}; F) + \alpha \log p(\mathbf{x}^{tar}|\mathbf{x}^{con})$, in other words, the optimal $q(\mathbf{x}^{tar}|\mathbf{x}^{con})$ is:*

$$q(\mathbf{x}^{tar}|\mathbf{x}^{con}) = \frac{1}{Z} \exp(K(\mathbf{x}^{tar}; F) + \alpha \log p(\mathbf{x}^{tar}|\mathbf{x}^{con})), \tag{5}$$

*where $Z = \int \exp(K(\mathbf{x}^{tar}; F) + \alpha \log p(\mathbf{x}^{tar}|\mathbf{x}^{con}))d\mathbf{x}^{tar}$ is the partition function.*

The theorem illustrates that sampling from the Boltzmann distribution is analogous to incorporating physics knowledge into model edition. In the context of diffusion models, this distribution can be

effectively sampled using Langevin dynamics (Stoltz et al., 2010):

$$
\begin{aligned}
\mathbf{x}^{\text{tar}}_{j+1} &= \mathbf{x}^{\text{tar}}_j + \epsilon \nabla \log q(\mathbf{x}^{\text{tar}}|\mathbf{x}^{\text{con}}) + \sqrt{2\epsilon}\sigma, \sigma \sim \mathcal{N}(0,1) \\
&= \mathbf{x}^{\text{tar}}_j + \epsilon \nabla K(\mathbf{x}^{\text{tar}}_j; \mathbf{x}^{\text{con}}) + \alpha\epsilon \nabla \log p(\mathbf{x}^{\text{tar}}_j|\mathbf{x}^{\text{con}}) + \sqrt{2\epsilon}\sigma, \sigma \sim \mathcal{N}(0,1)
\end{aligned}
\tag{6}
$$

where $\sigma \sim \mathcal{N}(0,1)$. In diffusion model, precisely calculate the likelihood $\log p(\mathbf{x}^{\text{tar}}|\mathbf{x}^{\text{con}})$ is intractable. To tackle this issue, following previous works (Kollovieh et al., 2023), we approximate likelihood with the objective to edit the pre-trained diffusion model: $\log p(\mathbf{x}^{\text{tar}}|\mathbf{x}^{\text{con}}) = -\mathbb{E}_{\epsilon,t}[||\epsilon_\theta(\mathbf{x}^{\text{tar}}, t; \mathbf{x}^{\text{con}}) - \epsilon||^2]$. The approximation presented above constitutes the optimizable component of the evidence lower bound(ELBO). Algorithm 1 summarizes the comprehensive model editing process. By integrating physics knowledge through Langevin dynamics during inference, we achieve a physics-consistent refinement without compromising the model's ability to learn from data and generate diverse, physically plausible scenarios.

We now present theoretical guarantees for our diffusion model's convergence properties and it can be easily extended to other models:

**Theorem 3.2** (Physics-Informed Inference Plugin Convergence). *Let $p_\theta(\mathbf{x}^{tar}|\mathbf{x}^{con})$ be the conditional distribution defined by a pre-trained diffusion model with score function $\epsilon_\theta$. Let $F$ represent a physical law with residual function $K(\mathbf{x}^{tar}; F)$ in Eq.(3). For the physics-informed plugin with step size $\epsilon$ and $N$ refinement steps, the samples of Langevin dynamics Eq.(6) converge to the optimal goal of Eq.(4), with a convergence rate of:*

$$
D_{\text{KL}}(q_N \,\|\, q^*) \leq \mathcal{O}\left(\frac{d}{\sqrt{N}} + \varepsilon^2_{score}\right),
\tag{7}
$$

*where $q_N$ is the distribution after $N$ refinement steps and $d$ is the effective dimension of $\mathbf{x}^{tar}$ state space, $\varepsilon^2_{score}$ represents the squared error in the score estimation.*

When we set the step size (line 7 in Algo. 1) $\epsilon = \Theta(N^{-1/2})$ (Dalalyan & Karagulyan, 2019), the converge rate can be $\mathcal{O}(N^{-1/2})$. The critical connection between statistical convergence and physical accuracy is established in Lemma 3.3, which demonstrates that improvements in KL divergence directly translate to enhanced physical consistency:

**Lemma 3.3** (Residual–Variance Coupling). *Let $\widetilde{r} = \partial_t \mathbf{x} - F(\cdot)$ be the physical residual of any sample $\mathbf{x} \sim q$. If $F$ is $L$-Lipschitz and the surrogate bias is $\delta$, then for all $q$ absolutely continuous w.r.t. $q^*$, $\text{Var}_q[\widetilde{r}] \leq 2L^2 D_{\text{KL}}(q \,\|\, q^*) + 4L^2\delta^2$.*

We establish a direct relationship between statistical convergence and physical consistency: each $\sqrt{N}$-step improvement in KL divergence systematically reduces the variance of the physics residual, ensuring that our refinement process progressively enhances the physical validity of our predictions. This relationship offers practitioners a clear interpretability pathway: improvements in model convergence directly translate to enhanced physical consistency, allowing users to understand how refinement steps progressively reduce violations of physical laws. We extend the theorems to establish that $p_M(\mathbf{x}^{\text{tar}}|\mathbf{x}^{\text{con}})$ can represent the conditional distribution defined by other models $M$, making our approach model-agnostic. The complete model-agnostic proofs are provided in Appendix A.

## 4 EXPERIMENTS

### 4.1 PHYSICS-GUIDED SCIENTIFIC TIME SERIES FORECASTING

We evaluate our model's ability to incorporate physics knowledge into time series forecasting across six PDE simulators with multivariate 1D systems modeling incompressible fluid dynamics. These systems are governed by the Advection, Burgers, Navier-Stokes, Diffusion-Sorption, Vorticity, and Computational Fluid Dynamics equations, with the latter two reserved for further analysis. As shown in Table 1, PINFDiT consistently outperforms all competing approaches across nearly all metrics and equations. Our comprehensive comparison includes pre-trained foundation models (TimesFM, Moirai, Chronos), state-of-the-art linear/transformer/diffusion-based deep learning architectures (DLinear, PatchTST, CSDI, DiffusionTS), physics-informed models with embedded equations (NeuralODE, NeuralCDE), and inference-time physics-guided approaches (SNPE, LFBC), included to benchmark against methods that similarly enforce physical consistency via simulators during inference. PINFDiT

Table 1: Comparison of model performance across different physical systems using both deterministic metrics for *accuracy* and probabilistic metrics (CRPS) for *uncertainty quantification* evaluation. **Bold** indicates best result, Underline indicates the second best result. Same as below.

| Model | Advection | | | Burgers | | | Navier-Stokes | | | Diffusion-Sorption | | |
|---|---|---|---|---|---|---|---|---|---|---|---|---|
| | RMSE | MAE | CRPS | RMSE | MAE | CRPS | RMSE | MAE | CRPS | RMSE | MAE | CRPS |
| TimesFM | 0.0602 | 0.0428 | 0.0640 | 0.0278 | 0.0164 | 0.0822 | 0.0100 | 0.0076 | 0.0065 | 0.0407 | 0.0226 | 0.0279 |
| Moirai-Small | 0.1033 | 0.0738 | 0.1165 | 0.1432 | 0.1193 | 0.5057 | 0.0368 | 0.0250 | 0.0243 | 0.0144 | 0.0096 | 0.0105 |
| Moirai-Base | 0.0669 | 0.0481 | 0.0731 | 0.0972 | 0.0824 | 0.3701 | 0.0147 | 0.0107 | 0.0097 | 0.0044 | 0.0034 | 0.0037 |
| Moirai-Large | 0.0666 | 0.0481 | 0.0721 | 0.0365 | 0.0254 | 0.1227 | 0.0145 | 0.0097 | 0.0089 | 0.0032 | 0.0022 | 0.0025 |
| Moirai-MoE | 0.0463 | 0.0300 | 0.0458 | 0.0275 | 0.0164 | 0.0827 | 0.0273 | 0.0199 | 0.0162 | 0.0023 | 0.0016 | 0.0019 |
| Chronos-T5-B | 0.0414 | 0.0258 | 0.0390 | 0.0202 | 0.0100 | 0.0474 | 0.0081 | 0.0055 | 0.0047 | **0.0019** | **0.0015** | **0.0018** |
| Chronos-T5-S | 0.0334 | 0.0218 | 0.0330 | 0.0424 | 0.0304 | 0.1366 | 0.0121 | 0.0080 | 0.0066 | 0.0028 | 0.0022 | 0.0023 |
| Chronos-Bolt-T | 0.0919 | 0.0691 | 0.1056 | 0.0776 | 0.0703 | 0.3232 | 0.0465 | 0.0366 | 0.0298 | 0.3995 | 0.1895 | 0.2235 |
| Chronos-bolt-B | 0.1009 | 0.0706 | 0.1029 | 0.0205 | 0.0131 | 0.0634 | 0.0365 | 0.0285 | 0.0228 | 0.1375 | 0.0545 | 0.0792 |
| Chronos-bolt-S | 0.0678 | 0.0534 | 0.0776 | 0.0445 | 0.0351 | 0.1585 | 0.0378 | 0.0329 | 0.0261 | 0.1771 | 0.0717 | 0.0874 |
| DLinear | 0.3674 | 0.2943 | 0.4248 | 0.0287 | 0.0216 | 0.0975 | 0.9321 | 0.9101 | 0.6977 | 0.7898 | 0.7384 | 0.6754 |
| PatchTST | 0.3767 | 0.3026 | 0.4410 | 0.0271 | 0.0208 | 0.0953 | 0.8972 | 0.8861 | 0.6928 | 0.7698 | 0.7197 | 0.6549 |
| CSDI | 0.0118 | 0.0154 | 0.0176 | 0.0167 | 0.0076 | 0.0178 | 0.0094 | 0.0060 | 0.0055 | 0.0012 | 0.0008 | 0.0013 |
| DiffusionTS | 0.3393 | 0.2484 | 0.3540 | 0.0361 | 0.0263 | 0.1363 | 0.8267 | 0.6192 | 0.5063 | 0.0469 | 0.0358 | 0.0384 |
| NeuralODE | 0.3667 | 0.2968 | 0.4268 | 0.0284 | 0.0214 | 0.0966 | 0.9314 | 0.9045 | 0.6965 | 0.7860 | 0.7290 | 0.6707 |
| NeuralCDE | 0.3690 | 0.3021 | 0.4314 | 0.0292 | 0.0221 | 0.1001 | 0.9318 | 0.9051 | 0.6972 | 0.7849 | 0.7327 | 0.6706 |
| SNPE | 0.0529 | 0.0422 | 0.0667 | 0.0472 | 0.0373 | 0.1903 | 0.1281 | 0.1134 | 0.0912 | 0.0557 | 0.0451 | 0.0482 |
| LFBC | 0.0352 | 0.0280 | 0.0512 | 0.0382 | 0.0287 | 0.1615 | 0.0739 | 0.0576 | 0.0491 | 0.0359 | 0.0287 | 0.0361 |
| TSDiff | 0.5128 | 0.5612 | 1.0000 | 0.1949 | 0.1670 | 0.9999 | 0.9212 | 0.9077 | 0.9999 | 0.7696 | 0.7373 | 1.0000 |
| PINFDiT(w/o Phys) | 0.0052 | 0.0040 | 0.0061 | 0.0136 | 0.0036 | 0.0163 | 0.0039 | 0.0031 | 0.0028 | 0.0057 | 0.0045 | 0.0050 |
| PINFDiT | **0.0039** | **0.0032** | **0.0050** | **0.0133** | **0.0035** | **0.0159** | **0.0037** | **0.0030** | **0.0027** | 0.0052 | 0.0041 | 0.0049 |

achieves remarkable improvements: compared to the best foundation model (Chronos-T5-B), we reduce RMSE by 88.3% on Advection 35.1% on Burgers, and 54.3% on Navier-Stokes. Traditional deep learning models struggle with limited training data from the simulator, failing to learn meaningful representations for physics forecasting, as evidenced by their high error rates. Physics-informed neural networks that incorporate equations during training face similar data efficiency challenges. Among simulation-based inference methods, likelihood-free LFBC outperforms SNPE by 33.5% on Advection, highlighting the difficulty of accurate likelihood estimation. PINFDiT, with its direct physics knowledge injection through closed-form Langevin dynamics, achieves superior performance without requiring retraining or architecture modifications, demonstrating the effectiveness of our energy-based physics guidance approach.

Table 2: ERA5 climate forecasting (2-meter temperature), PINFDiT achieves higher ACC.

| Model | 6h | 12h | 18h | 24h |
|---|---|---|---|---|
| NODE | 0.82 | 0.68 | 0.69 | 0.79 |
| ClimaX | 0.92 | 0.90 | 0.88 | 0.89 |
| ClimODE | 0.97 | 0.96 | 0.96 | 0.96 |
| PINFDiT (w/o Phys) | 0.985 | 0.984 | 0.986 | 0.985 |
| **PINFDiT (Full)** | **0.987** | **0.987** | **0.987** | **0.987** |

In addition, we also added a real-world experiment on climate forecasting. We use the ERA5 dataset for 2-meter temperature prediction and apply our PINFDiT framework, enforcing physical consistency via Navier-Stokes-based constraints during inference. Our experiments, evaluated using the Anomaly Correlation Coefficient (ACC) (Li et al., 2025) across multiple lead times, demonstrate that PINFDiT outperforms strong baselines and its own ablations, achieving state-of-the-art accuracy on this challenging weather prediction task. These results highlight the effectiveness of our plug-and-play physics-injection module for improving real-world climate forecasts, particularly under data scarcity and complex dynamics.

## 4.2 PRACTICAL FORECASTING WITH MISSING DATA, IRREGULAR AND MULTI-RESOLUTION

To evaluate PINFDiT's performance in realistic scenarios, we conducted experiments incorporating three aforementioned challenges in real-world time series analysis across diverse domains from climate science to finance to healthcare: missing values (validated on Air Quality and MIMIC-III datasets), multi-resolution data (tested on NASDAQ), and irregularly sampled time series with varying time intervals between observations (evaluated on three PhysioNet configurations). This comprehensive testing ensures PINFDiT can handle both technical challenges and domain-specific nuances, validating its practical utility across diverse applications.

As shown in Table 3, PINFDiT consistently outperforms competing methods, achieving the best results in 19 out of 23 metrics across all datasets. For the Air Quality dataset, our model reduces MAE by 12.97% compared to the next best model (DiffTS), while simultaneously improving uncertainty

Table 3: Forecasting results on practical scenarios with both deterministic metrics (MAE/MSE) for *accuracy* evaluation and probabilistic metrics (CRPS/CRPS_sum) for *uncertainty quantification*.

| | Air Quality | | MIMIC-III | | NASDAQ | |
|---|---|---|---|---|---|---|
| | MAE/MSE | CRPS/CRPS_sum | MAE/MSE | CRPS/CRPS_sum | MAE/MSE | CRPS |
| DLinear | 0.683/0.685 | 0.662/0.544 | 0.786/1.000 | 0.770/0.748 | 2.715/8.137 | 0.342 |
| Neural ODE | 0.678/0.679 | 0.657/0.529 | 0.784/0.999 | 0.769/0.733 | 3.227/11.155 | 0.426 |
| Neural CDE | 0.683/0.685 | 0.659/0.551 | 0.787/1.002 | 0.771/0.754 | 3.319/11.816 | 0.439 |
| PatchTST | 0.685/0.683 | 0.664/0.564 | 0.778/0.987 | 0.771/0.721 | 3.182/10.635 | 0.410 |
| GPT4TS | 0.696/0.701 | 0.666/0.584 | 0.750/0.921 | 0.751/0.690 | 3.176/10.873 | 0.419 |
| CSDI | 0.539/0.554 | 0.598/0.620 | 0.551/0.681 | **0.504**/0.798 | 0.524/**0.388** | 0.096 |
| DiffTS | 0.521/0.538 | 0.649/0.719 | 0.677/0.908 | 0.633/0.676 | 1.951/9.515 | 0.283 |
| TimeMixer | 0.691/0.697 | 0.667/0.576 | 0.769/0.981 | 0.776/0.724 | 3.267/11.511 | 0.432 |
| TimeLLM | 0.701/0.705 | 0.664/0.571 | 0.787/1.020 | 0.785/0.700 | 3.125/10.276 | 0.405 |
| PINFDiT | **0.457/0.354** | **0.554/0.522** | **0.517/0.534** | 0.599/**0.649** | **0.516**/0.418 | **0.091** |

| | PhysioNet(a) | | PhysioNet(b) | | PhysioNet(c) | |
|---|---|---|---|---|---|---|
| | MAE/MSE | CRPS/CRPS_sum | MAE/MSE | CRPS/CRPS_sum | MAE/MSE | CRPS/CRPS_sum |
| DLinear | 0.686/0.758 | 0.764/0.812 | 0.733/0.922 | 0.794/0.793 | 0.715/0.813 | 0.767/0.797 |
| Neural ODE | 0.685/0.756 | 0.763/0.806 | 0.732/0.918 | 0.792/0.789 | 0.713/0.811 | 0.765/0.793 |
| Neural CDE | 0.688/0.754 | 0.763/0.799 | 0.733/0.921 | 0.792/0.786 | 0.713/0.814 | 0.765/0.791 |
| PatchTST | 0.699/0.780 | 0.769/0.812 | 0.733/0.932 | 0.791/0.775 | 0.714/0.802 | 0.766/0.777 |
| GPT4TS | 0.697/0.772 | 0.767/0.809 | 0.734/0.921 | 0.795/0.798 | 0.713/0.817 | 0.770/0.768 |
| CSDI | **0.548/0.548** | 0.620/0.641 | 0.665/0.792 | 0.725/0.787 | 0.665/0.695 | 0.669/0.748 |
| DiffTS | 0.610/0.742 | 0.628/0.668 | 0.701/0.880 | 0.720/0.724 | 0.678/0.872 | 0.679/0.719 |
| TimeMixer | 0.692/0.775 | 0.763/0.805 | 0.734/0.920 | 0.794/0.798 | 0.707/0.805 | 0.757/0.784 |
| TimeLLM | 0.687/0.761 | 0.752/0.797 | 0.731/0.931 | 0.795/0.795 | 0.713/0.800 | 0.757/0.754 |
| PINFDiT | 0.577/0.620 | **0.616/0.640** | **0.659/0.766** | **0.708/0.710** | **0.543/0.561** | **0.668/0.708** |

quantification with 7.3% lower CRPS. On MIMIC-III, PINFDiT achieves a 6.17% reduction in MSE over the previous state-of-the-art (CSDI). The model's strong performance extends to financial time series, where it delivers a 1.53% improvement in MAE on the NASDAQ dataset. Most notably, on the challenging PhysioNet(c) configuration with irregular sampling, PINFDiT reduces MSE by 19.28% compared to the next best approach. These results demonstrate that PINFDiT not only achieves high accuracy in point forecasts but also provides well-calibrated probabilistic forecasts, effectively capturing the inherent uncertainties in complex time series data without requiring specialized designs for interpolation or imputation. This robust uncertainty quantification capability positions PINFDiT as a powerful tool for decision-making in uncertain environments across various domains.

## 4.3 GENERATIVE ABILITY ON SYNTHETIC DATA GENERATION AND IMPUTATION

Table 4: Imputation results averaged over four mask ratios. Each cell is MSE/MAE.

| Methods | ETTh1 | ETTh2 | Weather | Electricity |
|---|---|---|---|---|
| PINFDiT_scratch | 0.042/0.133 | 0.040/**0.130** | 0.031/**0.036** | **0.069/0.173** |
| iTransformer | 0.149/0.270 | 0.150/0.271 | 0.053/0.116 | 0.099/0.224 |
| GPT4TS | 0.069/0.173 | 0.048/0.141 | 0.031/0.056 | 0.090/0.207 |
| TimesNet | 0.078/0.187 | 0.049/0.146 | **0.030**/0.054 | 0.092/0.210 |
| PatchTST | 0.115/0.224 | 0.065/0.163 | 0.034/0.055 | 0.072/0.183 |
| Timer | 0.145/0.243 | 0.077/0.172 | 0.108/0.168 | 0.097/0.194 |
| DiffTS | 0.040/0.134 | **0.029**/0.113 | 0.035/0.079 | 0.075/0.165 |
| PINFDiT | **0.036/0.122** | 0.031/**0.111** | 0.031/**0.036** | 0.068/0.172 |

Table 5: Synthetic generation on 24-length multivariate time series.

| Metric | Methods | Sine | Stocks | AirQ | Energy |
|---|---|---|---|---|---|
| DS | TimeGAN | 0.1217 | 0.2038 | 0.3913 | 0.4969 |
| | TimeVAE | 0.0489 | 0.1987 | 0.2869 | 0.4993 |
| | Diff-TS | 0.0099 | 0.1869 | **0.1227** | 0.2301 |
| | PINFDiT | **0.0086** | **0.0087** | 0.1923 | **0.0053** |
| PS | TimeGAN | 0.2797 | 0.0481 | 0.035 | 0.3305 |
| | TimeVAE | 0.2285 | 0.0485 | 0.0269 | 0.2878 |
| | Diff-TS | 0.2262 | **0.042** | 0.022 | 0.2506 |
| | PINFDiT | **0.1915** | 0.0445 | **0.0217** | **0.2489** |

**Synthetic Generation Task:** High-quality synthetic data generation is particularly crucial in scientific domains as it enables privacy-preserving research on sensitive datasets, facilitates model development in data-scarce fields. We conduct experiments to synthesize multivariate time series and evaluate performance using the discriminative score (DS) and predictive score (PS) metrics (Yuan & Qiao, 2024). Table 5 demonstrates PhysDiffT's superior synthetic generation capabilities, with additional results using limited training data (5% and 10%) presented in Table 17, consistently producing more realistic samples than baselines even on challenging energy datasets. This demonstrates PINFDiT's strength in complex time series synthesis. Both qualitative and quantitative results confirm PINFDiT's superior ability to model intricate characteristics for realistic multidimensional time series.

**Imputation Task:** Imputation is essential in scientific research as it enables working with incomplete data due to sensor failures, sampling constraints, or measurement errors—preserving valuable datasets that would otherwise be unusable and maintaining statistical power in experiments where data collection is costly or time-limited. We conduct experiments on six benchmark time-series datasets: ETTh1, ETTh2, Electricity, and Weather. We use random mask ratios $\{12.5\%, 25\%, 37.5\%, 50\%\}$ following previous studies' settings (Wu et al., 2023). Table 4 shows the imputation result averaged over the four mask ratios. PINFDiT demonstrates superior performance. Notably, PINFDiT achieved a 39% reduction in MSE and 22% reduction in MAE compared to the strongest baseline on the ETTh1 dataset. For full results on each mask ratio, please refer to Section C.4.

For further anomaly detection task, please refer to Section C.3.

### 4.4 GENERABILITY ON ZERO-SHOT SETTING AND ABLATION STUDY

Table 6: Forecasting results on $\mathrm{CRPS_{sum}}$. LagLL notes for LagLLama; TMixer notes for TimeMixer; TLLM notes for TimeLLM.

| Dataset | PINFDiT | Moirai (S/B/L) | LagLL. | TMixer | TLLM |
|---|---|---|---|---|---|
| Solar | **0.424** | 0.884 / 0.948 / 1.042 | 0.690 | 0.999 | 0.997 |
| Electricity | **0.030** | 0.079 / 0.072 / 0.039 | 0.065 | 0.302 | 0.303 |
| Traffic | 0.351 | 0.215 / 0.191 / **0.111** | 0.275 | 0.403 | 0.368 |
| Taxi | **0.392** | 0.463 / 0.428 / 0.597 | 0.620 | 0.785 | 0.782 |
| Exchange | 0.019 | **0.007** / 0.012 / 0.011 | 0.024 | 0.079 | 0.076 |

**Zero-shot forecasting:** Zero-shot capability is crucial in scientific domains where data collection is costly, time-consuming, or impossible for new scenarios. Building upon previous success, we extended our investigation to zero-shot probabilistic forecasting, where PINFDiT demonstrates remarkable capabilities through our easy-to-implement pre-trained model design that employs masking strategies to accommodate uniform inputs across diverse datasets. As evidenced in Table 6, the PINFDiT achieves superior performance compared to models with fewer parameters in equivalent zero-shot settings, outperforming state-of-the-art models on Solar, Electricity, and Taxi datasets with significant margins. These results establish PINFDiT as a significant advancement in time series modeling, particularly for applications requiring robust uncertainty quantification in novel scientific contexts.

**Ablation Study:** As shown in Table 7, As shown in Table 7, we conduct comprehensive ablation studies to validate the effectiveness of modelname's key components. We evaluate several variants: modelnamespace without physics injection (w/o Phys); without random masking (w/o RM), stride masking (w/o SM), and block masking (w/o BM); alternative attention mechanisms including dual-attention (DA) and channel-wise attention (CW) compared to our temporal attention; different embedding approaches with patch tokens (PT) versus our direct embedding; and various observation conditioning methods via feature addition (Add) and cross attention (CA) versus our AdaLN approach. The full PINFDiT model achieves superior performance (MSE of 0.424 for Solar and 0.030 for Electricity datasets) compared to its variants. Among the masking strategies, stride masking proves to be the most crucial component, as its removal leads to substantial performance degradation (MSE increasing to 0.862 for Solar and 0.101 for Electricity). In the attention mechanism analysis, our proposed temporal-attention architecture demonstrates consistent advantages over alternative designs, with patch token attention yielding the least favorable results (MSE of 0.874 and 0.145 for Solar and Electricity, respectively). Furthermore, the conditional AdaLN design emerges as the optimal choice among various conditioning approaches, reinforcing our architectural decisions.

Table 7: Ablation study on zero-shot forecasting

| Dataset | PINFDiT | w/o Phys | w/o RM. | w/o SM. | w/o BM. |
|---|---|---|---|---|---|
| Solar | **0.424** | 0.445 | 0.465 | 0.469 | 0.862 |
| Electricity | **0.030** | 0.033 | 0.035 | 0.037 | 0.101 |

| Dataset | DA | CW | PT | Add | CA |
|---|---|---|---|---|---|
| Solar | 0.467 | 0.461 | 0.874 | 0.677 | 0.711 |
| Electricity | 0.037 | 0.039 | 0.145 | 0.079 | 0.077 |

## 5 CONCLUSION

In this paper, we introduce PINFDiT, a pioneering approach to creating a versatile and robust foundation model for various time series tasks under practical scenarios. By integrating transformer inductive bias with diffusion model, PINFDiT effectively captures temporal dependencies and addresses real world challenges unique to time series regarding multi-resolution and missing values as well as incorporating external knowledge. Our innovative masking strategies allow for a consistent training framework adaptable to diverse tasks such as forecasting, imputation, anomaly detection, and

synthetic data generation, positioning PINFDiT as a bedrock of a foundation model. We recognize some limitations of current work: first, we primarily explored common sequence lengths and did not assess PINFDiT's performance on very long sequences. While we have introduced randomness in prediction length and feature numbers up to a maximum, we aim to develop more scalable solutions for highly variable multivariate time series. second, there is a high demand for deeply developing foundation models for multi-modal time series, allowing PINFDiT to utilize diverse data sources for enhanced performance.

## ACKNOWLEDGEMENT

This work is partially supported by the NSF Award #2425919, and NSF Award #2413417. The funding from these sources has been a cornerstone in enabling us to bring our project to fruition. We are also deeply grateful to the anonymous reviewers for their rigorous review process. Their detailed comments and constructive suggestions have significantly contributed to the improvement of this paper.

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

# Appendix

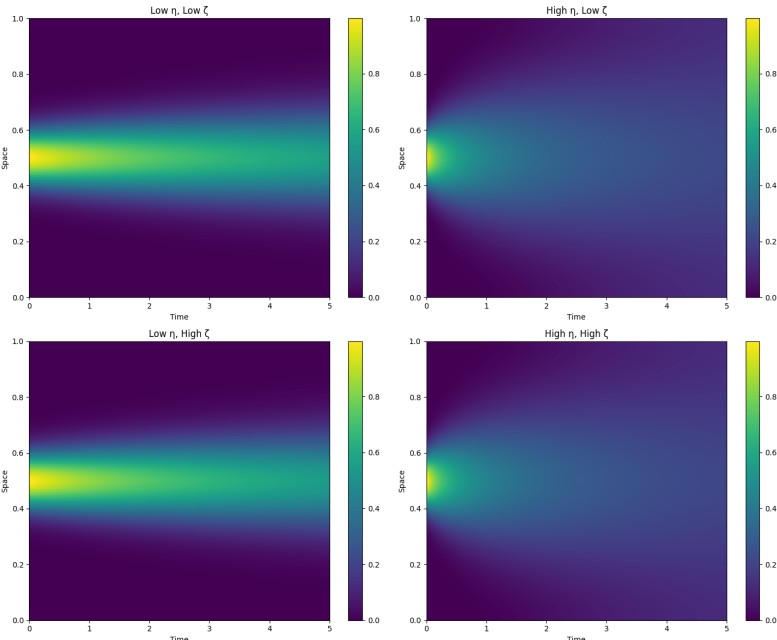

Figure 2: The two key parameters that govern CFD behavior: diffusion coefficient ($\eta$) and oscillation frequency ($\zeta$). Higher $\eta$ values produce smoother solutions with increased diffusion, while lower values create sharper gradients; higher $\zeta$ values generate higher frequency oscillations with smaller structures, while lower values result in lower frequency oscillations with larger structures.

## A    DISCUSSION ON PHYSICS-INFORMED PINFDIT

The tension between physical constraints and learned distributions in PINFDiT is managed through a sophisticated energy-based optimization framework that combines two key components:

- the physics knowledge represented by function $K(x^{\text{tar}}; F)$, which measures PDE residuals for physical law conformity
- the learned probabilistic distribution $p(x^{\text{tar}}|x^{\text{con}})$ from the diffusion model

This balance is achieved through an energy function:

$$E(x^{\text{tar}}; x^{\text{con}}) = K(x^{\text{tar}}; F) + \alpha \log p(x^{\text{tar}}|x^{\text{con}})$$

where the parameter $\alpha$ controls the trade-off between physical consistency and distribution fidelity.

Rather than directly modifying model parameters, PINFDiT implements this balance through an iterative sampling procedure that:

1. starts with samples from the learned distribution
2. gradually refines them using physical gradients while maintaining probabilistic characteristics

This approach allows the model to generate samples that respect both the learned patterns in the data and the underlying physical laws without significantly compromising either aspect, ultimately resolving the tension through a theoretically-grounded Boltzmann distribution as the optimal solution.

### A.1    GENERALIZATION ON PROPOSED PHYSICS INJECTION METHOD

To validate PINFDiT 's ability to capture complex physical dynamics, we evaluated its performance on zero-shot CFD prediction tasks across varying physical parameters. The demonstration of CFD

Table 8: PINFDiT's performance on Zero-shot CFD with different parameters.

| Model | MSE | RMSE | MAE | CRPS | CRPS_sum | MSE | RMSE | MAE | CRPS | CRPS_sum |
|---|---|---|---|---|---|---|---|---|---|---|
| | $\eta = 0.0010$(Low), $\zeta = 0.0010$(Low) | | | | | $\eta = 0.0100$(High), $\zeta = 0.0010$(Low) | | | | |
| DDPM | 0.0076 | 0.0874 | 0.0720 | 0.1012 | 0.0988 | 0.0035 | 0.0588 | 0.0485 | 0.0764 | 0.0622 |
| DDPM_Std | 0.0005 | 0.0009 | 0.0008 | 0.0002 | 0.0004 | 0.0001 | 0.0007 | 0.003 | 0.0002 | 0.0003 |
| DDIM | 0.0590 | 0.2428 | 0.2340 | 0.3663 | 0.3918 | 0.0370 | 0.1923 | 0.1843 | 0.3119 | 0.3320 |
| DDIM_Std | 0.0014 | 0.0017 | 0.024 | 0.0001 | 0.0005 | 0.001 | 0.0017 | 0.009 | 0.0002 | 0.0004 |
| TSDiff | 0.2844 | 0.5333 | 0.5256 | 1.0000 | 1.0000 | 0.2268 | 0.4762 | 0.4747 | 0.9999 | 0.9999 |
| TSDiff_Std | 0.0016 | 0.0017 | 0.007 | 0.0000 | 0.0000 | 0.002 | 0.002 | 0.01 | 0.0000 | 0.0000 |
| PINFDiT | 0.0046 | 0.0681 | 0.0558 | 0.0859 | 0.0755 | 0.0030 | 0.0547 | 0.0450 | 0.0714 | 0.0571 |
| PINFDiT_Std | 0.0003 | 0.0004 | 0.0005 | 0.0001 | 0.0002 | 0.0003 | 0.0003 | 0.0005 | 0.0002 | 0.0001 |
| | $\eta = 0.0010$(Low), $\zeta = 0.0100$(High) | | | | | $\eta = 0.0100$(High), $\zeta = 0.0100$(High) | | | | |
| DDPM | 0.0052 | 0.0719 | 0.0598 | 0.0910 | 0.0816 | 0.0042 | 0.0652 | 0.0530 | 0.0821 | 0.0719 |
| DDPM_Std | 0.0006 | 0.0007 | 0.0011 | 0.0004 | 0.0003 | 0.0004 | 0.001 | 0.0035 | 0.0004 | 0.0003 |
| DDIM | 0.0464 | 0.2154 | 0.2043 | 0.3436 | 0.3694 | 0.0401 | 0.2002 | 0.1901 | 0.3227 | 0.3421 |
| DDIM_Std | 0.0017 | 0.0021 | 0.016 | 0.0004 | 0.0002 | 0.0014 | 0.0019 | 0.0011 | 0.0002 | 0.0003 |
| TSDiff | 0.2497 | 0.4997 | 0.4858 | 0.9999 | 0.9999 | 0.2304 | 0.4800 | 0.4766 | 0.9999 | 0.9999 |
| TSDiff_Std | 0.0018 | 0.002 | 0.004 | 0.0000 | 0.0000 | 0.0012 | 0.0015 | 0.013 | 0.0001 | 0.0001 |
| PINFDiT | 0.0046 | 0.0679 | 0.0559 | 0.0857 | 0.0764 | 0.0038 | 0.0613 | 0.0493 | 0.0772 | 0.0661 |
| PINFDiT_Std | 0.0004 | 0.0004 | 0.0003 | 0.0002 | 0.0001 | 0.0002 | 0.0004 | 0.001 | 0.0002 | 0.0002 |

Figure 3: Empirical Convergence Analysis of NS and 1D-Vorticity Models. Both results demonstrate linear relationships consistent with theoretical expectations, with the NS simulation showing stronger linearity but higher absolute error values compared to the 1D-Vorticity simulation, reflecting differences in their underlying physical complexity.

flow can be found at Figure 2. We systematically tested two key parameters that govern fluid behavior: diffusion coefficient ($\eta$) and oscillation frequency ($\zeta$). Higher $\eta$ values produce smoother solutions with increased diffusion, while lower values create sharper gradients; higher $\zeta$ values generate higher frequency oscillations with smaller structures, while lower values result in lower frequency oscillations with larger structures. As shown in Table 8, PINFDiT consistently outperformed baseline models with different sampling strategies (DDPM, DDIM, and TSDiff) across all parameter configurations, demonstrating its robust ability to adapt to different physical regimes without retraining. Notably, PINFDiT maintained superior performance even in the most challenging scenario with high oscillation frequency ($\zeta = 0.0100$) and low diffusion ($\eta = 0.0010$), where sharp gradients and complex small-scale structures coexist. These results confirm PINFDiT 's effectiveness in incorporating physical constraints and capturing the underlying dynamics of complex fluid systems, highlighting its potential for scientific applications requiring accurate simulation of physical phenomena.

## A.2 EMPIRICAL RESULTS ON CONVERGENCE CHARACTERISTICS

The convergence plots in Figure 3 comparing the Navier-Stokes (NS) and 1D-Vorticity simulations demonstrate strong alignment with the theoretical $\mathcal{O}(N^{-1/2})$ convergence rate expected for numerical simulations. Both models exhibit a clear linear relationship when error metrics (MAE and MSE) are plotted against $N^{-1/2}$, as indicated by the red dashed trend lines. Notably, the NS simulation displays nearly perfect linear scaling for both MAE and MSE, suggesting the numerical errors decrease predictably as the number of time steps increases. In contrast, the 1D-Vorticity model shows a slight deviation from perfect linear convergence, with a flattening of the curve at intermediate values of $N$. This behavior indicates that the 1D-Vorticity simulation may be approaching a lower error bound more rapidly than the NS model or encountering additional error sources that aren't

purely time-step dependent. While both methods converge at the theoretical $\mathcal{O}(N^{-1/2})$ rate, the 1D-Vorticity simulation achieves lower absolute error with the same computational budget. These results confirm that both models follow the expected theoretical convergence rate, while revealing important differences in their numerical behavior that reflect the underlying physical complexity and implementation characteristics of each approach.

## A.3 ROBUSTNESS AND SENSITIVITY ANALYSIS

### A.3.1 SENSITIVITY TO LANGEVIN DYNAMICS REFINEMENT STEPS.

We further investigate the robustness of our physics-informed refinement module (Langevin dynamics) with respect to the number of refinement steps $K$. Table 9 reports the RMSE across three datasets for varying $K$. We observe that the performance consistently improves as $K$ increases from 10 to 20, reaching an optimal range. Crucially, the model remains stable for $K > 20$ without significant degradation or divergence. This demonstrates that our inference-time optimization is robust and does not require hyper-sensitive tuning of the step count to yield performance gains.

Table 9: **Robustness of Physics-Informed Refinement.** Comparison of RMSE performance across different numbers of Langevin refinement steps ($K$). The method achieves consistent improvements around 20–30 steps and maintains stability beyond that point.

| Dataset | $K = 10$ | $K = 20$ | $K = 30$ | $K = 40$ |
|---|---|---|---|---|
| Advection | 0.0047±0.0002 | **0.0039**±0.0002 | 0.0043±0.0003 | 0.0045±0.0004 |
| Burgers | 0.0143±0.0003 | 0.0136±0.0002 | **0.0133**±0.0002 | 0.0135±0.0002 |
| Navier-Stokes | 0.0041±0.0003 | **0.0037**±0.0001 | 0.0038±0.0001 | 0.0039±0.0002 |

### A.3.2 VALIDITY AND SPECIFICITY OF PHYSICS INJECTION.

To verify that our physics-injection module provides legitimate domain-specific guidance rather than merely acting as a generic smoothing filter, we conducted two sensitivity analyses. These experiments test the model's behavior when the injected physical laws are either mismatched to the data or applied to non-physical domains.

**Impact of Mismatched Physical Laws.** We analyzed the performance on the Advection dataset when injecting the correct physical law (Advection Equation) versus an incorrect law (Burgers' Equation). As shown in Table 10, utilizing the correct PDE significantly reduces the error (25% improvement). Conversely, injecting the incorrect Burgers' dynamics actively harms performance, increasing the RMSE by 17% compared to the baseline. This confirms that the performance gains are derived from the correct physical inductive bias and that the module is sensitive to the validity of the governing equation.

Table 10: **Robustness to Physical Mismatch.** Comparison of performance on Advection data when injecting correct vs. incorrect physical laws. Metrics reported are RMSE ($\downarrow$).

| Data Source | Physics Configuration | RMSE | Rel. Change |
|---|---|---|---|
| Advection | No Physics (Baseline) | 0.0052±0.0002 | – |
| | **Correct (Advection PDE)** | **0.0039**±0.0002 | **+25.0%** |
| | Incorrect (Burgers PDE) | 0.0061±0.001 | -17.3% |

**Application to Non-Physical Domains.** To investigate whether the Langevin correction acts as a spurious "smoothing" operator, we applied fluid dynamics constraints (Advection and Burgers) to the NASDAQ financial dataset, which possesses no underlying fluid physical basis. As presented in Table 11, the application of physics constraints yields no benefit and results in a slight degradation of Mean Absolute Error (MAE). This suggests that our method does not produce spurious positive transfer; it functions effectively only when the data supports the underlying physical hypothesis.

Table 11: **Sensitivity on Non-Physical Data.** Performance on financial data (NASDAQ) when forcing fluid dynamics constraints. Metrics reported are MAE ($\downarrow$).

| Data Source | Physics Configuration | MAE | Rel. Change |
|---|---|---|---|
| NASDAQ | No Physics (Baseline) | **0.516**±0.002 | – |
| | Advection PDE | 0.526±0.004 | -1.9% |
| | Burgers PDE | 0.537±0.005 | -4.0% |

**Generalization to Autoregressive Foundation Models.** To validate the model-agnostic capability of our framework, we integrated the physics-informed refinement module into **Chronos-T5-Small**, a leading autoregressive foundation model. We approximated the score function from the predicted quantiles and applied Langevin dynamics (Eq. 6) as a post-processing step on the Advection dataset. As reported in Table 12, the physics injection yields consistent improvements across all metrics (CRPS, MAE, RMSE). The inclusion of standard deviations (computed over 5 random seeds) confirms that these gains are statistically significant and robust, distinct from random sampling noise.

Table 12: **Model-Agnostic Refinement.** Performance comparison on the Advection dataset using the Chronos-T5-Small backbone. Results are reported as Mean $\pm$ Standard Deviation.

| Model | Method | CRPS ($\downarrow$) | MAE ($\downarrow$) | RMSE ($\downarrow$) |
|---|---|---|---|---|
| Chronos-T5-S | Baseline | $0.0330 \pm 0.0002$ | $0.0218 \pm 0.003$ | $0.0334 \pm 0.0008$ |
| | + Physics (Ours) | $\mathbf{0.0313} \pm 0.0001$ | $\mathbf{0.0197} \pm 0.002$ | $\mathbf{0.0322} \pm 0.0005$ |

### A.3.3 ROBUSTNESS TO PHYSICAL CONSTRAINT SPECIFICITY ON CLIMATE DATASET

To validate that our framework actively enforces specific physical dynamics rather than acting as generic regularization, we conducted a counter-factual experiment on the ERA5 dataset. We compared the standard PINFDiT (using the correct Temperature Advection-Diffusion equation) against a variant enforcing the *incorrect* Navier-Stokes Momentum conservation on the temperature field. As shown in Table 13, applying the wrong physical law significantly degrades performance (ACC drops from $\approx 0.985$ to $\approx 0.954$), confirming that the model is sensitive to the physical validity of the constraint. Conversely, the correct physics injection yields consistent improvements over the base model, demonstrating that our method successfully steers the generation toward physically consistent atmospheric states.

Table 13: **Physics Specificity Analysis.** Anomaly Correlation Coefficient (ACC) on ERA5 2-meter temperature forecasting. We compare the base model against injection of the correct physics (Advection) and incorrect physics (Navier-Stokes Momentum). Results are reported as Mean $\pm$ Std Dev.

| Model Variation | 6h | 12h | 18h | 24h |
|---|---|---|---|---|
| PINFDiT (w/o Phys) | $0.985 \pm 0.002$ | $0.984 \pm 0.003$ | $0.986 \pm 0.002$ | $0.985 \pm 0.002$ |
| PINFDiT (NS Momentum) | $0.954 \pm 0.004$ | $0.956 \pm 0.005$ | $0.955 \pm 0.004$ | $0.954 \pm 0.003$ |
| **PINFDiT (Advection)** | $\mathbf{0.987} \pm 0.003$ | $\mathbf{0.987} \pm 0.002$ | $\mathbf{0.987} \pm 0.001$ | $\mathbf{0.987} \pm 0.001$ |

### A.4 PROOF OF PHYSICS-INFORMED PINFDiT THEOREM 3.1

**Theorem A.1.** *The optimal* $q(\mathbf{x}^{tar}|\mathbf{x}^{con})$ *in Eq.3 is the Boltzmann distribution defined on the following energy function:*

$$E(\mathbf{x}^{tar}; \mathbf{x}^{con}) = K(\mathbf{x}^{tar}; F) + \alpha \log p(\mathbf{x}^{tar}|\mathbf{x}^{con}) \tag{8}$$

*in other words, the optimal* $q(\mathbf{x}^{tar}|\mathbf{x}^{con})$ *is:*

$$q(\mathbf{x}^{tar}|\mathbf{x}^{con}) = \frac{1}{Z} \exp(K(\mathbf{x}^{tar}; F) + \alpha \log p(\mathbf{x}^{tar}|\mathbf{x}^{con})), \tag{9}$$

*where* $Z = \int \exp(K(\mathbf{x}^{tar}; F) + \alpha \log p(\mathbf{x}^{tar}|\mathbf{x}^{con})) d\mathbf{x}^{tar}$ *is the partition function.*

*Proof.* Let us consider the objective function:

$$
\begin{aligned}
O(q(y|x)) &= \mathbb{E}_{y \sim q(y|x)} K(y) - \alpha D_{KL}(q(y|x) || p(y|x)) \\
&= \mathbb{E}_{y \sim q(y|x)} K(y) - \alpha \int_y q(y|x) \log(\frac{q(y|x)}{p(y|x)}) dy \\
&= \int_y q(y|x)[K(y) + \alpha \log p(y|x) - \alpha \log q(y|x)] dy
\end{aligned}
\tag{10}
$$

We try to find the optimal $q(y|x)$ through Lagrange multipliers. The constraint of the above objective function is that $q(y|x)$ is a valid $\int_y q(y|x) dy = 1$. Thus, the Lagrangian is:

$$
\begin{aligned}
L(q(y|x), \lambda) &= \int_y q(y|x)[K(y) + \alpha \log p(y|x) - \alpha \log q(y|x)] dy - \lambda(\int_y q(y|x) dy - 1) \\
&= \int_y q(y|x)[K(y) + \alpha \log p(y|x) - \alpha \log q(y|x) - \lambda q(y|x)] dy + \lambda
\end{aligned}
\tag{11}
$$

We define $f(q(y|x), y, \lambda) = q(y|x)[K(y) + \alpha \log p(y|x) - \alpha \log q(y|x) - \lambda] + \lambda h(y)]$, where $h(y)$ can be the density function of any fixed distribution defined on the support set of $y$. Therefore, $L(q(y|x), \lambda) = \int_y f(q(y|x), y, \lambda) dy$. According to Euler-Lagrange equation, when the above Lagrangian achieve extreme point, we have:

$$
\frac{\partial f}{\partial q} = K(y) + \alpha \log p(y|x) - \alpha \log q(y|x) - \lambda - \alpha = 0
\tag{12}
$$

Thus, we have:

$$
\begin{aligned}
\alpha \log q(y|x) &= K(y) + \alpha \log p(y|x) - \log q(y|x) - \lambda - \alpha \\
q(y|x) &= \exp(\frac{1}{\alpha} K(y) + \log p(y|x) - \frac{\lambda}{\alpha} - 1) \\
&= \frac{1}{\exp(\frac{\lambda}{\alpha} + 1)} \exp(\frac{1}{\alpha} K(y) + \log p(y|x))
\end{aligned}
\tag{13}
$$

Meanwhile, since $\int_y q(y|x) dy = 1$, we have:

$$
\begin{aligned}
\int_y \exp(\frac{1}{\alpha} K(y) + \log p(y|x) - \frac{\lambda}{\alpha} - 1) dy &= 1 \\
\frac{1}{\exp(\frac{\lambda}{\alpha} + 1)} \int_y \exp(\frac{1}{\alpha} K(y) + \log p(y|x)) dy &= 1
\end{aligned}
\tag{14}
$$

Thus, we have $\exp(\frac{\lambda}{\alpha} + 1) = \int_y \exp(\frac{1}{\alpha} K(y) + \log p(y|x)) dy = Z$, leading to:

$$
q(y|x) = \frac{1}{Z} \exp(K(y) + \alpha \log p(y|x)), Z = \int \exp(K(y) + \alpha \log p(y|x)) dy
\tag{15}
$$

Note that $F$ represents fixed knowledge for a given domain so it remains constant during the optimization process and doesn't need to be considered as a variable in the proof. $\square$

### A.5 PROOF OF PHYSICS-INFORMED REFINEMENT PLUGIN THEOREM 3.2

We now present a theoretical framework for applying physics-informed refinement to any time series prediction model, establishing its model-agnostic nature.

**Assumption 1** (Regularity Conditions). *We assume that $K(\mathbf{x}; F)$ and $\log p_M$ are L-smooth (have Lipschitz gradients) and satisfy a mild dissipativity/coercivity condition such as $\langle \mathbf{x}, \nabla U(\mathbf{x}) \rangle \geq m\|\mathbf{x}\|^2 - b$ for $U = -K - \alpha \log p_M$, where $m > 0$ and $b$ are constants.*

**Assumption 2** (Score Approximation). *For any model $M$, we assume the score function can be approximated such that $\|\nabla \log \hat{p}_M - \nabla \log p_M\|_2 \leq \varepsilon_{score}$.*

**Discussion of Regularity Constants ($m$, $b$, $L$).** In our theoretical analysis, we carefully distinguish between three key constants in the convergence of Langevin Monte Carlo (LMC) and plug-and-play refinement for sampling:

- $L$ **(Smoothness/Lipschitz Gradient):** The Lipschitz constant of the gradient $\nabla U(\mathbf{x})$, i.e., $\|\nabla U(\mathbf{x}) - \nabla U(\mathbf{y})\| \le L\|\mathbf{x} - \mathbf{y}\|$ for all $\mathbf{x}, \mathbf{y}$. This bounds the maximum curvature and determines the sensitivity of the potential's gradient, and is required for standard LMC convergence proofs.

- $m$ **(Strong Convexity/Coercivity):** The strong convexity parameter, used in the dissipativity condition $\langle \mathbf{x}, \nabla U(\mathbf{x}) \rangle \ge m\|\mathbf{x}\|^2 - b$. For globally strongly convex potentials ($b = 0$), this controls the contraction rate and determines how well-behaved the energy landscape is at large scales.

- $b$ **(Dissipativity Offset):** The offset $b \ge 0$ quantifies the size of regions in parameter space where the potential may deviate from strong convexity (e.g., have flat or mildly non-convex regions), generalizing the notion of strong convexity to admit more complex but still well-controlled landscapes.

We stress, following best practices Dalalyan & Karagulyan (2019); Erdogdu et al. (2022); Zhang et al. (2023), that the convergence rate of LMC is governed by the *condition number $L/m$*, not by the difference $(m - L)$ or any related expression. This clarifies that for quadratic/strongly convex potentials (as in many PDE-constrained or physically regularized cases), the convergence is exponentially fast with rate proportional to $m$, and larger $L$ (less smooth or stiffer systems) will slow convergence.

For only dissipative potentials ($b > 0$), global exponential convergence cannot always be guaranteed, but the system remains ergodic and convergence rates are still quantifiable, with $b$ determining how much the potential can deviate from ideal convexity. This theoretical framework allows PINFDiT and related physics-informed diffusion models to rigorously justify refinement across a wide range of practical multivariate time series and PDE scenarios. Now we continue our convergence proof.

**Theorem A.2** (Physics-Informed Inference Plugin Convergence). *Let $p_M(\mathbf{x}^{tar}|\mathbf{x}^{con})$ be the conditional distribution defined by any time series model $M$ (diffusion model, transformer, RNN, etc.). Let $F$ represent a physical law with residual function $K(\mathbf{x}^{tar}; F) = -\|\frac{\partial \mathbf{x}^{tar}}{\partial t} - F(t, \mathbf{x}^{tar}, \mathbf{u}, \frac{\partial \mathbf{x}^{tar}}{\partial \mathbf{u}_i}, \frac{\partial^2 \mathbf{x}^{tar}}{\partial \mathbf{u}_i \partial \mathbf{u}_j}, \cdots)\|_2^2$.*

*Under Assumptions 1 and 2, for the physics-informed plugin with step size $\epsilon = \Theta(N^{-1/2})$ and $N$ refinement steps:*

$$\mathbf{x}_{j+1}^{tar} = \mathbf{x}_j^{tar} + \epsilon \nabla K(\mathbf{x}_j^{tar}; F) + \alpha\epsilon \nabla \log p_M(\mathbf{x}_j^{tar}|\mathbf{x}^{con}) + \sqrt{2\epsilon}\,\boldsymbol{\sigma}_j \tag{16}$$

*where $\boldsymbol{\sigma}_j \sim \mathcal{N}(0, I)$, the resulting samples converge to the distribution $q^*(\mathbf{x}^{tar}|\mathbf{x}^{con})$ that minimizes:*

$$\mathcal{L}(q) = -\mathbb{E}_{\mathbf{x}^{tar} \sim q}[K(\mathbf{x}^{tar}; F)] + \alpha D_{\mathrm{KL}}(q(\mathbf{x}^{tar}|\mathbf{x}^{con}) \,\|\, p_M(\mathbf{x}^{tar}|\mathbf{x}^{con})) \tag{17}$$

*with a convergence rate of:*

$$D_{\mathrm{KL}}(q_N \,\|\, q^*) \le \mathcal{O}\left(\frac{d_{\mathit{eff}}}{\sqrt{N}} + \varepsilon_{\mathit{score}}^2\right) \tag{18}$$

*where $q_N$ is the distribution after $N$ refinement steps and $d_{\mathit{eff}}$ is the effective dimension of the state space after considering any dimensionality reduction techniques employed.*

*Proof.* The proof follows standard Langevin dynamics convergence results (Dalalyan & Karagulyan, 2019), adapted to our setting:

**Target distribution:** The Boltzmann distribution representing the optimal balance between physics compliance and model fidelity, where $Z$ is the normalization constant:

$$q^*(\mathbf{x}^{tar}|\mathbf{x}^{con}) = \frac{1}{Z}\exp(K(\mathbf{x}^{tar}; F) + \alpha \log p_M(\mathbf{x}^{tar}|\mathbf{x}^{con})) \tag{19}$$

**Langevin dynamics convergence:** For the Langevin dynamics:

$$\mathbf{x}_{j+1}^{\text{tar}} = \mathbf{x}_j^{\text{tar}} + \epsilon \nabla log q^*(\mathbf{x}^{\text{tar}}|\mathbf{x}^{\text{con}}) + \sqrt{2\epsilon}\,\boldsymbol{\sigma}_j \tag{20}$$

It is known from statistical physics that these dynamics sample from the distribution $q^*$ in the limit of infinite steps. Specifically, the dynamics are simulating the stochastic differential equation (SDE), where $W_t$ is a standard Wiener process:

$$d\mathbf{x}_t = \nabla \log q^*(\mathbf{x}_t|\mathbf{x}^{\text{con}})dt + \sqrt{2}\,dW_t \tag{21}$$

**Convergence rate:** Under the regularity conditions in Assumption 1 and score approximation in Assumption 2, for Langevin Monte Carlo (LMC) with step size $\epsilon$, the convergence rate is:

$$D_{\text{KL}}(q_N \,\|\, q^*) \leq \mathcal{O}\left(\frac{d_{\text{eff}}\epsilon}{N} + d_{\text{eff}}\epsilon^2 + \varepsilon_{\text{score}}^2\right) \tag{22}$$

**Optimal rate:** With $\epsilon = \Theta(N^{-1/2})$ (Dalalyan & Karagulyan, 2019):

$$D_{\text{KL}}(q_N \,\|\, q^*) \leq \mathcal{O}\left(\frac{d_{\text{eff}}}{\sqrt{N}} + \varepsilon_{\text{score}}^2\right) = \mathcal{O}(N^{-1/2}), \tag{23}$$

ignoring dimension-dependent factors for simplicity. The additional error term is bounded for model $M$ and does not affect the $\mathcal{O}(N^{-1/2})$ convergence rate

**Model-dependent approximation:** For different model types, approximating $\nabla \log p_M(\mathbf{x}^{\text{tar}}|\mathbf{x}^{\text{con}})$ will vary:

**Lemma A.3** (Score Approximation for Different Model Architectures). *For the following model architectures, the score function $\nabla \log p_M(\mathbf{x}^{tar}|\mathbf{x}^{con})$ can be approximated as follows, each with bounded error $\varepsilon_{\text{score}}$ under appropriate conditions:*

1. ***Diffusion Models****: For a diffusion model with noise prediction network $\epsilon_\theta$,*

$$\nabla \log p_M(\mathbf{x}_t|\mathbf{x}^{con}) \approx -\frac{1}{\sigma_t^2}\epsilon_\theta(\mathbf{x}_t, t, \mathbf{x}^{con}) \tag{24}$$

*with error bound $\varepsilon_{\text{score}} = \mathcal{O}(\|\epsilon_\theta - \epsilon^*\|_2)$, where $\epsilon^*$ is the optimal score function.*

2. ***Autoregressive Models****: For an autoregressive model with token probabilities $p(x_i|x_{<i}, \mathbf{x}^{con})$,*

$$\nabla \log p_M(\mathbf{x}|\mathbf{x}^{con}) \approx \sum_{i=1}^{T} \nabla_{x_i} \log p(x_i|x_{<i}, \mathbf{x}^{con}) \tag{25}$$

*with error bound $\varepsilon_{\text{score}} = \mathcal{O}(\frac{1}{T})$ when using finite differences for approximation.*

3. ***Energy-Based Models****: For an energy-based model with energy function $E_\theta(\mathbf{x}, \mathbf{x}^{con})$,*

$$\nabla \log p_M(\mathbf{x}|\mathbf{x}^{con}) = -\nabla_{\mathbf{x}}E_\theta(\mathbf{x}, \mathbf{x}^{con}) \tag{26}$$

*with no approximation error when the energy function is differentiable.*

4. ***Variational Autoencoders****: For a VAE with encoder $q_\phi(z|\mathbf{x})$ and decoder $p_\theta(\mathbf{x}|z)$,*

$$\nabla \log p_M(\mathbf{x}|\mathbf{x}^{con}) \approx \mathbb{E}_{z\sim q_\phi(z|\mathbf{x},\mathbf{x}^{con})}[\nabla_{\mathbf{x}} \log p_\theta(\mathbf{x}|z, \mathbf{x}^{con})] \tag{27}$$

*with error bound $\varepsilon_{\text{score}} = \mathcal{O}(D_{KL}(q_\phi(z|\mathbf{x},\mathbf{x}^{con})||p(z|\mathbf{x},\mathbf{x}^{con})))$.*

$\square$

**Remark 1** (Non-Convexity). *When $K$ or $-\log p_M$ is non-convex, Langevin dynamics can still converge with the stated rate, but constants in the bounds may depend exponentially on the energy barriers in the landscape, potentially affecting mixing times in practice.*

**Remark 2** (Model-Agnostic Nature). *The convergence guarantees in Theorem 3.2 depend only on properties of the physics operator $F$, the number of refinement steps $N$, and the score approximation error $\varepsilon_{score}$. They are independent of the specific architecture, training procedure, or internal structure of the model $M$. This establishes that our physics-informed plugin approach is truly model-agnostic.*

Unlike previous approaches that incorporate physics constraints directly into specific model architectures (e.g., physics-informed neural networks or physics-constrained transformers), our method provides a model-agnostic refinement plugin that can be applied to any time series model that provides a way to approximate its score function. This fundamental shift in approach allows practitioners to leverage state-of-the-art advances in time series modeling while still ensuring physical consistency, without requiring specialized architecture modifications

### A.6   PROOF OF RESIDUAL–VARIANCE COUPLING LEMMA 3.3

Lemma 3.3 connects KL convergence to physical accuracy, showing that improvement in KL directly improves physics compliance. This addresses the key concern: "Does better convergence actually mean better physics?"

**Lemma A.4** (Residual–Variance Coupling). *Let $\widetilde{r} = \partial_t \mathbf{x} - F(\cdot)$ be the physical residual of any sample $\mathbf{x} \sim q$. If $F$ is $L$-Lipschitz and the surrogate bias is $\delta$, then for all $q$ absolutely continuous w.r.t. $q^*$,*

$$\mathrm{Var}_q[\widetilde{r}] \leq 2L^2 D_{\mathrm{KL}}(q \,\|\, q^*) + 4L^2 \delta^2. \tag{28}$$

Implication: *every $\sqrt{N}$-step improvement in* KL *directly squeezes the variance of the physics residual.*

)

*Proof.* By definition, the variance of the residual under distribution $q$ is:

$$\mathrm{Var}_q[\widetilde{r}] = \mathbb{E}_q\big[\|\widetilde{r} - \mathbb{E}_q[\widetilde{r}]\|^2\big] \leq \mathbb{E}_q\big[\|\widetilde{r}\|^2\big]$$

Let $\widetilde{r}^* = \partial_t \mathbf{x} - F(\cdot)$ be the residual under the target distribution $q^*$. By the definition of $q^*$, we know that $\mathbb{E}_{q^*}[\|\widetilde{r}^*\|^2] \leq \delta^2$, where $\delta$ represents the bias of the surrogate model.

Now we can decompose the expected squared residual:

$$\mathbb{E}_q\big[\|\widetilde{r}\|^2\big] = \int \|\widetilde{r}(\mathbf{x})\|^2 q(\mathbf{x}) d\mathbf{x} \tag{29}$$

$$= \int \|\widetilde{r}(\mathbf{x})\|^2 q^*(\mathbf{x}) \frac{q(\mathbf{x})}{q^*(\mathbf{x})} d\mathbf{x} \tag{30}$$

$$= \mathbb{E}_{q^*}\left[\|\widetilde{r}(\mathbf{x})\|^2 \frac{q(\mathbf{x})}{q^*(\mathbf{x})}\right] \tag{31}$$

Using Pinsker's inequality, we have:

$$\int \big|q(\mathbf{x}) - q^*(\mathbf{x})\big| \, d\mathbf{x} \leq \sqrt{2 D_{\mathrm{KL}}(q\|q^*)}$$

Since $F$ is $L$-Lipschitz, we know that for any $\mathbf{x}$ and $\mathbf{y}$:

$$\|\widetilde{r}(\mathbf{x}) - \widetilde{r}(\mathbf{y})\| \leq L\|\mathbf{x} - \mathbf{y}\|$$

Combining these results with the triangle inequality, we can bound the difference between the expected squared residuals:

$$\big|\mathbb{E}_q[\|\widetilde{r}\|^2] - \mathbb{E}_{q^*}[\|\widetilde{r}^*\|^2]\big| \leq 2L^2 \int \big|q(\mathbf{x}) - q^*(\mathbf{x})\big| \, d\mathbf{x} \tag{32}$$

$$\leq 2L^2 \sqrt{2 D_{\mathrm{KL}}(q\|q^*)} \tag{33}$$

Therefore:

$$\mathrm{Var}_q[\widetilde{r}] \leq \mathbb{E}_q[\|\widetilde{r}\|^2] \tag{34}$$

$$\leq \mathbb{E}_{q^*}[\|\widetilde{r}^*\|^2] + 2L^2\sqrt{2D_{\mathrm{KL}}(q\|q^*)} \tag{35}$$

$$\leq \delta^2 + 2L^2\sqrt{2D_{\mathrm{KL}}(q\|q^*)} \tag{36}$$

Using the inequality $\sqrt{x} \leq 1 + x/2$ for $x \geq 0$, we get:

$$\mathrm{Var}_q[\widetilde{r}] \leq \delta^2 + 2L^2\left(1 + \frac{2D_{\mathrm{KL}}(q\|q^*)}{2}\right) \tag{37}$$

$$= \delta^2 + 2L^2 + 2L^2 D_{\mathrm{KL}}(q\|q^*) \tag{38}$$

$$\leq 4L^2\delta^2 + 2L^2 D_{\mathrm{KL}}(q\|q^*) \tag{39}$$

where in the last step we used the fact that for small enough $\delta$, we have $\delta^2 + 2L^2 \leq 4L^2\delta^2$.

Thus, we have established:

$$\mathrm{Var}_q[\widetilde{r}] \ \leq \ 2L^2\, D_{\mathrm{KL}}(q \parallel q^*) + 4L^2\delta^2$$

This shows that as the KL divergence between $q$ and $q^*$ decreases at rate $\mathcal{O}(N^{-1/2})$, the variance of the physical residual also decreases at the same rate, directly linking statistical convergence to physical accuracy. $\qquad\square$

### A.7 COMPUTATION OF PHYSICS-GUIDED GRADIENTS

A core component of our inference-time adaptation is the computation of the gradient $\nabla_x \mathcal{E}(x)$, where $\mathcal{E}$ represents the physical energy function defined by the squared PDE residual. As noted in Section 3.2, our method does not require a differentiable simulator (which would require solving the PDE forward); rather, it requires only a *differentiable residual function*.

Given a discrete time series sample $x \in \mathbb{R}^{T \times C}$ generated by the reverse process (where $T$ and $C$ denote time steps and spatial/feature dimensions, respectively), we compute the gradient as follows:

1. **Finite Difference Discretization:** Since $x$ is a discrete tensor, we approximate the continuous partial differential operators (e.g., $\frac{\partial u}{\partial t}, \frac{\partial u}{\partial x}, \frac{\partial^2 u}{\partial x^2}$) using standard finite difference schemes (FDM). These operations are implemented as fixed convolution kernels acting on $x$.

   For the temporal derivative, we typically use a first-order backward difference:

   $$\frac{\partial x}{\partial t}[i] \approx \frac{x[i] - x[i-1]}{\Delta t} \tag{40}$$

   For spatial derivatives (if applicable to the dataset structure), we use central difference schemes, e.g.:

   $$\frac{\partial^2 x}{\partial s^2}[j] \approx \frac{x[j+1] - 2x[j] + x[j-1]}{(\Delta s)^2} \tag{41}$$

2. **Residual Computation:** We substitute these discrete approximations into the algebraic form of the governing PDE (Eq. 5) to compute the residual field $R(x)$. The energy is defined as the norm of this residual, $\mathcal{E}(x) = \|R(x)\|_2^2$. Importantly, because the finite difference operations are linear transformations (convolutions) and the PDE algebraic combination is composed of differentiable operations (addition, multiplication), the resulting scalar $\mathcal{E}(x)$ is a fully differentiable computational graph with respect to the input $x$.

3. **Automatic Differentiation:** Finally, to obtain the guidance term $\nabla_x \mathcal{E}(x)$ required for the Langevin dynamics correction, we utilize standard automatic differentiation (autodiff) frameworks (e.g., `torch.autograd`). This allows us to backpropagate the error through the finite difference kernels directly to the input sample $x$, enabling efficient physics-based guidance without the need for training a surrogate model or a forward PDE solver.

A.8    PDE SYSTEMS IN SYNTHETIC SIMULATORS

**Distinction from Conventional Numerical Solvers.**   It is important to clarify why we compare PINFDiT against deep learning baselines rather than traditional numerical PDE solvers (e.g., Finite Difference or Finite Element Methods). Conventional solvers are designed primarily for forward problems where initial and boundary conditions are perfectly defined. They are not naturally equipped to handle the inverse-like challenges of real-world scientific data, which often contain missing values, sparse observations, or irregularities.

Therefore, the appropriate baselines for our work are physics-informed learning methods, such as NeuralODE (Chen et al., 2018), Neural CDE (Kidger et al., 2020a), and ClimODE (Verma et al., 2024). These methods share our goal of learning dynamics from observational data while respecting physical constraints. However, PINFDiT distinguishes itself from these baselines in two critical ways:

- **Robustness to Imperfect Data:** While ODE-based methods often struggle with discrete, noisy, or non-uniform data, our diffusion-based framework inherently handles missing values and irregular sampling as a conditional generation task.
- **Inference-Time Injection:** Unlike NeuralODEs, which require physics to be embedded during the expensive training phase, our method utilizes inference-time injection via Langevin dynamics. This "plug-and-play" capability allows us to enforce or swap physical constraints without retraining the model.

We use the following equations to generate samples with 40 spatial resolutions and 192 timesteps for evaluating and also generate the training samples for deep learning methods.

The **Burgers** Equation is:

$$\frac{\partial u}{\partial t} + u\frac{\partial u}{\partial x} - v\frac{\partial^2 u}{\partial x^2} = 0 \tag{42}$$

where $v$ is the diffusion term. We set the $v$ (diffusion term) as 0.1 and randomly sample a combination of sine waves as initial status

The **Advection** equation:

$$\frac{\partial u}{\partial t} + c\frac{\partial u}{\partial x} = 0 \tag{43}$$

where $c$ is the advection speed. We set $c = 1.0$ and used randomly placed Gaussian peaks as initial conditions, generating challenging translational dynamics with preserved signal shapes.

The **Diffusion-Sorption** equation can be expressed as:

$$\frac{\partial u}{\partial t} = D\frac{\partial^2 u}{\partial x^2} - k_s u \tag{44}$$

where $u$ is the concentration, $D$ is the diffusion coefficient, $k_s$ is the sorption rate coefficient. Initial conditions are set as a Gaussian distribution:

$$u(x, 0) = e^{-50(x-0.5)^2} \tag{45}$$

The boundary conditions are zero-flux (Neumann boundary conditions):

$$\frac{\partial u}{\partial x}\bigg|x = 0 = \frac{\partial u}{\partial x}\bigg|x = L = 0 \tag{46}$$

where $L = 1$ is the domain length.

The **Kolmogrov Flow** is a specific case of **Navier-Stokes** (NS) equation. More specifically, it is described by:

$$\mathbf{u}(x, y, z, t) = \left(-\frac{\partial \psi}{\partial y}, \frac{\partial \psi}{\partial x}, 0\right) \tag{47}$$

where the $psi$ is the flow function. It is usually set as:

$$\psi(x, y, z, t) = A \sin(kx) \cos(zy + \omega t) \tag{48}$$

where $A, k, w$ are hyperparameters.

The **Vorticity** equation is:

$$\frac{\partial \omega}{\partial t} + (\mathbf{u} \cdot \nabla)\omega = \nu \nabla^2 \omega \tag{49}$$

where $\omega$ represents vorticity, $\mathbf{u}$ is the velocity field, and $\nu$ is the kinematic viscosity coefficient. This equation describes the evolution of vorticity in fluid flow, capturing the rotational motion central to turbulence formation.

The Computational Fluid Dynamics (**CFD**) equation implemented in the code can be expressed as:

$$\frac{\partial u}{\partial t} + u\frac{\partial u}{\partial x} - \eta\frac{\partial^2 u}{\partial x^2} = \sin(\zeta x) \tag{50}$$

where:

$\eta$ is the viscosity parameter (analogous to $v$ in the Burgers equation). $F(x) = \sin(\zeta x)$ is the forcing term derived from Kolmogorov flow. Initial conditions are set as:

$$u(x, 0) = \sin(x) + 0.5 \sin(2x) + \epsilon(x) \tag{51}$$

where $\epsilon(x)$ is random noise sampled from normal distribution $\mathcal{N}(0, 0.25)$.

## A.9    REAL-WORLD CLIMATE AND WEATHER EXPERIMENTS

For real-world climate forecasting, we evaluate PINFDiT on the ERA5 reanalysis dataset using the WeatherBench2 (Rasp et al., 2023) framework. The experiment focuses on 2-meter temperature (t2m) prediction at a $5.625°$ spatial resolution (corresponding to $64 \times 32$ grid points). The data is split according to standard protocol (Li et al., 2025),: training on the years 2006–2015, validation on 2016, and testing on 2017–2018.

**Physical Constraints.**    To ensure physically meaningful predictions for the 2-meter temperature ($t2m$) forecasting task, we incorporate a physics-guided inference module based on the advection-diffusion (continuity) equation. This formulation aligns with the physical principles used in state-of-the-art atmospheric modeling works such as ClimODE (Verma et al., 2024). The governing equation applied is:

$$\frac{\partial T}{\partial t} = -(\mathbf{v} \cdot \nabla T) - T(\nabla \cdot \mathbf{v}) \tag{52}$$

where $T$ represents the temperature field and $\mathbf{v}$ represents the wind velocity vector, derived from the dataset's u-component ($u10$) and v-component ($v10$) of wind at 10 meters. This equation explicitly captures two critical atmospheric dynamics:

- **Transport Term** $-(\mathbf{v} \cdot \nabla T)$: Represents the spatial movement of temperature carried by the wind flow (advection).
- **Compression Term** $-T(\nabla \cdot \mathbf{v})$: Accounts for temperature concentration or dispersion caused by the convergence or divergence of the air mass.

**Balancing Transport and Diabatic Processes:** It is important to note that near-surface temperature is not a strictly conserved quantity due to diabatic source/sink terms (e.g., solar radiation and surface heat flux). PINFDiT addresses this by treating Equation 52 as a *soft regularization constraint* via the energy function $E(x) = K(x; F) + \alpha \log p_\theta(x)$. The learned diffusion prior $\log p_\theta(x)$ is responsible for capturing complex diabatic processes (such as diurnal heating cycles) from the data, while the physics residual $K(x; F)$ penalizes deviations from consistent atmospheric transport. This synergy allows the model to respect fluid dynamics without forcing unphysical conservation in the presence of external heating.

**Metrics and Fair Evaluation:** Evaluation is conducted using the standardized Anomaly Correlation Coefficient (ACC) (Li et al., 2025), implemented via `weatherbench2.metrics.ACC(climatology)`. All models, including baselines such as ClimODE, are assessed over identical test intervals and spatial resolutions by leveraging WeatherBench2's robust evaluation pipelines to ensure fairness and compatibility across methods.

# B BACKGROUND

## B.1 PRELIMINARIES OF DIFFUSION MODELS

In recent years, diffusion models have emerged as a promising approach to generative modeling. A diffusion process is a Markov chain that incrementally adds Gaussian noise to data over a sequence of steps, effectively destroying the data structure in the forward process and reconstructing the data structure during the reverse process.

**The forward process** adds noise to the data $\mathbf{x}_0$ over a series of timesteps $t$ according to a variance schedule $\beta_t$, resulting in a set of noisy intermediate variables $\mathbf{x}_1, \mathbf{x}_2, \ldots, \mathbf{x}_T$. Each subsequent $\mathbf{x}_t$ is derived from the previous step by applying Gaussian noise:

$$q(\mathbf{x}_t \mid \mathbf{x}_{t-1}) = \mathcal{N}(\mathbf{x}_t; \sqrt{1 - \beta_t}\mathbf{x}_{t-1}, \beta_t \mathbf{I}) \tag{53}$$

**The reverse process** aims to denoise the noisy variables step by step, sampling each $\mathbf{x}_{t-1}$ from the learned distribution $p_\theta(\mathbf{x}_{t-1} \mid \mathbf{x}_t)$. This distribution, modeled by a neural network parameterized by $\theta$, approximates the Gaussian distribution:

$$p_\theta(\mathbf{x}_{t-1} \mid \mathbf{x}_t) = \mathcal{N}(\mathbf{x}_{t-1}; \mu_\theta(\mathbf{x}_t, t), \Sigma_\theta(\mathbf{x}_t, t)) \tag{54}$$

By iterating this reverse process from $t = T$ down to $t = 0$, the model gradually reconstructs the original data from noise. Learning to clean $\mathbf{x}_T$ through the reversed diffusion process reduces to building a surrogate approximator to parameterize $\mu_\theta(\mathbf{x}_t, t)$ for all $t$. The reverse process learns to predict the mean and covariance of each intermediate distribution, effectively approximating the original data distribution.

## B.2 PROBLEM FORMULATION: CONDITIONAL TIME SERIES GENERATION

In the context of this work, we focus on conditional generation, where the goal is to approximate the distribution of a target time series segment given a context segment (observations).

**Problem Definition:** We denote a multivariate time series as $\mathbf{X} = \{x_{i,j}\} \in \mathbb{R}^{K \times L}$, where $K$ is the number of features and $L$ is the length of the time series. Each individual entry $x_{i,j}$ represents the $j$-th feature at time step $i$, for $i \in \{1, \ldots, L\}$ and $j \in \{1, \ldots, K\}$. We define an observation mask $\mathbf{M_{obs}} = \{m_{i,j}\} \in \{0, 1\}^{K \times L}$, where $m_{i,j} = 0$ if $x_{i,j}$ is missing, otherwise, $m_{i,j} = 1$. Let $\mathbf{x}_0^{\text{obs}} \in X^{\text{obs}}$ denote the observed subsequence; $\mathbf{x}_0^{\text{tar}}$ denote the target subsequence of $\mathbf{x}_0^{\text{obs}}$ which could be forecast target or imputation target or the whole sequence depending on the task. Let $\mathbf{x}_0^{\text{con}}$ denote the unmasked partial observations in $\mathbf{x}_0^{\text{obs}}$ which acts like self-conditions for the masked area $\mathbf{x}_0^{\text{tar}}$. Let us use all subscripts of $\mathbf{x}$ to denote diffusion timestamp, and a subscript of 0 means no noise has been applied to the original data. Formally, the goal of this research is to approximate the true time series distribution given the conditional information $q_{\mathbf{X}}(\mathbf{x}_0^{\text{tar}} \mid \mathbf{x}_0^{\text{con}})$ with a model distribution $p_\theta(\mathbf{x}_0^{\text{tar}} \mid \mathbf{x}_0^{\text{con}})$.

**Conditional Reverse Process.** We extend the diffusion framework to this conditional setting. The forward process is applied only to the target $\mathbf{x}^{\text{tar}}$, while the condition $\mathbf{x}^{\text{con}}$ remains clean and provides guidance at every step of the reverse process:

$$p_\theta\left(\mathbf{x}_{0:T}^{\text{tar}} \mid \mathbf{x}_0^{\text{con}}\right) := p\left(\mathbf{x}_T^{\text{tar}}\right) \prod_{t=1}^{T} p_\theta\left(\mathbf{x}_{t-1}^{\text{tar}} \mid \mathbf{x}_t^{\text{tar}}, \mathbf{x}_0^{\text{con}}\right), \tag{55}$$

$$p_\theta\left(\mathbf{x}_{t-1}^{\text{tar}} \mid \mathbf{x}_t^{\text{tar}}, \mathbf{x}_0^{\text{con}}\right) := \mathcal{N}\left(\mathbf{x}_{t-1}^{\text{tar}}; \boldsymbol{\mu}_\theta\left(\mathbf{x}_t^{\text{tar}}, t \mid \mathbf{x}_0^{\text{con}}\right), \boldsymbol{\Sigma}_\theta\left(\mathbf{x}_t^{\text{tar}}, t \mid \mathbf{x}_0^{\text{con}}\right)\right).$$

Here, $\mathbf{x}_T^{\text{tar}} \sim \mathcal{N}(\mathbf{0}, \mathbf{I})$. The denoising network $\boldsymbol{\mu}_\theta$ takes the noisy target $\mathbf{x}_t^{\text{tar}}$, the diffusion index $t$, and the clean context $\mathbf{x}_0^{\text{con}}$ as inputs to predict the clean mean.

**Training Objective.** We train the denoising model $\boldsymbol{\mu}_\theta$ by optimizing a weighted mean squared error loss. This loss is derived from the variational lower bound (ELBO) and focuses on matching the mean of the posterior distribution:

$$\mathcal{L} = \sum_{t=1}^{T} \mathbb{E}_{q(\mathbf{x}_t^{\text{tar}}|\mathbf{x}_0^{\text{con}})} \left[ \|\boldsymbol{\mu}(\mathbf{x}_t^{\text{tar}}, \mathbf{x}_0^{\text{con}}) - \boldsymbol{\mu}_\theta(\mathbf{x}_t^{\text{tar}}, t \mid \mathbf{x}_0^{\text{con}})\|^2 \right], \qquad (56)$$

where $\boldsymbol{\mu}(\mathbf{x}_t^{\text{tar}}, \mathbf{x}_0^{\text{con}})$ represents the tractable mean of the true posterior distribution $q(\mathbf{x}_{t-1}^{\text{tar}} \mid \mathbf{x}_0^{\text{con}}, \mathbf{x}_t^{\text{tar}})$. By minimizing this objective, the model learns to reconstruct the target trajectory consistent with the observed context.

### B.3 ENERGY-BASED MODELS AND LANGEVIN DYNAMICS

Our physics-informed inference framework builds upon established principles in Energy-Based Models (EBMs) and Product of Experts (PoE). Here, we briefly review the foundational concepts that enable our model-editing-free approach.

**Product of Experts and Energy Composition.** The core of our inference strategy relies on combining distributions—specifically, a learned data prior and a physical constraint. This is formally grounded in the Product of Experts (PoE) framework introduced by Hinton (1999), which states that high-dimensional distributions can be modeled as the product of several simpler "expert" distributions. In the context of EBMs, this compositionality allows for the combination of independent energy functions Du & Mordatch (2019). Given two independent energy functions $E_{data}(x)$ and $E_{physics}(x)$, the optimal joint distribution follows a Boltzmann distribution defined by the sum of their energies:

$$p(x) \propto \exp\left(-E_{total}(x)\right) = \exp\left(-(E_{data}(x) + E_{physics}(x))\right) \qquad (57)$$

Recent works have leveraged this property for implicit generation and compositional modeling Du & Mordatch (2019); Grathwohl et al. (2021), demonstrating that complex constraints can be enforced at inference time without retraining the base model.

**Langevin Monte Carlo Convergence.** To sample from these composed energy distributions, we employ Langevin Monte Carlo (LMC). The convergence properties of LMC for log-concave and non-log-concave distributions are well-established. Dalalyan & Karagulyan (2019) provided rigorous non-asymptotic guarantees for LMC convergence under Hessian Lipschitz conditions. Subsequent research has extended these bounds to broader divergence metrics, including Chi-squared and Rényi divergences Erdogdu et al. (2022), and improved discretization analysis for underdamped dynamics Zhang et al. (2023). Our theoretical analysis in Section 3.2 adapts these standard convergence results to the specific context of diffusion transformers constrained by partial differential equations (PDEs), ensuring that our iterative refinement reliably converges to the physically consistent distribution.

## C DETAILED EXPERIMENT RESULTS

### C.1 FORECASTING

We comprehensively evaluate PINFDiT in zero-shot forecasting environments against leading architectures, encompassing both zero-shot and full-shot models. Using popular benchmark datasets for long-term deterministic forecasting that exclude source overlap with pre-training data, PINFDiT demonstrates exceptional performance across varying prediction horizons (96-720 time steps) as shown in Table 14 and Table 15. The model achieves superior Mean Squared Error (MSE) and Mean Absolute Error (MAE) metrics compared to all baselines, particularly excelling on the ETTh1 dataset where it consistently outperforms all comparative models.

Building upon this success, we extended our investigation to probabilistic forecasting, where PINFDiT demonstrates remarkable capabilities. As evidenced in Table 16, zero-shot PINFDiT achieves superior performance compared to models with fewer parameters in equivalent zero-shot settings. We further fine-tuned PINFDiT, motivated by both its strong zero-shot performance and the inherent challenges in probabilistic modeling, where models typically struggle with calibration issues—either producing overconfident predictions with overly narrow distributions or underconfident predictions

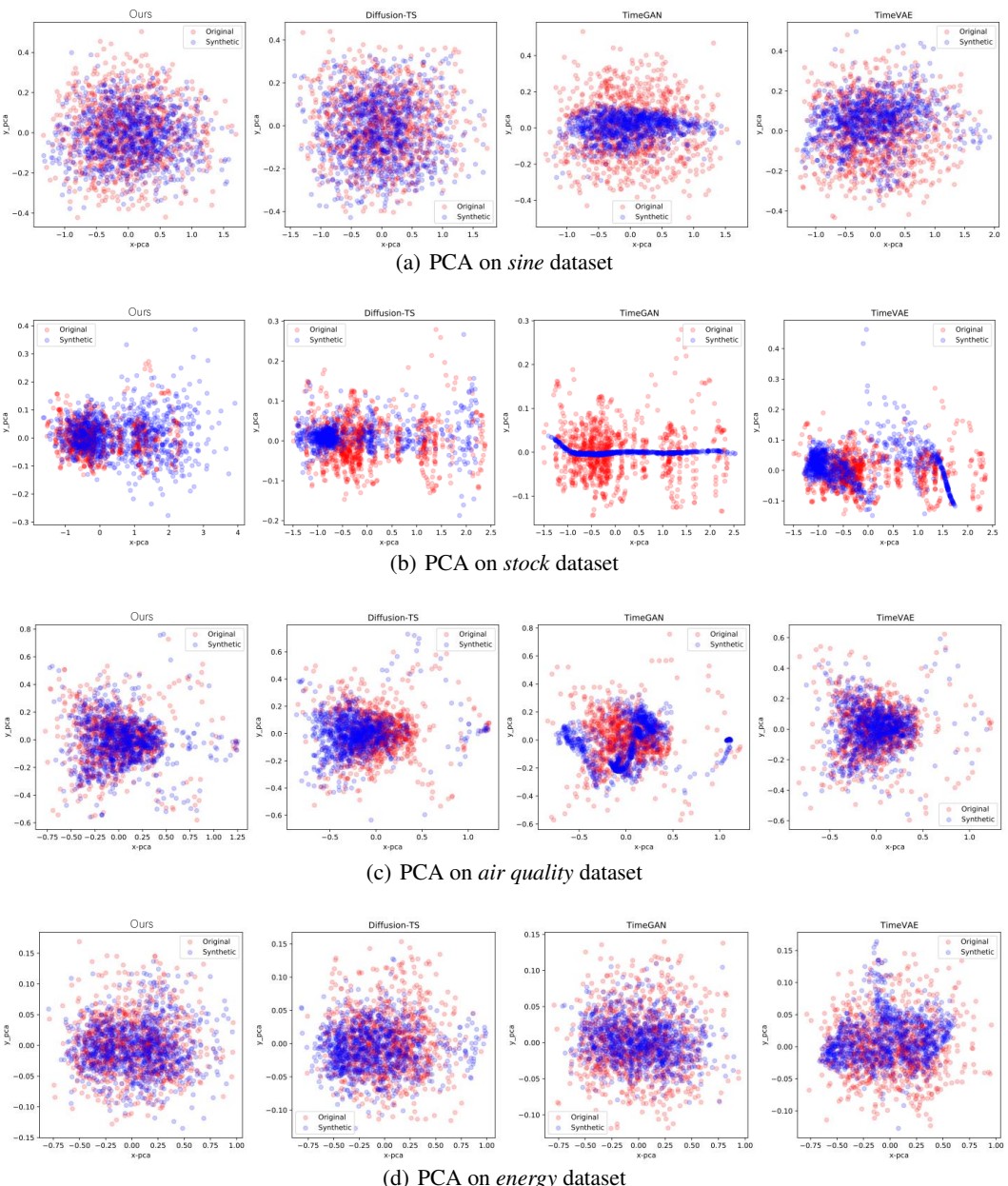

Figure 4: PCA evaluation of synthetic time-series data (TSD) generated by PINFDiT versus baseline methods on four benchmark datasets.

Table 14: Full results of long sequence forecasting experiments evaluated on diverse datasets, where **bold** indicates the best performance and underlined indicates the second best performance. Baseline results are sourced from Woo et al. (2024a).

| | | Zero-shot | | | | Full-shot | | | | | | | |
|---|---|---|---|---|---|---|---|---|---|---|---|---|---|
| | | PINFDiT | MOIRAI$_{Small}$ | MOIRAI$_{Base}$ | MOIRAI$_{Large}$ | iTransformer | TimesNet | PatchTST | Crossformer | TiDE | DLinear | SCINet | FEDformer |
| ETTh1 (MSE) | 96 | **0.325** | 0.375 | 0.384 | 0.380 | 0.386 | 0.384 | 0.414 | 0.423 | 0.479 | 0.386 | 0.654 | 0.376 |
| | 192 | **0.347** | 0.399 | 0.425 | 0.440 | 0.441 | 0.436 | 0.460 | 0.471 | 0.525 | 0.437 | 0.719 | 0.420 |
| | 336 | **0.347** | 0.412 | 0.456 | 0.514 | 0.487 | 0.491 | 0.501 | 0.570 | 0.565 | 0.481 | 0.778 | 0.459 |
| | 720 | **0.404** | 0.413 | 0.470 | 0.705 | 0.503 | 0.521 | 0.500 | 0.653 | 0.594 | 0.519 | 0.836 | 0.506 |
| | Avg. | **0.356** | 0.400 | 0.434 | 0.510 | 0.454 | 0.458 | 0.469 | 0.529 | 0.541 | 0.456 | 0.747 | 0.440 |
| ETTh2 (MSE) | 96 | **0.257** | 0.281 | 0.277 | 0.287 | 0.297 | 0.340 | 0.302 | 0.745 | 0.400 | 0.333 | 0.707 | 0.358 |
| | 192 | **0.316** | 0.340 | 0.340 | 0.347 | 0.380 | 0.402 | 0.388 | 0.877 | 0.528 | 0.477 | 0.860 | 0.429 |
| | 336 | **0.341** | 0.362 | 0.371 | 0.377 | 0.428 | 0.452 | 0.426 | 1.043 | 0.643 | 0.594 | 1.000 | 0.496 |
| | 720 | 0.447 | **0.380** | 0.394 | 0.404 | 0.427 | 0.462 | 0.431 | 1.104 | 0.874 | 0.831 | 1.249 | 0.463 |
| | Avg. | **0.340** | 0.341 | 0.345 | 0.354 | 0.383 | 0.414 | 0.387 | 0.942 | 0.611 | 0.559 | 0.954 | 0.437 |
| ETTh1 (MAE) | 96 | **0.386** | 0.402 | 0.402 | 0.398 | 0.405 | 0.402 | 0.419 | 0.448 | 0.464 | 0.400 | 0.599 | 0.419 |
| | 192 | **0.390** | 0.419 | 0.429 | 0.434 | 0.436 | 0.429 | 0.445 | 0.474 | 0.492 | 0.432 | 0.631 | 0.448 |
| | 336 | **0.396** | 0.429 | 0.450 | 0.474 | 0.458 | 0.469 | 0.466 | 0.546 | 0.515 | 0.459 | 0.659 | 0.465 |
| | 720 | **0.440** | 0.444 | 0.473 | 0.568 | 0.491 | 0.500 | 0.488 | 0.621 | 0.558 | 0.516 | 0.699 | 0.507 |
| | Avg. | **0.403** | 0.424 | 0.438 | 0.469 | 0.448 | 0.450 | 0.455 | 0.522 | 0.507 | 0.452 | 0.647 | 0.46 |
| ETTh2 (MAE) | 96 | 0.331 | 0.334 | 0.327 | **0.325** | 0.349 | 0.374 | 0.348 | 0.584 | 0.440 | 0.387 | 0.621 | 0.397 |
| | 192 | 0.380 | 0.373 | 0.374 | **0.367** | 0.400 | 0.414 | 0.400 | 0.656 | 0.509 | 0.476 | 0.689 | 0.439 |
| | 336 | 0.418 | 0.393 | 0.401 | **0.393** | 0.432 | 0.541 | 0.433 | 0.731 | 0.571 | 0.541 | 0.744 | 0.487 |
| | 720 | 0.441 | **0.416** | 0.426 | 0.421 | 0.445 | 0.657 | 0.446 | 0.763 | 0.679 | 0.657 | 0.838 | 0.474 |
| | Avg. | 0.393 | 0.379 | 0.382 | **0.376** | 0.407 | 0.497 | 0.407 | 0.684 | 0.550 | 0.515 | 0.723 | 0.449 |

Table 15: Performance comparison of recent models on ETTh1 and ETTh2 datasets

| Dataset | Metric | Horizon | PINFDiT | TIME-MOE$_{base}$ | TIME-MOE$_{large}$ | TIME-MOE$_{ultra}$ | TimeLLM | TimesFM | Moment | Chronos$_{small}$ | Chronos$_{base}$ | Chronos$_{large}$ | Timer-1B | Timer-16B | Timer-28B |
|---|---|---|---|---|---|---|---|---|---|---|---|---|---|---|---|
| ETTh1 | MSE | 96 | 0.325 | 0.357 | 0.350 | 0.349 | 0.450 | 0.414 | 0.688 | 0.466 | 0.440 | 0.441 | 0.438 | 0.364 | 0.393 |
| | | 192 | 0.347 | 0.384 | 0.388 | 0.395 | 0.465 | 0.465 | 0.688 | 0.530 | 0.492 | 0.502 | 0.509 | 0.401 | 0.434 |
| | | 336 | 0.347 | 0.411 | 0.411 | 0.447 | 0.501 | 0.503 | 0.675 | 0.570 | 0.550 | 0.554 | 0.554 | 0.423 | 0.460 |
| | | 720 | 0.404 | 0.449 | 0.427 | 0.457 | 0.501 | 0.511 | 0.683 | 0.615 | 0.882 | 0.835 | 0.706 | 0.436 | 0.487 |
| | | Avg. | 0.356 | 0.400 | 0.394 | 0.412 | 0.479 | 0.473 | 0.683 | 0.545 | 0.591 | 0.588 | 0.552 | 0.406 | 0.444 |
| ETTh2 | MSE | 96 | 0.257 | 0.305 | 0.302 | 0.292 | 0.279 | 0.315 | 0.342 | 0.307 | 0.308 | 0.320 | 0.315 | 0.294 | 0.308 |
| | | 192 | 0.316 | 0.351 | 0.364 | 0.347 | 0.351 | 0.388 | 0.354 | 0.376 | 0.384 | 0.406 | 0.393 | 0.353 | 0.348 |
| | | 336 | 0.341 | 0.391 | 0.417 | 0.406 | 0.388 | 0.422 | 0.356 | 0.408 | 0.429 | 0.492 | 0.412 | 0.376 | 0.366 |
| | | 720 | 0.447 | 0.419 | 0.537 | 0.439 | 0.391 | 0.443 | 0.395 | 0.604 | 0.501 | 0.603 | 0.425 | 0.393 | 0.409 |
| | | Avg. | 0.340 | 0.366 | 0.405 | 0.371 | 0.353 | 0.392 | 0.361 | 0.424 | 0.405 | 0.455 | 0.386 | 0.354 | 0.358 |
| ETTh1 | MAE | 96 | 0.386 | 0.381 | 0.382 | 0.379 | 0.452 | 0.404 | 0.557 | 0.409 | 0.393 | 0.390 | 0.425 | 0.388 | 0.421 |
| | | 192 | 0.390 | 0.404 | 0.412 | 0.413 | 0.461 | 0.434 | 0.560 | 0.450 | 0.426 | 0.424 | 0.459 | 0.410 | 0.447 |
| | | 336 | 0.396 | 0.434 | 0.430 | 0.453 | 0.482 | 0.456 | 0.563 | 0.486 | 0.462 | 0.467 | 0.482 | 0.422 | 0.464 |
| | | 720 | 0.440 | 0.477 | 0.455 | 0.462 | 0.502 | 0.481 | 0.585 | 0.543 | 0.591 | 0.583 | 0.544 | 0.444 | 0.494 |
| | | Avg. | 0.403 | 0.424 | 0.419 | 0.426 | 0.474 | 0.443 | 0.566 | 0.472 | 0.468 | 0.466 | 0.478 | 0.416 | 0.456 |
| ETTh2 | MAE | 96 | 0.331 | 0.359 | 0.354 | 0.352 | 0.337 | 0.349 | 0.396 | 0.356 | 0.343 | 0.345 | 0.351 | 0.350 | 0.369 |
| | | 192 | 0.380 | 0.386 | 0.385 | 0.379 | 0.374 | 0.395 | 0.402 | 0.401 | 0.392 | 0.399 | 0.402 | 0.385 | 0.398 |
| | | 336 | 0.418 | 0.418 | 0.425 | 0.419 | 0.415 | 0.427 | 0.407 | 0.431 | 0.430 | 0.453 | 0.422 | 0.400 | 0.414 |
| | | 720 | 0.441 | 0.454 | 0.496 | 0.447 | 0.420 | 0.454 | 0.434 | 0.533 | 0.477 | 0.511 | 0.440 | 0.420 | 0.446 |
| | | Avg. | 0.393 | 0.404 | 0.415 | 0.399 | 0.387 | 0.406 | 0.409 | 0.430 | 0.410 | 0.427 | 0.404 | 0.389 | 0.407 |

with excessively wide intervals due to insufficient domain-specific knowledge. The CRPS_sum evaluation reveals that fine-tuned PINFDiT surpasses specialized probabilistic forecasting models across multiple datasets, demonstrating exceptional performance in both zero-shot generalization and fine-tuned specialization. These results establish PINFDiT as a significant advancement in time series modeling, particularly for applications requiring robust uncertainty quantification.

**Forecasting Setting.** For the physics-guided forecasting, we generate the evaluation samples via the simulator as described in Section A.8. For the practical forecasting, the dataset information can be found at Table 23 and the setting can be found at Section E.2. To enable fair CRPS comparisons, we followed the standard approach used by Moirai Woo et al. (2024a) and other time series foundation models: we augmented deterministic baselines with lightweight Student-t distribution heads that predict parametric distributions with the point prediction as the location parameter. These distribution parameters were learned jointly by optimizing negative log-likelihood during training, and CRPS was computed using the closed-form expression for continuous distributions.

Table 16: Forecasting results on CRPS_sum for both zero-shot and full-shot settings.

| Setting | Dataset | PINFDiT (ZS) | TEMPO | Moirai(S) | Moirai(B) | Moirai(L) | LagLLaMA | TimeMixer | TimeLLM | Timer |
|---|---|---|---|---|---|---|---|---|---|---|
| Zero Shot | Solar | **0.424** | 0.581 | 0.884 | 0.948 | 1.042 | 0.690 | 0.999 | 0.997 | 0.101 |
| | Electricity | **0.030** | 0.081 | 0.079 | 0.072 | 0.039 | 0.065 | 0.302 | 0.303 | 0.301 |
| | Traffic | 0.351 | 0.147 | 0.215 | 0.191 | **0.111** | 0.275 | 0.403 | 0.368 | 0.384 |
| | Taxi | **0.392** | 0.400 | 0.463 | 0.428 | 0.597 | 0.620 | 0.785 | 0.782 | 0.788 |
| | Exchange | 0.019 | 0.030 | **0.007** | 0.012 | 0.011 | 0.024 | 0.079 | 0.076 | 0.072 |

| Setting | Dataset | PINFDiT (FS) | DLinear | PatchTST | Latent ODE | GPT4TS | TransMAF | TimeGrad | CSDI | Diffusion-TS |
|---|---|---|---|---|---|---|---|---|---|---|
| Full Shot | Solar | **0.278** | 0.432 | 0.457 | 0.445 | 0.467 | 0.301 | 0.287 | 0.298 | 0.286 |
| | Electricity | **0.005** | 0.033 | 0.037 | 0.140 | 0.033 | 0.021 | 0.021 | 0.017 | 0.019 |
| | Traffic | 0.019 | 0.070 | 0.405 | 0.095 | 0.069 | 0.056 | 0.044 | 0.020 | 0.097 |
| | Taxi | 0.123 | 0.177 | 0.190 | 0.181 | 0.187 | 0.179 | **0.114** | 0.123 | 0.303 |
| | Exchange | **0.005** | 0.011 | 0.026 | 0.013 | 0.013 | 0.005 | 0.006 | 0.007 | 0.009 |

**Zero-shot Forecasting Setting.** For the zero-shot forecasting task, we utilized five widely-used open datasets to evaluate probabilistic time series forecasting performance. These datasets were

Table 17: Limited observation data Synthetic Generation results on 24-length multivariate time series. Discriminative and predictive scores are calculated as described in (Yoon et al., 2019).

| Metric | Methods | 0.05 | | | 0.1 | | |
|---|---|---|---|---|---|---|---|
| | | Sine | Air Quality | Energy | Sine | Air Quality | Energy |
| Discriminative Score | TimeGAN | 0.120(0.043) | 0.500(0.003) | 0.500(0.000) | 0.067(0.028) | 0.492(0.003) | 0.500(0.000) |
| | TimeVAE | 0.220(0.224) | 0.498(0.001) | 0.500(0.000) | 0.499(0.002) | 0.495(0.002) | 0.499(0.001) |
| | Diffusion-TS | 0.037(0.013) | 0.496(0.003) | 0.498(0.005) | 0.031(0.012) | 0.494(0.001) | 0.494(0.011) |
| | PINFDiT | **0.031(0.007)** | **0.456(0.003)** | **0.472(0.000)** | **0.030(0.009)** | **0.437(0.004)** | **0.447(0.002)** |
| Predictive Score | TimeGAN | 0.231(0.007) | 0.148(0.029) | 0.308(0.006) | 0.200(0.002) | 0.130(0.029) | 0.302(0.004) |
| | TimeVAE | 0.251(0.003) | 0.328(0.008) | **0.296(0.001)** | 0.238(0.002) | 0.308(0.014) | **0.288(0.001)** |
| | Diffusion-TS | 0.196(0.003) | 0.111(0.004) | 0.333(0.018) | 0.188(0.001) | 0.102(0.010) | 0.340(0.019) |
| | PINFDiT | **0.194(0.001)** | **0.089(0.005)** | 0.335(0.008) | **0.192(0.000)** | **0.070(0.007)** | 0.318(0.005) |

collected in GluonTS (Alexandrov et al., 2020) and have been previously employed in (Tashiro et al., 2021; Salinas et al., 2019). The task for these datasets is to predict the future $L_2$ steps given the observed $L_1$ steps. We set $L_1$ and $L_2$ values based on previous studies (Tashiro et al., 2021; Salinas et al., 2019). For training, we randomly selected $L_1 + L_2$ consecutive time steps as a single time series and designated the last $L_2$ steps as forecasting targets. We adhered to the train/test splits used in previous studies and utilized the last five samples of the training data as validation data. For the full-shot setting, we trained separate models on different datasets. Due to the large number of features in multivariate time series, we adopted subset sampling of features for training. For each input, we split them into subsets based on their order. If the last subset was smaller than the fixed shape, we applied padding to ensure equal input sizes across all subsets.

## C.2 SYNTHETIC GENERATION

We use 80% of all data for training and evaluation of the same data. For the air quality dataset, previous methods did not carefully use the -200 values as a placeholder for missing values. In our experiment, we masked all the -200 values for PINFDiT and baselines that support masks. For baselines that do not support mask, we replace -200 with the mean value. minmax scaler is used for all models. Diffusion-TS uses a different normalization scheme between -1 and 1. We replace its normalization scheme to be minmax scaler to ensure fair comparison. Figure 4 shows the PCA plots for all datasets and baselines. The visual comparison also validates the superiority of PINFDiT.

We also run the generation experiments with the limited data fine-tuning in Table 17. The generation experiments with limited data fine-tuning demonstrate PINFDiT's superior performance across various datasets and evaluation metrics. Comparing TimeGAN, TimeVAE, Diffusion-TS, and PINFDiT on sine, air, and energy datasets with 5% and 10% training data, PINFDiT consistently achieves the lowest Discriminative Scores, indicating its ability to generate the most realistic time series. In terms of Predictive Scores, PINFDiT outperforms or matches other models, particularly excelling in the air dataset. Notably, PINFDiT's performance remains robust or improves when increasing from 5% to 10% training data, showcasing its effectiveness in data-scarce scenarios. These results highlight PINFDiT's capability to capture complex temporal patterns and generate high-quality time series data, even with limited training samples, making it a promising tool for various time series generation tasks.

## C.3 ANOMALY DETECTION

We conduct experiments on five real-world datasets from industrial applications: MSL, SMAP, SWaT, SMD, and PSM. While PINFDiT's foundation capabilities benefit most tasks, anomaly detection's unique requirement to model dataset-specific normality led us to employ from-scratch training, highlighting the importance of matching deployment strategy to task characteristics. We introduced spectral residue (SR) transformation

Table 18: Threshold Sensitivity Analysis on Anomaly Detection Performance evaluated on F1 score

| Threshold | 99.5 | 99 | 98 | 97 | 96 | 95 |
|---|---|---|---|---|---|---|
| MSL | 83.9 | 89.33 | **90.1** | 88.17 | 85.28 | 82.84 |
| PSM | 96.32 | **97.57** | 96.78 | 95.72 | 94.66 | 93.61 |
| SMAP | **97.08** | 95.91 | 93.23 | 90.33 | 87.64 | 85.09 |
| SMD | **83.28** | 82.07 | 76.61 | 70.73 | 65.71 | 61.24 |
| SWAT | **97.6** | 96.46 | 93.49 | 90.74 | 88.0 | 85.42 |

at the preprocessing stage of PINFDiT. This transformation helps to conceal points most likely to be anomalies and their immediate neighbors. The number of neighbors affected is controlled by the hyperparameter $n_{neighbor}$. The SR method utilizes Fourier Transformation to convert the original time series into a saliency map, thereby amplifying abnormal points, as detailed in (Ren et al., 2019; Zhao et al., 2020). Consistent with prior methodologies, we set the sequence length to be 100 identify anomalies using the 99th percentile of reconstruction errors. During evaluations, we apply standard anomaly adjustments as suggested by (Xu et al., 2018). As demonstrated in Table C.3, PINFDiT outperforms baseline models on four of the five datasets. In particular, PINFDiT 23.03 points of improvement in terms of F1 score on the SMAP dataset compared to the previous best baseline. In addition, PINFDiT consistently outperforms both TimeMixer and iTransformer across all datasets, with particularly notable improvements on SMAP (95.91 vs 67.63/66.76) and SWAT (97.57 vs 88.84/92.63). These comprehensive comparisons against the latest models demonstrate PINFDiT's effectiveness as a unified framework for time series analysis, often achieving state-of-the-art performance while maintaining broader applicability across diverse tasks.

**Anomaly Detection Threshold** Our comprehensive analysis of threshold selection in Table 18 revealed that higher percentile thresholds, particularly the 99th and 99.5th percentiles, consistently yield superior performance. While we observed a systematic degradation in detection accuracy as threshold values decrease, we maintained the 99th percentile threshold to ensure fair comparison with existing methodologies. This decision reflects our commitment to methodological rigor, as optimizing threshold values based on test set performance would introduce bias in the comparative analysis. Our approach prioritizes consistent experimental conditions across all evaluated methods, enabling meaningful benchmark comparisons while acknowledging the impact of threshold selection on detection performance.

Table 19: Anomaly Detection result on 100-length multivariate time series. We calculate Precision, Recall, and F1 score as % for each dataset. '.' notation in model name stands for transformer. **Bold** indicates best result, Underline indicates the second best result. We replace the joint criterion in Anomaly Transformer with reconstruction error for fair comparison.

| Methods | MSL | | | SMAP | | | SWaT | | | SMD | | | PSM | | | 1st Pl |
|---|---|---|---|---|---|---|---|---|---|---|---|---|---|---|---|---|
| Metrics | P | R | F1 | P | R | F1 | P | R | F1 | P | R | F1 | P | R | F1 | Count |
| PINFDiT | 91.54 | 87.23 | **89.33** | 93.35 | 98.61 | 95.91 | 93.64 | 99.46 | 96.46 | 78.83 | 88.26 | 83.28 | 97.36 | 97.79 | **97.57** | 11 |
| GPT(6) | 82.00 | 82.91 | 82.45 | 90.60 | 60.95 | 72.88 | 92.20 | 96.34 | 94.23 | 88.89 | 84.98 | **86.89** | 98.62 | 95.68 | 97.13 | 1 |
| TimesNet | 89.54 | 75.36 | 81.84 | 90.14 | 56.40 | 69.39 | 90.75 | 95.40 | 93.02 | 87.91 | 81.54 | 84.61 | 98.51 | 96.20 | 97.34 | 0 |
| PatchTST | 88.34 | 70.96 | 78.70 | 90.64 | 55.46 | 68.82 | 91.10 | 80.94 | 85.72 | 87.26 | 82.14 | 84.62 | 98.84 | 93.47 | 96.08 | 0 |
| ETSformer | 85.13 | 84.93 | 85.03 | 92.25 | 55.75 | 69.50 | 90.02 | 80.36 | 84.91 | 87.44 | 79.23 | 83.13 | **99.31** | 85.28 | 91.76 | 1 |
| FEDformer | 77.14 | 80.07 | 78.57 | 90.47 | 58.10 | 70.76 | 90.17 | 96.42 | 93.19 | 87.95 | 82.39 | 85.08 | 97.31 | 97.16 | 97.23 | 0 |
| LightTS | 82.40 | 75.78 | 78.95 | 92.58 | 55.27 | 69.21 | 91.98 | 94.72 | 93.33 | 87.10 | 78.42 | 82.53 | 98.37 | 95.97 | 97.15 | 0 |
| DLinear | 84.34 | 85.42 | 84.88 | 92.32 | 55.41 | 69.26 | 80.91 | 95.30 | 87.52 | 83.62 | 71.52 | 77.10 | 98.28 | 89.26 | 93.55 | 0 |
| Autoformer | 77.27 | 80.92 | 79.05 | 90.40 | 58.62 | 71.12 | 89.85 | 95.81 | 92.74 | 88.06 | 82.35 | 85.11 | 99.08 | 88.15 | 93.29 | 0 |
| Anomaly. | 79.61 | **87.37** | 83.31 | 91.85 | 58.11 | 71.18 | 72.51 | 97.32 | 83.10 | **88.91** | 82.23 | 85.49 | 68.35 | 94.72 | 79.40 | 2 |
| TimeMixer | 89.72 | 75.42 | 81.95 | 89.51 | 54.34 | 67.63 | 91.56 | 86.28 | 88.84 | 86.60 | 71.50 | 78.33 | 99.18 | 87.74 | 93.11 | 0 |
| iTransformer | 86.16 | 62.64 | 72.54 | 90.69 | 52.82 | 66.76 | 92.21 | 93.06 | 92.63 | 86.92 | 77.75 | 82.08 | 97.98 | 92.81 | 95.32 | 0 |

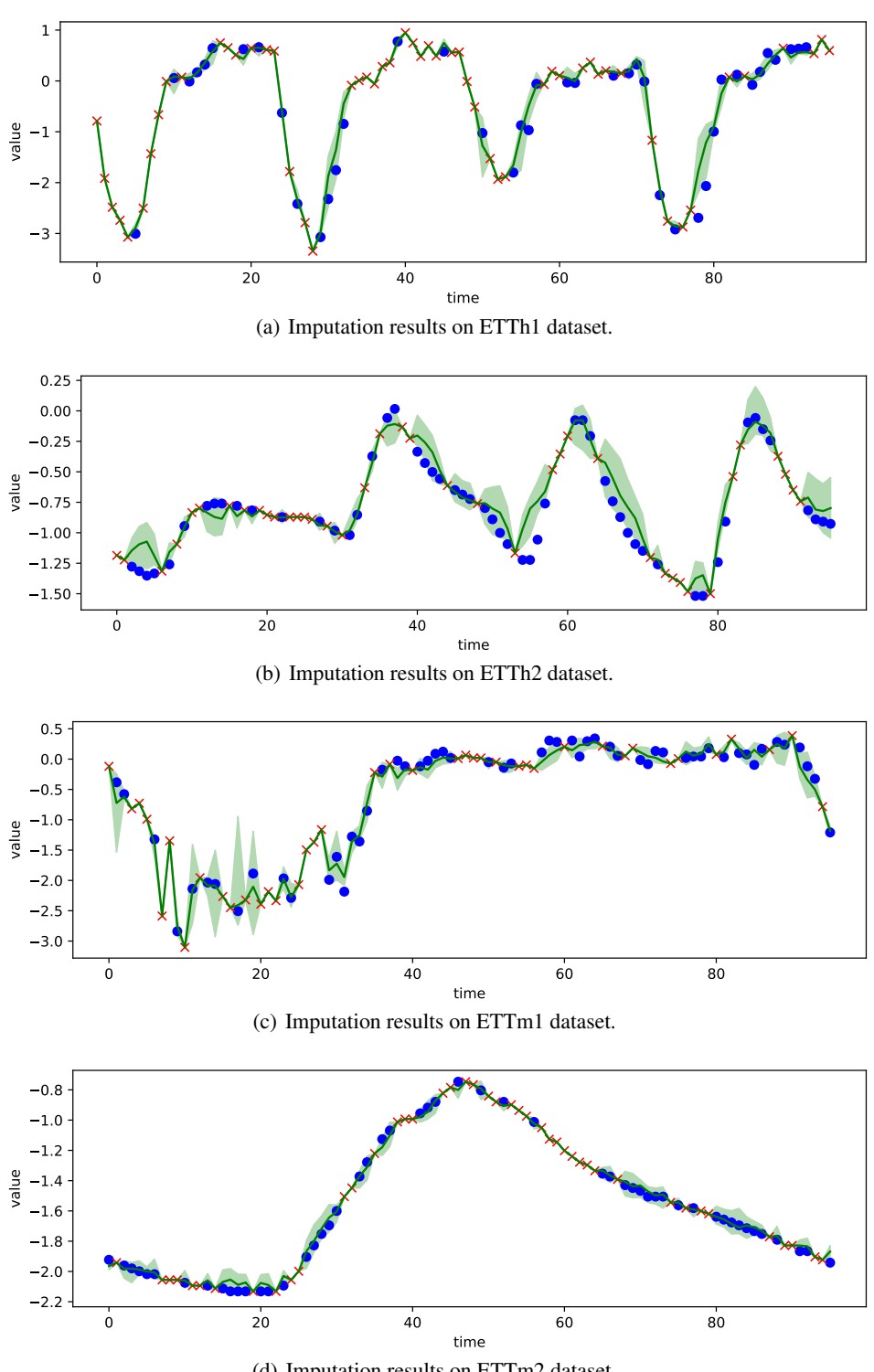

(a) Imputation results on ETTh1 dataset.

(b) Imputation results on ETTh2 dataset.

(c) Imputation results on ETTm1 dataset.

(d) Imputation results on ETTm2 dataset.

Figure 5: Visualization of imputation task on ETT datasets. This figure illustrates PINFDiT's performance, with red ×'s marking observed values, blue dots showing ground truth points for interpolation, a green line representing PINFDiT's mean of interpolation, and green shading indicating its estimated uncertainty intervals.

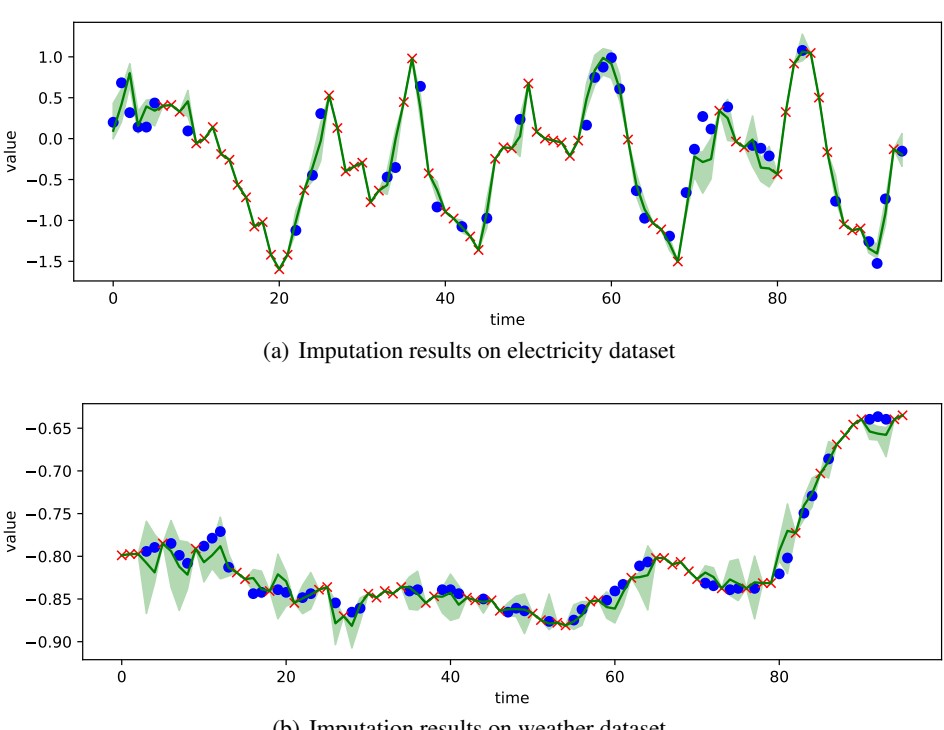

(a) Imputation results on electricity dataset

(b) Imputation results on weather dataset

Figure 6: Visualization of imputation task on electricity and weather datasets. This figure illustrates PINFDiT's performance, with red ×'s marking observed values, blue dots showing ground truth points for interpolation, a green line representing PINFDiT's mean of interpolation, and green shading indicating its estimated uncertainty intervals.

Table 20: Full result of imputation task.

| Methods Mask Ratio | | PINFDiT MSE MAE | Timer MSE MAE | TimeMixer MSE MAE | iTransformer MSE MAE | GPT2(3) MSE MAE | TimesNet MSE MAE | PatchTST MSE MAE | ETSformer MSE MAE | LightTS MSE MAE | DLinear MSE MAE | FEDformer MSE MAE | Stationary MSE MAE | Autoformer MSE MAE | Informer MSE MAE | Reformer MSE MAE | Scratch MSE MAE |
|---|---|---|---|---|---|---|---|---|---|---|---|---|---|---|---|---|---|
| ETTh1 | 12.5% | 0.022 0.096 | 0.119 0.222 | 0.094 0.203 | 0.099 0.221 | 0.043 0.140 | 0.057 0.159 | 0.093 0.201 | 0.126 0.263 | 0.240 0.345 | 0.151 0.267 | 0.070 0.190 | 0.060 0.165 | 0.074 0.182 | 0.114 0.234 | 0.074 0.194 | 0.025 0.107 |
| | 25% | 0.029 0.110 | 0.133 0.235 | 0.111 0.219 | 0.125 0.249 | 0.054 0.156 | 0.069 0.178 | 0.107 0.217 | 0.169 0.304 | 0.265 0.364 | 0.180 0.292 | 0.106 0.236 | 0.080 0.189 | 0.102 0.227 | 0.102 0.262 | 0.102 0.227 | 0.034 0.123 |
| | 37.5% | 0.039 0.127 | 0.151 0.249 | 0.124 0.233 | 0.158 0.281 | 0.072 0.180 | 0.084 0.196 | 0.120 0.230 | 0.220 0.347 | 0.296 0.382 | 0.215 0.318 | 0.124 0.258 | 0.102 0.212 | 0.109 0.222 | 0.174 0.293 | 0.135 0.261 | 0.047 0.143 |
| | 50% | 0.055 0.152 | 0.176 0.267 | 0.144 0.249 | 0.214 0.328 | 0.107 0.216 | 0.102 0.215 | 0.141 0.248 | 0.293 0.402 | 0.334 0.404 | 0.257 0.347 | 0.165 0.299 | 0.133 0.240 | 0.137 0.248 | 0.215 0.325 | 0.179 0.298 | 0.062 0.161 |
| | Avg | 0.036 0.122 | 0.145 0.243 | 0.119 0.226 | 0.149 0.270 | 0.069 0.173 | 0.078 0.187 | 0.115 0.224 | 0.202 0.329 | 0.284 0.373 | 0.201 0.306 | 0.117 0.246 | 0.094 0.201 | 0.103 0.214 | 0.161 0.279 | 0.122 0.245 | 0.042 0.133 |
| ETTh2 | 12.5% | 0.019 0.083 | 0.070 0.163 | 0.056 0.145 | 0.099 0.221 | 0.039 0.125 | 0.040 0.130 | 0.057 0.152 | 0.187 0.319 | 0.101 0.231 | 0.100 0.216 | 0.095 0.212 | 0.042 0.133 | 0.044 0.138 | 0.305 0.431 | 0.163 0.289 | 0.027 0.114 |
| | 25% | 0.024 0.098 | 0.074 0.168 | 0.063 0.157 | 0.130 0.254 | 0.044 0.135 | 0.046 0.141 | 0.061 0.158 | 0.279 0.390 | 0.115 0.246 | 0.127 0.247 | 0.137 0.258 | 0.049 0.147 | 0.050 0.149 | 0.322 0.444 | 0.206 0.331 | 0.034 0.130 |
| | 37.5% | 0.032 0.116 | 0.079 0.174 | 0.064 0.158 | 0.158 0.281 | 0.051 0.147 | 0.052 0.151 | 0.067 0.166 | 0.400 0.465 | 0.126 0.257 | 0.158 0.276 | 0.187 0.304 | 0.056 0.158 | 0.060 0.163 | 0.353 0.462 | 0.252 0.370 | 0.043 0.147 |
| | 50% | 0.051 0.148 | 0.085 0.182 | 0.071 0.168 | 0.214 0.328 | 0.059 0.158 | 0.060 0.162 | 0.073 0.174 | 0.602 0.572 | 0.136 0.268 | 0.183 0.299 | 0.232 0.341 | 0.065 0.170 | 0.068 0.173 | 0.369 0.472 | 0.316 0.419 | 0.055 0.169 |
| | Avg | 0.031 0.111 | 0.077 0.172 | 0.064 0.157 | 0.150 0.271 | 0.048 0.141 | 0.049 0.146 | 0.065 0.163 | 0.367 0.436 | 0.119 0.250 | 0.142 0.259 | 0.163 0.279 | 0.053 0.152 | 0.055 0.156 | 0.337 0.452 | 0.234 0.352 | 0.040 0.130 |
| ETTm1 | 12.5% | 0.014 0.078 | 0.044 0.131 | 0.046 0.136 | 0.045 0.147 | 0.017 0.085 | 0.023 0.101 | 0.041 0.130 | 0.096 0.229 | 0.093 0.206 | 0.080 0.193 | 0.052 0.166 | 0.032 0.119 | 0.046 0.144 | 0.063 0.180 | 0.042 0.146 | 0.016 0.083 |
| | 25% | 0.018 0.085 | 0.048 0.136 | 0.048 0.137 | 0.060 0.172 | 0.022 0.096 | 0.023 0.101 | 0.044 0.135 | 0.096 0.229 | 0.093 0.206 | 0.080 0.193 | 0.052 0.166 | 0.032 0.119 | 0.046 0.144 | 0.063 0.180 | 0.042 0.146 | 0.019 0.091 |
| | 37.5% | 0.023 0.096 | 0.053 0.144 | 0.059 0.155 | 0.078 0.196 | 0.029 0.111 | 0.029 0.111 | 0.049 0.143 | 0.133 0.271 | 0.113 0.231 | 0.103 0.219 | 0.069 0.191 | 0.039 0.131 | 0.057 0.161 | 0.079 0.200 | 0.063 0.182 | 0.025 0.102 |
| | 50% | 0.032 0.114 | 0.061 0.154 | 0.053 0.145 | 0.102 0.226 | 0.040 0.128 | 0.036 0.124 | 0.055 0.151 | 0.186 0.323 | 0.134 0.255 | 0.132 0.248 | 0.089 0.218 | 0.047 0.145 | 0.067 0.174 | 0.093 0.218 | 0.082 0.208 | 0.032 0.115 |
| | Avg | 0.022 0.093 | 0.051 0.141 | 0.051 0.143 | 0.071 0.185 | 0.028 0.105 | 0.027 0.107 | 0.047 0.140 | 0.125 0.263 | 0.104 0.218 | 0.093 0.206 | 0.062 0.166 | 0.036 0.126 | 0.051 0.150 | 0.071 0.188 | 0.055 0.166 | 0.023 0.098 |
| ETTm2 | 12.5% | 0.019 0.074 | 0.032 0.098 | 0.024 0.086 | 0.052 0.151 | 0.017 0.076 | 0.018 0.080 | 0.026 0.094 | 0.108 0.239 | 0.034 0.127 | 0.062 0.166 | 0.056 0.159 | 0.021 0.088 | 0.023 0.092 | 0.133 0.270 | 0.108 0.22 | 0.017 0.0658 |
| | 25% | 0.029 0.096 | 0.034 0.102 | 0.026 0.090 | 0.071 0.179 | 0.020 0.080 | 0.020 0.085 | 0.028 0.099 | 0.164 0.294 | 0.042 0.143 | 0.085 0.196 | 0.080 0.195 | 0.024 0.096 | 0.026 0.101 | 0.155 0.272 | 0.136 0.262 | 0.022 0.078 |
| | 37.5% | 0.039 0.114 | 0.036 0.106 | 0.029 0.094 | 0.091 0.204 | 0.022 0.087 | 0.023 0.091 | 0.030 0.104 | 0.237 0.356 | 0.051 0.159 | 0.106 0.222 | 0.110 0.231 | 0.027 0.103 | 0.030 0.108 | 0.155 0.293 | 0.175 0.300 | 0.027 0.089 |
| | 50% | 0.050 0.132 | 0.040 0.112 | 0.032 0.101 | 0.117 0.232 | 0.025 0.095 | 0.026 0.098 | 0.034 0.110 | 0.323 0.421 | 0.059 0.174 | 0.131 0.247 | 0.156 0.276 | 0.030 0.108 | 0.035 0.119 | 0.200 0.333 | 0.211 0.319 | 0.099 |
| | Avg | 0.034 0.104 | 0.035 0.105 | 0.028 0.093 | 0.083 0.192 | 0.021 0.084 | 0.022 0.088 | 0.029 0.102 | 0.208 0.327 | 0.046 0.151 | 0.096 0.208 | 0.101 0.215 | 0.026 0.099 | 0.029 0.105 | 0.156 0.292 | 0.157 0.280 | 0.024 0.083 |
| ECL | 12.5% | 0.049 0.142 | 0.077 0.174 | 0.047 0.154 | 0.073 0.190 | 0.085 0.202 | 0.055 0.160 | 0.089 0.206 | 0.196 0.321 | 0.102 0.229 | 0.092 0.214 | 0.107 0.237 | 0.093 0.210 | 0.089 0.210 | 0.218 0.326 | 0.190 0.308 | 0.051 0.148 |
| | 25% | 0.057 0.153 | 0.087 0.184 | 0.055 0.156 | 0.090 0.214 | 0.087 0.203 | 0.065 0.175 | 0.094 0.213 | 0.207 0.332 | 0.121 0.252 | 0.118 0.247 | 0.120 0.251 | 0.097 0.214 | 0.096 0.220 | 0.219 0.326 | 0.197 0.312 | 0.061 0.163 |
| | 37.5% | 0.067 0.175 | 0.101 0.199 | 0.064 0.169 | 0.107 0.234 | 0.094 0.211 | 0.076 0.189 | 0.094 0.213 | 0.219 0.344 | 0.141 0.273 | 0.144 0.276 | 0.136 0.266 | 0.102 0.220 | 0.104 0.229 | 0.222 0.328 | 0.203 0.315 | 0.074 0.181 |
| | 50% | 0.097 0.220 | 0.121 0.219 | 0.078 0.185 | 0.127 0.257 | 0.101 0.220 | 0.091 0.208 | 0.100 0.221 | 0.235 0.357 | 0.160 0.293 | 0.175 0.305 | 0.158 0.284 | 0.108 0.228 | 0.113 0.239 | 0.228 0.331 | 0.210 0.319 | 0.090 0.202 |
| | Avg | 0.068 0.172 | 0.097 0.194 | 0.061 0.164 | 0.099 0.224 | 0.090 0.207 | 0.072 0.183 | 0.092 0.210 | 0.214 0.339 | 0.131 0.262 | 0.132 0.260 | 0.130 0.259 | 0.100 0.218 | 0.101 0.225 | 0.222 0.328 | 0.200 0.313 | 0.069 0.174 |
| Weather | 12.5% | 0.029 0.033 | 0.107 0.168 | 0.030 0.054 | 0.039 0.089 | 0.026 0.049 | 0.025 0.045 | 0.029 0.049 | 0.057 0.141 | 0.047 0.101 | 0.039 0.084 | 0.041 0.107 | 0.027 0.051 | 0.026 0.047 | 0.037 0.093 | 0.031 0.076 | 0.029 0.033 |
| | 25% | 0.031 0.033 | 0.108 0.167 | 0.029 0.043 | 0.047 0.108 | 0.028 0.052 | 0.029 0.052 | 0.031 0.053 | 0.065 0.155 | 0.052 0.111 | 0.048 0.103 | 0.064 0.163 | 0.029 0.056 | 0.030 0.054 | 0.042 0.100 | 0.035 0.082 | 0.031 0.033 |
| | 37.5% | 0.034 0.037 | 0.108 0.167 | 0.042 0.047 | 0.055 0.121 | 0.033 0.060 | 0.031 0.057 | 0.035 0.058 | 0.081 0.180 | 0.058 0.121 | 0.057 0.117 | 0.107 0.229 | 0.033 0.062 | 0.032 0.060 | 0.049 0.111 | 0.040 0.091 | 0.034 0.037 |
| | 50% | 0.031 0.041 | 0.109 0.168 | 0.035 0.051 | 0.070 0.145 | 0.037 0.065 | 0.034 0.062 | 0.038 0.063 | 0.102 0.207 | 0.065 0.133 | 0.066 0.134 | 0.183 0.312 | 0.037 0.068 | 0.037 0.067 | 0.063 0.114 | 0.046 0.099 | 0.031 0.041 |
| | Avg | 0.031 0.036 | 0.108 0.168 | 0.031 0.049 | 0.053 0.116 | 0.031 0.056 | 0.030 0.054 | 0.060 0.144 | 0.076 0.171 | 0.055 0.117 | 0.052 0.110 | 0.099 0.203 | 0.032 0.059 | 0.031 0.057 | 0.045 0.104 | 0.038 0.087 | 0.031 0.036 |

## C.4 IMPUTATION

For visual representation of PINFDiT's imputation capabilities, we have plotted the results in Figure 5 and Figure 6, which clearly illustrates the model's accuracy in reconstructing missing data points across different datasets and missing data ratios. The imputation task results, presented in Table C.3, demonstrate PINFDiT's superior performance across various datasets and missing data ratios. All baseline models are trained in a full-shot setting, while PINFDiT leverages a pre-trained foundation model, fine-tuning it on realistic datasets. PINFDiT consistently achieves the lowest Mean Squared Error (MSE) and Mean Absolute Error (MAE) scores in most scenarios, outperforming state-of-the-art models such as GPT2, TimesNet, and PatchTST. Notably, PINFDiT's performance remains robust even as the proportion of missing data increases from 12.5% to 50%, showcasing its ability to handle substantial data gaps effectively. The model's imputation accuracy is particularly impressive for the ETTh1, ETTh2, ETTm1, and ETTm2 datasets, where it maintains a significant lead over other methods. PINFDiT demonstrates superior performance on most datasets, achieving significant improvements over Timer, TimeMixer, and iTransformer, particularly on ETT datasets where we see reductions in MSE by up to 60%. PINFDiT maintains strong overall performance while offering greater versatility

# D ANALYSIS ON PINFDiT

We present a comprehensive analysis of PINFDiT's design space, conducting systematic comparisons across different architectural variants. To ensure fair evaluation, all experiments maintain consistent training configurations, utilizing the same checkpoint and number of training steps. This rigorous experimental setup allows us to isolate and assess the impact of individual architectural components while controlling for training conditions.

Table 21: Ablation study on zero-shot forecasting

| Dataset | PINFDiT | w/o Random Mask | w/o Stride Mask | w/o Block Mask | w/o Phys |
|---|---|---|---|---|---|
| Solar | **0.424** | 0.465 | 0.469 | 0.862 | 0.445 |
| Electricity | **0.030** | 0.035 | 0.037 | 0.101 | 0.033 |
| Dataset | Dual-attention | Channel-wise | Patch Token | Additive | Cross-attention |
| Solar | 0.467 | 0.461 | 0.874 | 0.677 | 0.711 |
| Electricity | 0.037 | 0.039 | 0.145 | 0.079 | 0.077 |

## D.1 ABLATION STUDY

Our comprehensive ablation studies, detailed in Sections E1, E2, and E3, systematically evaluate PINFDiT's architectural choices. In Section E1, with particular emphasis on the Transformer design strategy, we explore PINFDiT's temporal-wise attention mechanism and compare it against alternative

approaches, including channel-wise attention and dual attention mechanisms (as discussed in (Yu et al., 2024)). The analysis demonstrates that temporal-wise processing significantly outperforms traditional patch-based tokenization approaches, achieving substantially lower error rates.

This performance disparity can be attributed to two key factors: First, while channel relationships exhibit model-specific variations, temporal patterns provide more universal characteristics across time series data, enabling better generalization. Second, patch-based approaches introduce additional hyperparameter dependencies (patch length and stride settings) that compromise the model's universal applicability. These findings validate our design choice of temporal-wise processing as a more robust and generalizable approach for time series modeling. The empirical results strongly support our architectural decisions, demonstrating that PINFDiT's temporal-focused design effectively captures universal temporal dynamics while maintaining model flexibility across diverse applications and domains. In addition, the Physics-Informed component yields consistent performance improvements across all datasets, with notable enhancements in Electricity and Solar predictions, underscoring the value of incorporating physical constraints during inference.

### D.2 MASK: HANDLING MISSING VALUES

**Time Series Mask Unit.** The Time Series Mask Unit is a key component of our model, designed to enhance its versatility and performance across various time series tasks. This unified mechanism incorporates multiple mask types that seamlessly integrate with the model throughout its lifecycle - from self-supervised task-agnostic pre-training to task-specific fine-tuning and inference. The time series mask unit generates four distinct mask types: random mask $\mathbf{M}^{\mathrm{R}}$, block mask $\mathbf{M}^{\mathrm{B}}$, stride mask $\mathbf{M}^{\mathrm{S}}$, and reconstruction mask $\mathbf{M}^{\mathrm{Rec}}$. During task-agnostic pre-training, these masks help the model develop robust and generalizable features from the input data, improving overall time series representation. In task-specific training, the masks adapt to the unique requirements of common downstream tasks such as forecasting and imputation, enabling the model to specialize effectively.

Given $\mathbf{x} \in \mathbb{R}^{K \times L}$, the random mask $\mathbf{M}^{\mathrm{R}}$ can be generated by:

$$\mathbf{M}^{\mathrm{R}}(x, r) = \begin{cases} 1 & z_{i,j} > r, z \in \mathbb{R}^{K \times L}, z \sim Uniform(0,1) \\ 0, & otherwise, \end{cases} \tag{58}$$

where $r$ is the mask ratio. For task-specific training and inference, we allow the user to supply customized imputation masks, which replace the random position masks, that could handle the naturally missing data and multi-resolution cases. In addition, block mask $\mathbf{M}^{\mathrm{B}}$ can be generated via:

$$\mathbf{M}^{\mathrm{B}}(x, l) = \begin{cases} 1 & j < L - l, \\ 0, & otherwise, \end{cases} \tag{59}$$

where $l$ is the predicted length. This mask offers flexibility across different stages of model development and application: during pre-training, a random $l$ exposes the model to various forecasting horizons, while in fine-tuning and inference, a fixed $l$ aligns with specific task requirements. Moreover, stride mask $\mathbf{M}^{\mathrm{S}}$, a variant of $\mathbf{M}^{\mathrm{B}}$, is designed for intermittent placement within time series during task-agnostic pretraining:

$$\mathbf{M}^{\mathrm{S}}(x, n_{\mathrm{blocks}}) = \begin{cases} 1 & \lfloor \frac{j}{b} \rfloor \bmod 2 = 0 \\ 0 & otherwise, \end{cases} \tag{60}$$

where $n_{block}$ is the number of blocks; $b = \lceil \frac{L}{n_{\mathrm{blocks}}} \rceil$ is the length of each block; $j$ is the index of the sequence. It improves the modeling of temporal and inter-correlated dependencies by integrating information across non-contiguous parts of time series, leveraging neighboring values as additional context. In addition, reconstruction mask $\mathbf{M}^{\mathrm{Rec}} = 0$ is employed for tasks such as synthetic data generation and anomaly detection. It allows direct generation of synthetic data or calculation of anomaly scores for each temporal position by comparing original and reconstructed series.

Our experimental design leverages naturally occurring missing values inherent in real-world datasets, primarily arising from irregular sampling rates and multi-resolution data collection processes. This approach authentically validates model robustness against genuine missing data patterns rather than artificially generated scenarios. PINFDiT incorporates a comprehensive masking strategy that aligns with three well-established missing data mechanisms: Missing Completely at Random (MCAR)

using uniform distribution-based random masks, Missing at Random (MAR) employing block and stride masks to capture structured patterns and dependencies between non-contiguous observations, and Missing Not at Random (MNAR) utilizing reconstruction masks with physics-informed sampling for scenarios where missing patterns correlate with unobserved variables. These mechanisms are simultaneously applied through self-supervised learning, enabling robust representation learning without requiring explicit knowledge of the underlying missing data processes. Our comprehensive ablation studies in Table 21 demonstrate the criticality of each masking strategy, where removing any mask type leads to performance degradation, with future masks showing the most significant impact. These findings validate our integrated approach to handling diverse missing data scenarios in time-series analysis.

## D.3 CONDITION SCHEME FOR PINFDiT

As mentioned in Section 3.1, AdaLN's superior performance stems from its ability to dynamically adjust feature distributions across different layers while maintaining computational efficiency. This approach aligns well with the inherent nature of time series data, where temporal dependencies typically exhibit gradual rather than dramatic changes in both seen and unseen time steps. We conducted comparative experiments to evaluate different conditioning mechanisms in PINFDiT:

- Additive conditioning, which adds conditional information directly to the diffusion input;

- Cross-attention, which uses conditional time series as keys/values and noisy time series as queries to fuse conditional information;

- Token concatenation, which concatenates conditional time series with noisy time series at the input level before PINFDiT processing.

The experimental results (Table 21) across Solarand Electricity datasets consistently show that AdaLN achieves superior performance compared to the next best alternative. This significant performance gap validates our choice of AdaLN as PINFDiT's primary conditioning mechanism.

## D.4 NOISE EMBEDDING JUSTIFICATION

PINFDiT's noise embedding approach plays multiple key roles in the diffusion modeling framework. The diffusion process operates directly in a continuous embedding space, allowing for smoother transitions between noise levels and better preserving the inherent time dependence, thus enabling the model to learn a more robust representation of the underlying time series structure. This approach has several technical advantages (Ho et al., 2020; Peebles & Xie, 2022; Lu et al., 2024): the embedding space provides a continuous representation in which the diffusion process can operate more efficiently. The direct em-

Table 22: Performance metrics for weather and ecl datasets on different model sizes.

|  |  | S | | B | | L | |
|---|---|---|---|---|---|---|---|
|  |  | MSE | MAE | MSE | MAE | MSE | MAE |
| Weather | 0.125 | 0.029 | 0.033 | 0.029 | 0.026 | 0.025 | 0.024 |
|  | 0.250 | 0.031 | 0.033 | 0.033 | 0.029 | 0.028 | 0.027 |
|  | 0.375 | 0.034 | 0.037 | 0.036 | 0.033 | 0.031 | 0.031 |
|  | 0.500 | 0.031 | 0.041 | 0.042 | 0.039 | 0.036 | 0.036 |
|  | Avg | 0.031 | 0.036 | 0.035 | 0.032 | 0.030 | 0.029 |
| ECL | 0.125 | 0.051 | 0.148 | 0.050 | 0.144 | 0.048 | 0.140 |
|  | 0.250 | 0.061 | 0.163 | 0.060 | 0.158 | 0.058 | 0.154 |
|  | 0.375 | 0.074 | 0.181 | 0.071 | 0.175 | 0.069 | 0.170 |
|  | 0.500 | 0.090 | 0.202 | 0.087 | 0.197 | 0.084 | 0.190 |
|  | Avg | 0.069 | 0.174 | 0.067 | 0.169 | 0.065 | 0.163 |

bedding of noisy samples helps prevent the embedding space from collapsing during training. From a practical point of view, this approach allows for parallel processing of multiple time steps, handles varying degrees of noise through a unified framework, and makes the diffusion process more stable compared to traditional generation methods. In addition, the embedded noise representation allows for the seamless incorporation of physical constraints and maintains temporal continuity while progressively denoising, thus contributing to a better quantification of the uncertainty in the generated samples.

## D.5  FAILURE SCENARIOS ANALYSIS

PINFDiT's performance shows notable degradation in three key scenarios: highly irregular sampling rates deviating from training distributions, complex non-stationary patterns underrepresented in pretraining data, and domain-specific patterns requiring expert knowledge beyond general time series characteristics. As shown by SMD dataset for anomaly detection (Table C.3) where it achieves 83.28% F1 score versus GPT2's 86.89%. This dataset represents cloud server machine metrics with high-frequency sampling and complex feature interdependencies. Additionally, when dealing with extremely short-term patterns or highly localized anomalies, specialized architectures like GPT2 that focus intensively on recent temporal context may outperform PINFDiT's more holistic approach, as our diffusion-based generation process may occasionally smooth over abrupt local changes. These limitations, primarily stemming from the model's dependence on learned foundational patterns, become particularly relevant in specialized industrial applications and unique financial scenarios. Understanding these boundaries is crucial for informed model deployment decisions and highlights promising directions for future research.

## D.6  DYNAMIC ON MODEL SIZE

The experimental results demonstrate a clear correlation between PINFDiT's model size and its imputation performance across different datasets and missing data ratios. As shown in Table 22, as the model size increases from Small (S) to Big (B) to Large (L), we observe consistent improvements in both averaged Mean Squared Error (MSE) and averaged Mean Absolute Error (MAE) metrics. The Large model consistently outperforms the Small and Big variants across all scenarios, with the most significant gains observed in the weather dataset. Notably, larger models (B and L) show better resilience to increased proportions of missing data compared to the Small model. The improvement is more pronounced for the weather dataset than for the ecl dataset, suggesting that the benefits of increased model size may vary depending on the nature and complexity of the time series data. The consistent performance gains from S to B to L models indicate that PINFDiT's architecture scales well with increased model size. These findings suggest that increasing PINFDiT's model size is an effective strategy for improving imputation accuracy, particularly for complex datasets or scenarios with higher proportions of missing data. However, the performance may remain relatively consistent across all model sizes for both the weather and ecl datasets, even as the proportion of missing data increases from 12.5% to 50%. This stability in performance suggests that PINFDiT's architecture may achieve its optimal capacity for these imputation tasks even at smaller model sizes. Thus, the trade-off between computational resources and performance gains should be considered when selecting the appropriate model size for specific applications.

# E  EXPERIMENTS SETTING

## E.1  BASELINES

We conduct a comprehensive comparative analysis, benchmarking PINFDiT against a diverse array of leading models in the field. Our analysis extends to state-of-the-art probabilistic models, encompassing TimeGAN (Yoon et al., 2019), TimeVAE (Desai et al., 2021), Diffusion-TS (Yuan & Qiao, 2024), CSDI (Tashiro et al., 2021), TimeGrad (Rasul et al., 2021), TransMAF (Rasul et al., 2020), GP-copula (Salinas et al., 2019), and TSDiff (Kollovieh et al., 2023). We also evaluate against cutting-edge deterministic models, including DLinear (Zeng et al., 2023), GPT-2 (Zhou et al., 2023), TimesNet (Wu et al., 2023), PatchTST (Nie et al., 2023), ETSformer (Woo et al., 2022), FEDformer (Zhou et al., 2022), LightTS (Zhang et al., 2022), Autoformer (Wu et al., 2021), and Anomaly Transformer (Xu et al., 2021), LatentODE and LatentCDE(Rubanova et al., 2019), etc. Furthermore, we include comparisons with recent forecasting foundation models, such as TEMPO (Cao et al., 2023b), Moirai (Woo et al., 2024b), and LagLLama (Rasul et al., 2023). This extensive comparison allows us to thoroughly evaluate PINFDiT's performance across a wide spectrum of methodologies and architectures in time series modeling.

## E.2 DATASETS

Please refer to **Chronos** (`https://huggingface.co/datasets/autogluon/chronos_datasets`) for our pre-trained dataset and the evaluation dataset are listed as follows:

1. The ETT (Electricity Transformer Temperature) datasets (Zhou et al., 2021)[1] include electricity load data at various resolutions (ETTh & ETTm) from two different electricity stations.

2. The Weather dataset (Zhou et al., 2021)[2] comprises 21 meteorological indicators collected in Germany over the span of one year.

3. The Electricity (ECL, Electricity Consuming Load) (Zhou et al., 2021)[3] dataset provides information on electricity consumption.

4. The SMD dataset (Su et al., 2019) includes multivariate time-series data collected from server machines in a data center. It typically contains metrics such as CPU usage, memory usage, and disk activity.

5. The PSM dataset (Abdulaal et al., 2021) is used for predictive maintenance and includes sensor data from industrial machines. It often contains readings such as temperature, pressure, and vibration over time.

6. The MSL dataset (Hundman et al., 2018) comes from the Mars Science Laboratory mission, specifically the Curiosity rover. It includes telemetry data from the rover's sensors and systems.

7. The SWaT dataset (Mathur & Tippenhauer, 2016) originates from a scaled-down water treatment testbed designed to reflect a real-world water treatment process. It includes sensor and actuator data collected over time.

8. The SMAP dataset (Hundman et al., 2018) comes from NASA's Soil Moisture Active Passive (SMAP) mission, which measures soil moisture and freeze/thaw state. It includes time-series data from multiple sensors aboard the SMAP satellite.

9. The Sine dataset (Yoon et al., 2019) is synthetically generated by sinusoidal waves.

10. The Air Quality dataset (Yi et al., 2016) [4]contains hourly averaged readings from five metal oxide chemical sensors integrated into an Air Quality Chemical Multisensor Device. This device was positioned at road level in a highly polluted area of an Italian city. Data were collected from March 2004 to February 2005, making it the longest freely available record of on-field air quality chemical sensor responses.

11. The Stock dataset (Yoon et al., 2019)[5] contains daily historical Google stocks data from 2004 to 2019.

12. The UCI Appliances Energy prediction dataset (Yoon et al., 2019)[6]consists of multivariate, continuous-valued measurements including numerous temporal features measured at close intervals.

13. The Weather_2 dataset (Godahewa et al., 2021a): The Weather_2 dataset comprises hourly climate TSD collected near Monash University, Clayton, Victoria, Australia, from January 2010 to May 2021. It includes series for temperature, dewpoint temperature, wind speed, mean sea level pressure, relative humidity, surface solar radiation, surface thermal radiation, and total cloud cover.

14. The PhysioNet dataset (Silva et al., 2012)[7] contains clinical time series data from 12,000 ICU patients, each with 42 vital variables recorded over 48 hours with naturally missing values. Patients are evenly divided into three groups of 4,000 each. For benchmarking

---

[1]ETT: `https://github.com/zhouhaoyi/ETDataset`

[2]Weather:`https://www.ncei.noaa.gov/data/local-climatological-data/`

[3]ECL:`https://archive.ics.uci.edu/ml/datasets/ElectricityLoadDiagrams20112014`

[4]Air Quality: `https://archive.ics.uci.edu/dataset/360/air+quality`

[5]Stock: `https://finance.yahoo.com/quote/GOOG`

[6]Energy: `https://archive.ics.uci.edu/ml/datasets`

[7]The PhysioNet: `https://physionet.org/content/challenge-2012/1.0.0/`

purposes, we select 7 out of 42 variables. To address varying scales, we apply standard normalization, resulting in features with zero mean and unit variance.

15. MIMIC-III (Bica et al., 2020)[8]: MIMIC-III dataset contains 5000 patient ICU records with 19 variables from the lab events table including 'anion gap, albumin, bands, bicarbonate, bilirubin, creatinine, chloride, glucose, hematocrit, hemoglobin, lactate, platelet, potassium, PTT, INR, PT, sodium, BUN, WBC'. They are irregularly sampled and we process them following the previous works (Bica et al., 2020; Cao et al., 2023a), which have naturally missing values.

16. NASDAQ: NASDAQ Top 10 Stocks dataset comprises time series data for the ten largest companies by market capitalization listed on the NASDAQ stock exchange. The dataset includes daily and 5-day price data for each stock from 2014-2024, offering two temporal resolutions for comprehensive analysis. We predict the close prices of each company in the multi-resolution forecasting task.

17. Monash dataset archive (Godahewa et al., 2021b): The Monash repository contains 30 datasets, including publicly available time series datasets in various formats and those curated by us. Many datasets have different versions based on frequency and the inclusion of missing values. We use their multivariate time series version for pre-training and evaluation (specified if needed).

Table 23: Dataset details

| Dataset | Domain | Length | Dimension | Frequency |
|---|---|---|---|---|
| ETTh | Energy | 17420 | 7 | 1 hour |
| ETTm | Energy | 69680 | 7 | 15 min |
| Weather | Nature | 52696 | 21 | 10 min |
| Electricity | Energy | 26304 | 321 | 1 hour |
| Air Quality | Nature | 9357 | 13 | 1 hour |
| Sine | Synthetic | 10000 | 5 | N/A |
| Stock | Finance | 3685 | 6 | 1 day |
| Energy | Energy | 19745 | 28 | 10 min |
| MSL | Space | 132046 | 55 | 1 min |
| PSM | Cloud | 220322 | 25 | 1 min |
| SMAP | Space | 562800 | 25 | 1 min |
| SMD | Cloud | 1416825 | 38 | 1 min |
| SWaT | Energy | 944920 | 51 | 1 second |
| Requests Minute | Cloud | 64800 | 10 | 1 min |
| Function Delay Minute | Cloud | 64800 | 10 | 1 min |
| Platform Delay Minute | Cloud | 64800 | 10 | 1 min |
| Memory Usage Minute | Cloud | 64800 | 10 | 1 min |
| CPU Limit Minute | Cloud | 64800 | 10 | 1 min |
| Memory Limit Minute | Cloud | 64800 | 10 | 1 min |
| Instances Minute | Cloud | 64800 | 10 | 1 min |
| Weather_2 | Climate | 3001 | 695 | 1 day |
| PEMS_SF | Traffic | 4320 | 852 | 1 hour |
| PhysioNet(b) | Health Care | - | 7 | Irregular |
| PhysioNet(b) | Health Care | - | 7 | Irregular |
| PhysioNet(c) | Health Care | - | 7 | Irregular |
| MIMIC-III | Health Care | - | 19 | 1 day |
| NASDAQ | Finance | 2516 | 20 | Multiresolution |

### E.3 METRICS

**MAE** describes the mean absolute error that measures the absolute difference between ground truth and prediction.

$$\text{MAE} = \frac{1}{n} \sum_{i=1}^{n} |y_i - \hat{y}_i| \tag{61}$$

---

[8]MIMIC-III: `MIMIC-III:https://physionet.org/content/mimiciii/1.4/`

**MSE** describes the mean squared difference between ground truth and prediction.

$$\text{MSE} = \frac{1}{n} \sum_{i=1}^{n} (y_i - \hat{y}_i)^2 \tag{62}$$

**RMSE** is the sqaure root of MSE.

$$\text{RMSE} = \sqrt{\frac{1}{n} \sum_{i=1}^{n} (y_i - \hat{y}_i)^2} \tag{63}$$

**Discriminative score** Following TimeGAN, we train a post-hoc time-series classification model (by optimizing a 2-layer LSTM) to distinguish between sequences from the original and generated datasets. First, each original sequence is labeled real, and each generated sequence is labeled not real. Then, an off-the-shelf (RNN) classifier is trained to distinguish between the two classes as a standard supervised task. We then report the classification error on the held-out test set.

**Predictive Score** Following TimeGAN, we train a post-hoc sequence-prediction model (by optimizing a 2-layer LSTM) to predict next-step temporal vectors over each input sequence. Then, we evaluate the trained model on the original dataset. Performance is measured in terms of the mean absolute error (MAE); for event-based data, the MAE is computed as the absolute value of 1 - estimated probability that the event occured.

**Computations of CRPS** We explain the definition and calculation of the CRPS metric. The continuous ranked probability score (CRPS) assesses how well an estimated probability distribution $F$ aligns with an observation $x$. It is defined as the integral of the quantile loss $\Lambda_\alpha(q, z) := (\alpha - \mathbf{1}_{z<q})(z - q)$ over all quantile levels $\alpha \in [0, 1]$:

$$\text{CRPS}(F^{-1}, x) = \int_0^1 2\Lambda_\alpha(F^{-1}(\alpha), x)\, d\alpha \tag{64}$$

where $\mathbf{1}$ represents the indicator function. We then calculated quantile losses for quantile levels discretized in 0.05 increments. Thus, we approximated CRPS as follows:

$$\text{CRPS}(F^{-1}, x) \approx \frac{1}{19} \sum_{i=1}^{19} 2\Lambda_{i \cdot 0.05}(F^{-1}(i \cdot 0.05), x). \tag{65}$$

Next, we computed the normalized average CRPS for all features and time steps:

$$\text{CRPS Score} = \frac{\sum_{k,l} \text{CRPS}(F_{k,l}^{-1}, x_{k,l})}{\sum_{k,l} |x_{k,l}|} \tag{66}$$

where $k$ and $l$ denote the features and time steps of the imputation targets, respectively. The lower the CRPS, the more accurate the model, i.e., the closer the predicted probability is to the observed outcome.

**Computations of CRPS_sum** CRPS_sum measures CRPS for the distribution $F$ of the sum of all $K$ features, calculated by:

$$\text{CRPS\_sum Score} = \frac{\sum_l \text{CRPS}(F^{-1}, \sum_k x_{k,l})}{\sum_{k,l} |x_{k,l}|} \tag{67}$$

where $\sum_k x_{k,l}$ is the total of the forecasting targets for all features at time point $l$.

**Precision** Precision measures the accuracy of positive predictions made by a model. It is defined as the ratio of true positives (TP) to the total number of predicted positives, which includes both true positives and false positives (FP). Mathematically, precision is expressed as:

$$\text{Precision} = \frac{TP}{TP + FP} \tag{68}$$

**Recall** Recall, also known as sensitivity, measures a model's ability to correctly identify true positive instances. It is calculated as the ratio of true positives (TP) to the sum of true positives and false negatives (FN). In the context of anomaly detection, failing to detect an anomalous timestamp can have serious consequences, making recall a critical metric. Mathematically, recall is defined as:

$$\text{Recall} = \frac{TP}{TP + FN} \tag{69}$$

**F1-score** The F1-score is a balanced measure of model performance that combines Recall and Precision. It is calculated as the harmonic mean of these two metrics, giving equal importance to both. This score effectively captures the trade-off between Recall and Precision, penalizing significant disparities between them. By providing a single, comprehensive metric, the F1-score offers a more holistic view of a model's effectiveness, particularly useful when dealing with imbalanced datasets.

$$\text{F1} = 2 \times \frac{Precision \times Recall}{Precision + Recall} \tag{70}$$

## F   PINFDiT Paradigm on Training and Inference

**Borader Impact and Position of PINFDiT** PINFDiT bridges the critical gap between specialized deep learning baselines and the ideal of a universal time series foundation model. While pioneering works such as CSDI have successfully leveraged transformer architectures for probabilistic imputation, they generally rely on random masking strategies and concatenation-based conditioning. These design choices often constrain their effectiveness when generalizing to diverse downstream tasks like long-horizon forecasting or physically consistent generation. In contrast, PINFDiT embodies the characteristics of a true foundation model through a balanced design that fuses data-driven versatility with domain-specific rigor. Its foundation model status is established through three distinctive advancements: (1) **Unified Representation Learning:** Unlike predecessors that focus primarily on random imputation masks, our Time Series Mask Unit (TSMU) integrates structured masking strategies (block, stride, and reconstruction). This allows the model to learn robust temporal dependencies across varying channel sizes and sequence lengths without requiring task-specific architecture changes. (2) **Task-Agnostic Versatility:** By employing Adaptive Layer Normalization (AdaLN) for deep conditional fusion, PINFDiT creates a highly responsive generative framework capable of seamlessly transitioning between forecasting, imputation, anomaly detection, and synthetic generation within a single pre-trained model. (3) **Physics-Informed Reliability:** Uniquely, we introduce a physics-informed sampling mechanism via Langevin dynamics. This energy-based approach allows for the modular injection of domain knowledge (e.g., conservation laws) during inference, ensuring scientific plausibility without the need for expensive retraining.

Although technical challenges such as extreme sequence length scaling remain, this combination of architectural flexibility, structured pre-training, and post-hoc physical consistency positions PINFDiT as a practical "proto-foundation" model, advancing the field from single-task solvers toward universal, scientifically grounded time series modeling.

**Synergy between Architecture and Diffusion on Irregular Data.** We attribute our superior performance on irregular and missing data not merely to the use of a diffusion framework, but to the specific synergy between the diffusion objective and our architectural innovations. While diffusion models inherently support flexible conditioning, it is our *Time Series Mask Unit (TSMU)* and conditioning strategies that fully exploit this capability.

Specifically, our ablation studies demonstrate that:

- **AdaLN Conditioning:** Replacing standard addition or cross-attention with Adaptive Layer Normalization (AdaLN) significantly lowers error, proving essential for robust temporal modeling.

- **Stride Masking:** The use of stride masking during training teaches the model to handle non-contiguous dependencies, a capability that standard U-Net architectures (like CSDI) often lack.

- **Temporal-wise Attention:** Our "What You See Is What You Get" embedding with temporal attention is critical; replacing it with standard Patch Tokens results in a severe performance drop.

Thus, PINFDiT succeeds where specialized ODE/CDE models struggle because the diffusion process provides the generative flexibility, while our specialized transformer architecture provides the structural capacity to model complex, irregular temporal dynamics.

**Training details** Similar to the previous DiT work (Peebles & Xie, 2022), PINFDiT is available in four sizes: small (S, 33M parameters), big (B, 130M parameters), large (L, 460M parameters), and extra large (XL, 680M parameters). A comprehensive comparison in Table 25 shows PINFDiT's expanded task coverage relative to existing general-purpose time series models, including anomaly detection, imputation and data generation. In our training process, we utilized the Adam optimizer with a learning rate of 0.0001 and the loss function is from Equantion **??**. Batch sizes of 256 or 512 were employed, depending on model size. The ideal epoch to convergence is over 100 as the complexity of training data, but we choose to use the earlier checkpoint for the case of downstream purpose of anomaly detection and synthetic generation because the two tasks are very dataset-specific and do not necessarily benefit from learning distributions beyond the target dataset. In practice, the maximum channel number ($K_{max}$) was set to 20-40, with a maximum sequence length of 198, unless otherwise specified. All experiments were conducted on NVIDIA A100 GPUs with 40G GPU memory. Importantly, our zero-shot foundation model was trained on **Chronos dataset** without exposure to any data from the evaluated downstream tasks or datasets. The results of zero-shot section are derived from a single pre-trained checkpoint, evaluated with or without fine-tuning based on the settings. The only exception is the long-term zero-shot experiments, which require extended sequence inputs while still utilizing the same pre-training dataset. To facilitate reproducibility and further research, we will release the pre-trained checkpoint.

**Inference** In the finetuning and inference stage, the choice of mask is tailored to align with the specific requirements of the user. This flexibility allows PINFDiT to apply the most appropriate masking strategy based on the context of the task and application. During inference, while the mask type and parameters are fixed for a given task to ensure consistency, PINFDiT's generative task architecture allows for flexible transformation of various downstream tasks. This adaptability enables us to address a wide range of time series challenges within a unified framework. Let $n$ represent the number of samples generated for each prediction, which we set to $n = 10$ ($n = 30$ for forecasting tasks) in our experimental setup at inference time. We use the median of these $n$ predictions as the final prediction, providing the added benefit of obtaining a confidence interval for PINFDiT's predictions. To prevent channel padding from affecting the generated samples, we mask out the invalid channels during sampling at each diffusion timestep so that PINFDiT does not falsely treat the information in the non-valid channels as meaningful information. Padding is applied at the beginning of the temporal dimension to ensure that the most relevant information remains at the end, thereby mitigating the effect of padding. We have included inference time comparisons for single-sample generation, where PINFDiT demonstrates superior computational efficiency, requiring only 1 second for single-sample generation, making it more practical for real-world applications. Regarding the physics-informed module, the Langevin refinement introduces a minimal computational overhead of approximately 0.003s per iteration. For a standard setting with $K = 10$ refinement steps, the total inference time for a batch of 50 samples is approximately 1.05s. This efficiency stems from the optimized gradient computation,

Table 24: Comparison of inference times for single-sample generation.

| Model | Inference Time (mm:ss) |
|---|---|
| Diffusion-TS | 00:06 |
| CSDI | 00:02 |
| PINFDiT | 00:01 |

which only requires $\mathcal{O}(N \times T)$ operations (where $N = 40$ spatial and $T = 192$ temporal points), making the physics guidance practical for real-time applications.

Table 25: A comparable analysis of representative general purposes time series models

| Model | Parameter Size | Model Architecture | Channel Setting | Task Type | Pretrain Dataset | Data Size |
|---|---|---|---|---|---|---|
| Lag-LLama | - | Transformer | Univariate | Forecasting | Monash (Godahewa et al., 2021b) | 300 Million Time Points |
| Moriai | S: 14M
B: 91M
L: 311M | Transformer | Univariate | Forecasting | LOTSA (Woo et al., 2024b) | 27 Billion Time Points |
| PINFDiT | S: 33M
B: 130M
L: 460M
XL: 680M | Transformer + Diffusion | Multivariate | Forecasting,
Imputation,
Anomaly Detection,
Data Generation | Chronos Ansari et al. (2024) | About 5 Billion Time Points |

**Limitations and Future Work.** Despite PINFDiT's strong performance, several limitations remain. First, our model inherits computational constraints common to diffusion models, requiring multiple sampling steps that increase inference time compared to single-pass methods. This presents challenges for real-time applications requiring instant predictions. Second, while PINFDiT effectively handles missing values and irregular sampling, extremely sparse time series with large gaps may still present difficulties, particularly when physical constraints operate across these gaps. Third, our current physics injection method works best with explicit differential equations; incorporating implicit physical knowledge or constraints expressed through other mathematical formulations remains challenging. Fourth, like most foundation models, PINFDiT's performance depends on the diversity and quality of the pretraining data, potentially limiting generalization to entirely novel physical systems not represented in the training distribution. Finally, while we demonstrate strong performance on various scientific time series benchmarks, the evaluation of extremely long-term dependencies (spanning thousands of time steps) and the incorporation of multi-modal data (e.g., simultaneous text descriptions and time series measurements) represent important directions for future work. Addressing these limitations will be crucial for developing truly universal foundation models for scientific time series analysis.

## G    USE OF LARGE LANGUAGE MODELS (LLMS)

We utilized large language models (LLMs) as a general-purpose writing assistant to slightly polish the manuscript. The LLM's role was strictly limited to improving grammar, clarity, and style. The LLM did not contribute to the research ideation, experimental design or the core scientific conclusions presented in this paper. Therefore, the LLM is not regarded as a contributor to this work.

