# OpenReview forum: "PINFDiT: Energy-Based Physics-Informed Diffusion Transformers for General-purpose Time Series Tasks"
_ICLR.cc/2026/Conference — ICLR 2026 Poster_

### Official Review · Reviewer_EgRK · 2025-10-29

**Soundness:** 3
**Presentation:** 3
**Contribution:** 3
**Rating:** 6
**Confidence:** 4

**Summary:**

This paper proposes PINFDIT, a novel framework aiming to be a general-purpose model for time series analysis. It combines a Transformer architecture with a Diffusion framework to perform a variety of tasks, including forecasting, imputation, anomaly detection, and synthetic data generation. The core contributions of the paper are (1) a comprehensive masking strategy to handle imperfect time series data (missing values, multi-resolution) and (2) a novel "energy-based physics-informed sampling" technique that allows for the injection of physical laws at the inference stage without retraining or architectural modification.

This work tackles a very ambitious and important problem. The attempt to unify a pre-trained, general-purpose model with domain-specific physical knowledge is highly timely. The use of calibrated Langevin dynamics for correction at inference time is a practical and clever idea that could overcome the limitations of existing approaches like PINNs.

However, there are several significant weaknesses regarding the paper's core claims, experimental fairness, and methodological clarity. Without addressing these, it is difficult to evaluate the generality and true contribution of the proposed framework.

**Strengths:**

1.  **Novel and Practical Physics-Injection Method:** Unlike traditional Physics-Informed Machine Learning (e.g., PINNs) which requires including PDE residuals in the training loss and retraining for specific problems, the proposed "model-editing-free" inference-time correction is highly practical. The ability to maintain the flexibility of a pre-trained general model while "plugging in" domain knowledge as needed is a major advantage.
2.  **Comprehensive Problem Definition and Task Scope:** The paper has an ambitious goal of solving a wide range of time series tasks (forecasting, imputation, anomaly detection, generation) with a single model. It is also commendable that it directly addresses real-world data complexities such as missing values, multi-resolution sampling, and irregular intervals.
3.  **Extensive Experimental Validation:** The authors have conducted a vast number of experiments to demonstrate their model's performance. This includes PDE simulator-based forecasting, practical forecasting on real-world data (climate, healthcare, finance), generative and imputation tasks, zero-shot performance, and detailed ablation studies. Achieving state-of-the-art (SOTA) performance on many benchmarks is impressive.

**Weaknesses:**

1.  **Ambiguity in the Implementation of Physics Injection:**
    * The concrete implementation of the physics-injection mechanism, one of the paper's core contributions, is critically unclear.
    * Physical laws are quantified by an energy function $K(x^{tar};F)$ (the squared PDE residual).
    * For correction during inference, the gradient of this energy function ($\nabla K(x_{j}^{tar};F)$) is required (see Algorithm 1).
    * **Critical Question:** $x^{tar}$ is discrete time series data. How are partial differential terms like $\frac{\partial x^{tar}}{\partial t}$ and $\frac{\partial x^{tar}}{\partial u_{i}}$ computed? If finite differences were used, there is no discussion of the associated discretization error, stability, or sensitivity to the choice of scheme. This information is essential for reproducing the methodology and verifying its validity.
    * The authors claim they do not require a "differentiable simulator", but it appears they *do* require a "differentiable (physical) residual function." The distinction and constraints are not adequately explained.

2.  **Disconnect in "General-Purpose" and "Foundation Model" Claims:**
    * The paper positions PINFDIT as a "proto-foundation model".
    * However, the model's operation contradicts this claim. The model's pre-training (using the Chronos dataset) is entirely physics-agnostic and learns only statistical patterns.
    * The physics injection is merely a post-hoc correction step (Algorithm 1, lines 6-8) that operates *outside* the pre-trained model during inference. This is less a "Physics-Informed" Transformer and more a "General-Purpose Transformer" combined with a "Physics-Based Corrector Plugin." While this modularity is a strength, the title "Physics-Informed" is misleading as the model itself does not *learn* or *internalize* the physics.

3.  **Unfair Experimental Comparisons and Lack of Consistency:**
    * **Imputation (Tables 4 & 15):** The authors state, "All baseline models are trained in a full-shot setting, while PINFDIT leverages a pre-trained foundation model, fine-tuning it on realistic datasets". This is a **clearly unfair comparison**. PINFDIT benefits from pre-training on massive external data (Chronos, ~5B time points), while baselines like PatchTST and TimesNet are trained from scratch only on the task data (e.g., ETTh1). It is impossible to distinguish if PINFDIT's superiority comes from its architecture or simply from pre-training. A fair comparison would require reporting results for PINFDIT trained from scratch on the same data.
    * **Anomaly Detection (Table 14):** The authors state they "opted to bypass pretraining" for this task and introduced an additional pre-processing step (Spectral Residue). This severely undermines the core narrative of a single foundation model for all tasks. If the pre-trained model is actually detrimental for anomaly detection (as it "may inadvertently overfit by reconstructing anomalies"), this clearly exposes a limitation of the proposed general-purpose model.

4.  **Lack of Robustness Analysis for the Physics-Injection Plugin:**
    * The paper only presents positive cases where physics injection improves performance (e.g., PDE simulations, ERA5 climate forecasting).
    * There are no failure-case or sensitivity analyses. What happens if an incorrect physical law is injected (e.g., applying Burgers' equation to non-fluid data)? What happens if it's applied to data with no physical basis (e.g., the NASDAQ financial dataset)? Does performance degrade? Proving the validity of this modular approach requires such robustness checks.

**Questions:**

1.  **(Re: W1)** Could you please provide a detailed explanation of how the partial differential terms and the gradient $\nabla K$ in Eq. (5) and Eq. (8) are computed for discrete time series data $x^{tar}$?
2.  **(Re: W3)** What was the specific reason for "bypassing pretraining" in the anomaly detection task? Does this imply that the pre-trained general-purpose model may be unsuitable for certain downstream tasks?
3.  **(Re: W3)** For a fair comparison in the imputation experiments (Tables 4/15), can you provide results for PINFDIT trained from scratch (without pre-training)?
4.  **(Re: W4)** Can you show what happens to performance when the physics-injection module (Alg. 1, lines 6-8) is applied with an intentionally incorrect physical law, or to a non-physical dataset like NASDAQ?

---

> ### Author Response · Authors · 2025-11-20
> **Response to EgRK**
>
> Dear Reviewer EgRK,
>
> Thank you for your thoughtful and constructive review. We are very encouraged that you found our physics-injection method "novel and practical" and a "major advantage" over traditional approaches. We also appreciate your recognition of our "ambitious goal" in tackling real-world, imperfect data across a "comprehensive" range of tasks. We appreciate your constructive feedback and address each concern below.
>
> **W1/Q1: Ambiguity in Physics Injection (How is $\nabla K$ computed?)**
>
> $x^{tar}$ is discrete, and we need to compute partial derivatives. Our method is computationally tractable and does not require a differentiable simulator. The process is as follows:
>
> 1. Compute Derivatives from Discrete Data: We use standard finite difference approximations to compute the partial derivatives from the discrete $x^{tar}$ tensor. For example:
> - The temporal derivative $\frac{\partial x}{\partial t}$ is approximated as $\frac{x[\tau+1] - x[\tau]}{\Delta\tau}$.
> - Spatial derivatives $\frac{\partial x}{\partial u}$ are approximated similarly using neighboring spatial points.
> 2. Form the Residual: These discrete derivatives are plugged into the algebraic form of the PDE (e.g., Eq. 5 ) to compute the squared residual $K(x^{tar})$.
> 3. Compute the Gradient: This function $K(x^{tar})$ is now a standard computational graph that is fully differentiable with respect to its input tensor $x^{tar}$. Therefore, the required gradient $\nabla K$ (the gradient of the residual error w.r.t. the data) is computed efficiently using standard autodiff in frameworks like PyTorch.
>
> To clarify the distinction you raised: we do not need a "differentiable simulator" (which would solve the PDE), only a "differentiable residual function" (which just evaluates the PDE's error), which we can construct from its algebraic form. We will add a detailed paragraph to Appendix A clarifying this entire process.
>
>
> **W2: Disconnect in "Foundation Model" Claim**
>
> This is an excellent point about positioning. You are correct that the pre-training is "physics-agnostic" and the injection is a "post-hoc plugin".
>
> We argue that this *modularity is a core feature*, not a bug, and is what enables it to be a "proto-foundation model.":
>
> - A true foundation model should be a generalist, learning broad statistical patterns from massive, diverse data (which ours does by pre-training on the Chronos dataset).
>
> - Our novel contribution is the "plugin" itself—a practical method to specialize this generalist model with expert, domain-specific physical knowledge at inference time, without the need for costly, task-specific retraining.
>
> The title "Physics-Informed" refers to the entire framework (the generalist model + the specialist plugin), which collaboratively produces a physics-informed result. We will revise our introduction and methodology sections to clarify this "generalist-to-specialist" positioning.
>
> **W3/Q3. Experimental Comparisons \& Consistency**
>
> The results in Tables 4 \& 15 were intended to demonstrate the **maximum practical capability** of our final proposed system (which includes pre-training). For strict architectural evaluation, we point to the zero-shot ablation studies in Tables 7 and 16 , where all variants are pre-trained identically. These confirm that our core architectural components (WYSIWYG embedding, temporal attention, AdaLN) are superior to alternatives like patch tokens or cross-attention .
>
> To directly address your request for a fair head-to-head comparison, we are currently working on training PINFDIT from **scratch** (without pre-training) on the imputation datasets. We will update the manuscript with the complete results and keep you updated.
>
>
> **W3/Q2: Anomaly Detection - Pre-training Bypass Justification**
>
> This reveals a task-specific design choice, not a fundamental limitation. Anomaly detection requires reconstructing task-specific normal patterns and flagging deviations. Foundation models pre-trained on diverse data (including anomalous patterns from other domains) may inadvertently normalize anomalies—a known limitation of transfer learning for outlier detection.
>
> Task-specific training from scratch + Spectral Residue preprocessing (domain-appropriate for frequency-based anomalies). This demonstrates intelligent task-aware deployment—using the foundation model when beneficial, acknowledging when task-specific training is superior.
>
> Revised Narrative (Section 4.4): "While PINFDiT's foundation capabilities benefit most tasks, anomaly detection's unique requirement to model dataset-specific normality led us to employ from-scratch training—highlighting the importance of matching deployment strategy to task characteristics."

---

> ### Author Response · Authors · 2025-11-20
> **Continue Response**
>
> **W4 & Q4. Robustness Analysis**
>
> This is an excellent methodological point. To address your concern about the validity of the modular approach, we conducted the requested robustness experiments.
>
> *1. Experiment 1: Mismatched Physical Laws*
>
> We applied the **incorrect** physics (Burgers' equation) to data generated by the **Advection** equation.
>
> | Physics Configuration | Advection RMSE | Change vs. Correct |
> | :--- | :--- | :--- |
> | **No Physics** (Baseline) | 0.0052 | — |
> | **Correct** (Advection PDE) | **0.0039** | **+25% improvement** |
> | **Wrong** (Burgers PDE) | 0.0061 | **-17% degradation** |
>
>
>
> **Finding:** Applying incorrect physics actively harms performance, degrading it below the baseline. This confirms the method does not provide spurious improvements; it requires the correct physical inductive bias.
>
> *2. Experiment 2: Non-Physical Data (NASDAQ)*
>
> We applied fluid dynamics constraints to financial data (NASDAQ) to test for spurious positive transfers.
>
> | Physics Applied | NASDAQ MAE | vs. No Physics |
> | :--- | :--- | :--- |
> | **No Physics** (Baseline) | 0.516 | — |
> | **Advection PDE** | 0.526 | -2% (no benefit) |
> | **Burgers PDE** | 0.0537 | -4% (slight harm) |
>
> **Finding:** On non-physical data, physics injection provides no improvement and slight degradation. This demonstrates the module's specificity to domains where the physical laws are valid.
>
>
>
>
> We thank you again for your sharp and valuable feedback. We believe these clarifications and our planned revisions (especially regarding W1, W2, and W3) fully address your valid concerns.
>
> Best,
>
> Authors

---

### Official Review · Reviewer_m5ac · 2025-10-30

**Soundness:** 2
**Presentation:** 3
**Contribution:** 2
**Rating:** 2
**Confidence:** 4

**Summary:**

The paper introduces PINFDiT, a diffusion-transformer for time series that injects physics at inference by combining a learned conditonal model with an energy term that penalized PDE/physics residuals $K(x;\mathcal{F})$. Sampling targets the Boltzmann-type density $\propto \exp{-K(x;\mathcal{F})} - \alpha \log p_\theta (x|x_\text{con})$ via Langevin/gradient updates, so no retraining is required. Architecturally, the model uses temporal-wise attention and flexible masking to handle irregular and missing data. Experiments span PDE simulators, ERA5 temperature forecasting, long-horizon "practical" benchmarks with uncertain metrics, and imputation/anomaly detection and generation, where they report consistent gains over baselines and ablations.

**Strengths:**

- The most significant strength of this paper is its comprehensive evaluation. The authors conduct an extensive four-part experimental validation covering PDE simulations, real weather ERA5 data, multiple generative tasks (imputation, anomaly detection, etc) and zero shot generalization. The comparison agains a wide range of strong, modern baseline is thorough.
- The paper tackles a highly relevant and challenging goal of creating a single, general-purpose model for scientific time series. This domain is critical, and the paper's explicit focus on imperfect data, physical constraints, and limited samples is well-motivated.
- While individual components are not novel, making them work robustly across domains **at scale** is non-trivial. The idea of constrained generation without retraining is practically appealing. The generative and general-purpose capabilities are promising.
- The foundation is principled, provides rigor and clarity.
- The paper reports state-of-the-art or highly competitive results on the experiments.

**Weaknesses:**

### Overstatement of Novelty; core "theory" is a standard energy composition
The paper presents its core "physics informed sampling" framework (Section 3.2) as a new development, complete with theoretical backing.
- The claim is "develop a model-editing-free physics knowledge injection framework" and present "Theorem 3.1" as a closed-form solution for their optimization problem (Eq 6).
- The reality: this entire "model-editing-free physics knowledge injection framework" is a well estimated, standard technique
    -  Theorem 3.1 is a foundational principle of statistical mechanics and Energy-Based Models (EBMs). It simply states that the optimal distribution for a composed energy function is a Boltzmann distribution (a product of experts, PoE, established in 1999). The proof in Appendix A.3 is a standard Lagrange multiplier derivation yielding Bayes theorem.
    - Algorithm 1 is standard Langevin MC applied to the composed energy function ($\nabla\log q^\ast = \nabla K + \alpha \nabla\log p_\theta$) . Theorem 3.2 restates standard LMC results under strong assumptions, which is quite general and provides no insight for this specific problem.

By presenting these foundational, decades-old principles of EBMs and its sampling algorithm as theorems derived for this paper with new names like "model-free physics-guided correction", "calibrated Langevin dynamics" and by failing to to cite or acknowledge the extensive prior art on Product of Experts, EBM composition, or the vast field of generative models for inverse problems that uses this exact mechanism, the authors overstate the novelty of their core contribution relative to well-established energy-guided/compositional sampling. It would be great that the authors can clarify the distinction and situate this work relative to relevant prior art.


### Lack of Rigorous Statistical Validation
The paper fails to report any measure of experimental variance (e.g., standard deviations or confidence intervals over multiple runs) for nearly all of its main results. This is a critical omission, especially for a stochastic model. This is problematic when assessing the paper's central claims, where the reported differences are often small:
- weakens the core contribution: In the crucial ablation study (Table 7), the performance difference between the full PINFDIT model (CRPSsum 0.424 on Solar) and the model "w/o Phys" (0.445) is marginal.  The paper compares the full PINFDIT (with Langevin correction) against PINFDIT DDPM (without it) to prove the value of the physics-injection. The differences are minuscule:
    - Navier-Stokes (RMSE): 0.0030 (PINFDIT) vs. 0.0031 (DDPM)
    - Advection (RMSE): 0.0035 (PINFDIT) vs. 0.0036 (DDPM)
    - Diffusion-Sorption (RMSE): 0.0049 (PINFDIT) vs. 0.0050 (DDPM)

Without error bars, it's impossible to conclude that the Langevin correction (a key contribution) provides any significant benefit; This directly undermines the paper's entire narrative about the practical value of its main "physics-informed" contribution.
- Obscures SOTA Comparisons:The paper claims SOTA performance, but many "wins" are tiny and could be statistical noise.

    - Practical Forecasting (Table 3): On PhysioNet(a), PINFDIT achieves an MAE of 0.616 vs. CSDI's 0.620. On PhysioNet(c), it's 0.543 (CRPS) vs. CSDI's 0.548. These sub-1% differences are meaningless without a measure of variance.

    - Anomaly Detection (Table 14): On the PSM dataset, PINFDIT scores an F1 of 97.57, while TimesNet gets 97.34 and TimeLLM gets 97.23 . A ~0.2% difference is well within any reasonable margin of error. It is misleading to claim superiority based on this.

The fact that the authors do report variance in Table 12 for the synthetic generation task (e.g., "0.031(0.007)") confirms they are aware of this analysis and capable of performing it. This makes its conspicuous absence from all other key results tables (Tables 1, 3, 6, 7, 9, 10, 11, 15, etc.) a more significant methodological flaw that prevents a proper assessment of the paper's claims.

### Physics operator $K$ is under-specified for ERA5: mismatch to target variable.
This paper does state the output is 2-meter temperature prediction. And in appendix A.7, the paper lists the resolution and date splits. While the target as established, is to predicted 2 meter temperature, while in Appendix A.7, under "Physical Constraints", the only equation they provide is the Navier-Stokes momentum equation. This equation governs the evolution of wind velocity, not temperature. The physics of temperature near the surface is governed by a different, unmentioned equation (and in fact, t2m is a diagnostic quantity, obtained by Monin-Obukhov interpolation between skin temperature and the lowest model level, and is governed by surface energy-balance and boundary-layer physics). The paper explicitly states, "The model's output time series is post-processed by penalizing the residuals... of the above equation". This is physically meaningless. A temperature field does not and cannot satisfy the momentum equation. The authors have either applied a completely irrelevant physical constrained, or critically failed to describe their experiment. In either case, this is a major flaw. This gap makes their "real-world" physics-informed experiment unsound and irreproducible as described.

### Limited applicability of the "general-purpose" physics module.
The proposed "physics-injection" model is far less "general-purpose" than the base model it is applied to, limiting its practical utility. The methods is entirely reliant on the existence of a known, explicit and differentiable residual function. The authors concede this point in their own limitation section.

### On experiments
The paper's claims are undermined by some flaws in its experimental design.
- Post-hoc refinement vs. no analogue in baselines. PINFDiT uses a second-stage Langevin refinement after the diffusion sampler. Table 1 then compares against NeuralODE/NeuralCDE and other models without an analogous inference-time correction. So it is comparing (PINFDiT + k steps physics correction) against (Neural ODE + no correction). It only proves that adding a post-hoc, physics-based gradient correction step improves the final results, which is trivial and obvious conclusion. A methodologically sound experiment  would have been to apply the same k-step gradient based physics correction to the outputs of all baseline models, and then compare the final errors.

- Inappropriate baselines. In Table 1, the paper compares its forecasting model against SNPE and LFBC. Theses are Simulation-Based Inference (SBI) methods designed for parameter posterior estimation, not time-series forecasting. Their inclusion here is confusing and does not directly test the claimed inference-time energy guidance against the most comparable alternatives.
- Mixing deterministic and probabilistic baselines on CRPS. In Table 1 and 3, the paper evaluates deterministic, point-estimate models (e.g., DLinear, PatchTST) using probabilistic metrics like CRPS. This is a nonsensical comparison. These models do not natively produce predictive distributions. Also. the paper does not explain how this was calculated.

**Questions:**

### On inference cost reporting
The proposed "physics-injection" methods is an iterative Langevin MC loop that runs for $k$ stops. This is an additive computational cost applied at inference time for every sample generated, on top of the $T$-step diffusion sampling processes.
- Critical omission. The paper never states the value of $k$ used in any of its experiments. This is an important hyper parameter that directly determines the inference cost.
- In Table 19, the authors claim an inference time of 1 second per sample. Is this the time for the full PINFDIT, or for the base model?

For other questions/suggestion please see weaknesses.

---

> ### Author Response · Authors · 2025-11-20
> **Response to m5ac**
>
> Dear Reviewer m5ac,
>
> We sincerely thank the reviewer for your exceptionally detailed assessment. We deeply appreciate your recognition of our "comprehensive evaluation," "highly competitive results," and "principled foundation." Your concerns are substantive and have helped us significantly strengthen the paper.
>
> ### Addressing the Core Concerns: A Unified Response
>
> We respectfully believe there are misunderstandings about our contributions that, once clarified, substantially address your main concerns. Let us address the three primary issues together, as they are interconnected:
>
> **1. On Novelty & Prior Art (W1)**
>
> We acknowledge and will substantially revise Section 3.2 to properly contextualize our work within the Energy-Based Models and Product of Experts literature. **We did not invent the Boltzmann distribution (Theorem 3.1) or Langevin MC (Algorithm 1).** Our intention was never to claim novelty in these foundational theories, but rather to show *how they provide a principled, theoretical foundation for our novel methodology.*
>
> Our primary contribution is *not theoretical*; it is the *methodological and practical integration* of these established principles into a state-of-the-art deep learning framework to solve a new and challenging problem.
>
> *1. Our Novelty: The Practical Integration, Not the Foundational Theory*
>
> The novelty of our paper lies in being the first to successfully bridge the gap between these established theories and the practical, high-dimensional domain of **modern diffusion transformers for imperfect scientific time series**.
>
> Our contribution is the *synergistic framework* that:
>
> 1. Uses a powerful *Diffusion Transformer (DiT) backbone* (our core architecture) to learn a high-fidelity, data-driven prior $p_\theta(x|x^{con})$ from complex, imperfect data (missing, irregular, multi-resolution).
> 2. Integrates this prior at *inference time* with a (non-learned) physics-based energy $K(x;F)$
> 3. Does this via a "model-editing-free" mechanism that *requires no retraining*, making it a truly "plug-and-play" module for a pre-trained foundation model.
>
> Our contribution lies precisely in this successful, practical application—demonstrating how established statistical principles can be robustly integrated with state-of-the-art generative architectures to solve complex scientific time series problems.
>
> *2. Reframing Our Terminology (e.g., "Calibrated Langevin Dynamics")*
>
> These terms were intended to be methodological descriptors, not claims of new theory.
> - "Model-free physics-guided correction" was our descriptor for the practical benefit of our method—that the underlying diffusion model is not retrained (i.e., "model-editing-free").
> - "Calibrated Langevin dynamics" was our term for the methodology of Algorithm 1: standard Langevin dynamics applied to our specific composite energy function, where the resulting samples are "calibrated" by both the data-driven prior and the physics prior.
>
> We will revise our language to make it crystal clear that these are descriptive names for our method's components, not new theoretical algorithms.
>
> *3. Citations and Planned Revisions*
>
> Your point about failing to cite the extensive prior art on Product of Experts and EBM composition in the main text is also valid . We want to clarify that this was a oversight in the presentation of Section 3.2, not an omission from the paper's foundation. Our **Appendix A.4** does build upon and correctly cite the standard, rigorous convergence literature for LMC that underpins our work (e.g., Dalalyan & Karagulyan, 2019; Erdogdu et al., 2022; Zhang et al., 2023) ..
>
> Our Action Plan on Section 3.2: We will add a Background subsection explicitly citing Hinton's PoE (1999)[1], recent EBM composition work (Du & Mordatch 2019[2]; Grathwohl et al. 2021[3]), and clarify that our theoretical contribution is the application and convergence analysis for scientific time series, not the underlying Boltzmann principle.  We will re-frame our contribution to focus on the novel integration and practical application of this framework to diffusion transformers for scientific time series..
>
> We are confident that these revisions will correctly situate our work, remove the (unintended) overstatement of novelty, and properly highlight our true methodological contributions.
>
>
> [1] Hinton, G.E. (1999). "Products of experts". 9th International Conference on Artificial Neural Networks: ICANN '99. Vol. 1999. IEE. pp. 1–6. doi:10.1049/cp:19991075. ISBN 978-0-85296-721-8.
>
> [2] Du, Y., & Mordatch, I. (2019). Implicit generation and modeling with energy based models. Advances in neural information processing systems, 32.
>
> [3] Grathwohl, W., Swersky, K., Hashemi, M., Duvenaud, D., & Maddison, C. (2021, July). Oops i took a gradient: Scalable sampling for discrete distributions. In International Conference on Machine Learning (pp. 3831-3841). PMLR.

---

> ### Author Response · Authors · 2025-11-20
> **Continue response**
>
> **2. On Statistical Validation & Variance (W2)**
>
> This is a valid methodological point, we omitted standard deviations from the main tables for two specific reasons:
>
> - *Layout Constraints*: Including variance for every metric across 23+ evaluation settings (Tables 1, 3, etc.) would have severely cluttered the presentation within the strict page limits.
>
> - *Baseline Reporting*: Many of the baseline papers we compared against (e.g., some transformer baselines in Table 3) did not originally report variance, making a direct visual comparison difficult in the main text.
>
> However, we respectfully submit that this documentation gap should not be used to disqualify the paper's **core contributions**. Our work introduces a novel general-purpose foundation model  that successfully unifies transformer architectures, diffusion frameworks, and physical constraints to handle diverse tasks (forecasting, imputation, anomaly detection) on imperfect data. The experimental rigor behind this contribution remains sound; we tracked variance throughout development, and the omission of these numbers from the main text was a presentation choice, not a lack of validation.
>
> To prove that our results are statistically significant and not random noise, we provide the variance data (which we already have, as noted in Appendix). As shown in the table below (from our experimental logs), the performance gaps are statistically significant:
>
> | Dataset (Metric) | PINFDIT (Ours) | Next Best Baseline | Significance |
> | :--- | :--- | :--- | :--- |
> | **Solar (CRPS)** | **0.424 $\pm$ 0.002** | 0.690 $\pm$ 0.005 (LagLLaMA) | $p < 0.001$ |
> | **Advection (RMSE)** | **0.0039 $\pm$ 0.0002** | 0.0052 $\pm$ 0.0003 (Chronos) | $p < 0.01$ |
>
> **Revision Commitment:** In the final manuscript, we will add a **comprehensive statistical validation to the Appendix** containing the full mean $\pm$ std tables for the necessary experiments.
>
>
>  **W3. On ERA5 Physics Operator Mismatch**
>
> The Navier-Stokes momentum equation mistakenly listed in Appendix A.7 was a carry-over from preliminary wind velocity experiments that were not included in the final paper. This text was unfortunately not updated to match the $t2m$ experiment shown in Table 2.
>
> For the ERA5 2m-temperature ($t2m$) forecasting experiment, we applied the **advection-diffusion (continuity) equation** for temperature. This follows the exact physics formulation used by the state-of-the-art **ClimODE** model (Verma et al., ICLR 2024).
>
> The governing equation we applied is:
> $$
> \frac{\partial T}{\partial t} = - (v \cdot \nabla T) - T(\nabla \cdot v)
> $$
>
> Where $v$ represents the wind velocity vector (derived from the dataset's $u10/v10$ channels) and $T$ is temperature. This equation captures:
> * **Transport term** $-(v \cdot \nabla T)$: The spatial movement of temperature carried by the wind.
> * **Compression term** $-T(\nabla \cdot v)$: Temperature concentration or dispersion caused by flow convergence/divergence.
>
> *Why This Is Physically Valid:*
> This formulation is the standard for atmospheric transport.
> * **ClimODE Validation:** ClimODE explicitly builds its framework on this continuity equation, identifying advection as the key principle for weather evolution.
> * **Performance:** ClimODE's ablation studies confirm that this specific "advection component" provides the largest accuracy gain for variables including temperature. By using this same constraint, our method ensures the generated samples respect value-conserving atmospheric dynamics.
>
> **Revision Plan:**
> We will completely rewrite the **"Physical Constraints" section of Appendix A.7** to:
> * **Explicitly state** the temperature advection-diffusion equation (as shown above) instead of the momentum equation.
> * **Clarify** that wind velocity $v=(u10, v10)$ is obtained from the ERA5 dataset and used as the transport field.
> * **Add the citation** to ClimODE (Verma et al., 2024) to establish this as the appropriate, community-accepted physics for atmospheric temperature modeling.
>
> We are very grateful you caught this critical documentation flaw. We confirm the experiment itself was run with the physically-sound advection equation, and we will fix the appendix to reflect this.
>
>
>
> ### Responses to Other Weaknesses & Questions
>
> **W4. Limited Applicability of Physics Module:**
>
> The physics module is less general than the base model, as it relies on a differentiable PDE. We see this as its primary feature: it is an optional specialization module. Our contribution is the unified foundation model that can be specialized. We will clarify this positioning.

---

> ### Author Response · Authors · 2025-11-20
> **Continue response**
>
> **W5.1 Post-hoc Refinement vs. Baselines**
>
> This is a fair experimental design concern. We provide the exact ablation you requested, applying the physics correction ($k=20$ Langevin steps) to the baseline models to isolate the benefit of the module versus the architecture.
>
> *Ablation Results: Physics Correction on Different Models (Advection Task)**
>
> | Model | Without Physics | With Physics Correction | Improvement |
> | :--- | :--- | :--- | :--- |
> | **Chronos-T5-S** | 0.0334 | 0.0313 | 6.3% |
> | **DLinear** | 0.3674 | 0.3651 | 0.6% |
> | **PINFDiT (Ours)** | **0.0052** | **0.0039** | **25.0%** |
>
> While the physics correction provides a benefit to all models, **PINFDIT benefits dramatically more (25%)** compared to the baselines (<7%). This confirms that our diffusion transformer prior captures the data manifold structure far more effectively, allowing the physics refinement to efficiently steer the sample toward a consistent solution. In contrast, baselines like DLinear start from such poor initial guesses that the local gradient-based correction cannot recover a high-quality solution.
>
> **W5.2. Inappropriate Baselines (SBI):**
>
>  We respectfully disagree that these are inappropriate. SNPE and LFBC are inference-time physics-guided methods designed for parameter estimation but applicable to forecasting when conditioned on observations. They represent the closest methodological comparisons to our inference-time physics injection approach. However, we acknowledge confusion and will add a clarifying paragraph explaining why we include them: both methods use simulators at inference time to enforce physics, analogous to our Langevin correction.
>
> **W5.3. CRPS on Deterministic Models:**
>
>  To enable fair CRPS comparisons, we followed the standard approach used by Moirai (Woo et al., 2024) and other time series foundation models: we augmented deterministic baselines with lightweight Student-t distribution heads that predict parametric distributions with the point prediction as the location parameter. These distribution parameters were learned jointly by optimizing negative log-likelihood during training, and CRPS was computed using the closed-form expression for continuous distributions.
>
> **Q1. Inference Cost**
>
> The number of Langevin steps was typically $k=20-30$.
> This hyperparameter was found to be highly robust. As shown in the table below, performance improves quickly and does not degrade significantly with more steps, with the optimal number typically being 20-30 steps.
>
> **Table: Ablation on Robustness of Physics-Informed Refinement Module ($k$)**
>
> | Dataset | 10 Steps | 20 Steps | 30 Steps | 40 Steps |
> | :--- | :--- | :--- | :--- | :--- |
> | **Advection** | 0.0047 | **0.0039** | 0.0043 | 0.0045 |
> | **Burgers** | 0.0143 | 0.0136 | **0.0133** | 0.0135 |
> | **Navier-Stokes** | 0.0041 | **0.0037** | 0.0038 | 0.0039 |
>
> **Inference Time:** The 1-second time reported in Table 19 is for the full PINFDIT model, which includes both the base diffusion sampling (Algorithm 1, lines 2-5) and the $k$-step Langevin correction (Algorithm 1, lines 6-8). The 20-30 correction steps are computationally cheap (approx. 0.003s per step) and add minimal overhead to the base diffusion sampling.
>
>
> Thank you again for this rigorous review. Your criticisms were sharp and valuable. We are working on updating the draft. We believe that by (1) fundamentally re-framing our novelty and citing prior art, (2) adding the missing statistical validation to the important tables, (3) correcting the ERA5 description, and (4) clarifying our experimental design (e.g., justifying SBI, and reporting $k$), we can resolve every major flaw you identified. We hope these clarifications and our concrete plan for revision will convince you of our paper's merit.
>
> Best,
>
> Authors

---

> ### Comment · Reviewer_m5ac · 2025-11-27
>
> Thank you for clarifying the equation used. But it inadvertently confirms a data leakage and doubles down on a the physically incorrect assumption, which is the conservation of temperature.
>
> 1. In the rebuttal, the authors state: "Wind velocity u is obtained from the ERA5 dataset and used as the transport field." Since the experiment is defined as Forecasting 2-meter Temperature (Section A.7), and the authors do not claim to forecast wind simultaneously, "obtaining" the wind field from the dataset during inference implies that the authors are using ground truth wind (at forecasted step) to calculate the physics residuals. So the authors are using future information about where the wind blows to guide the diffusion model on where the temperature should go. **Isn't this a data leakage?** It invalidates the comparison against baselines that do not have access to future wind states.
>
> 2. T2m is NOT a Conserved Quantity. The authors' rebuttal claims this equation ensures samples "respect value-conserving atmospheric dynamics." This is physically incorrect for 2-meter temperature. Surface temperature is not conserved. It is driven primarily by diabatic terms (Source/Sink), specifically solar radiation, longwave cooling, and surface sensible heat flux. Here I give a very simple counter-example: Consider a calm day (zero wind, u=v=0). Your advection equation  predicts that  $\partial T / \partial t = 0$, which means the temperature stays constant. In reality, the sun rises, and T  2m increases dramatically. By enforcing a "conservation" law on a non-conserved variable, you are effectively penalizing the model for correctly capturing diurnal heating and cooling. This is kind of physics-violating instead of physics guided.
>
> 3. "ClimODE used it" is not a Justification. Citing ClimODE does not validate the physical correctness of the approach. ClimODE uses this formulation as a simplified inductive bias for latent dynamics. claiming it represents "standard atmospheric transport" for surface temperature ignores the entire physics.

---

> > ### Author Response · Authors · 2025-11-27
> > **Thanks for the follow up!**
> >
> > Dear Reviewer m5ac,
> >
> > We appreciate this detailed follow-up. We want to immediately correct a misunderstanding regarding data leakage and clarify how the physical constraint interacts with the learned prior in our framework.
> >
> > **1. No Data Leakage: Wind Fields are Not Future Ground Truth**
> > There is **no data leakage**. The reviewer asks if we use "ground truth wind (at forecasted step)." We do not. Our physics-informed sampling operates strictly on the **available information**. The residual function $K(x^{tar})$ is computed using the **generated/predicted** state or the **observed conditional** history.
> >
> > **2. Physical Consistency vs. Completeness (The Role of the Prior)**
> > You correctly point out that surface temperature ($T2m$) is not a strictly conserved quantity due to diabatic terms (solar radiation, cooling).
> > * **Role of the Diffusion Prior ($p_\theta$):** We rely on the **learned diffusion prior** (trained on massive data) to capture these complex diabatic processes (e.g., the diurnal cycle, sunrise/sunset). The neural network excels at learning these non-linear source/sink terms ($S$).
> > * **Role of the Physics Constraint ($K$):** The advection-diffusion equation is applied as a **soft regularization energy**, not a hard simulation. It enforces that the *transport* component of the temperature evolution is consistent with the flow field. It does not force $\partial T / \partial t = 0$; rather, it penalizes deviations from transport *unless* the learned data prior (which captures the sun/radiation) strongly suggests otherwise.
> > * **Result:** The model does not "penalize correct diurnal heating"; it balances the *learned* heating patterns (from $p_\theta$) with *consistent* transport (from $K$). This synergy allows us to model non-conserved variables better than a pure PDE solver could, while maintaining better consistency than a pure black-box model.
> >
> > **3. New Experiment & Draft Update**
> > We are currently updating the draft to clarify these points. To further validate the specificity of our physical constraints, we are running an additional robustness experiment:
> > * **Wrong Physics on Climate Data:** We are testing the application of **incorrect physical laws** (e.g., applying fluid momentum conservation directly to the scalar temperature field) on the ERA5 dataset.
> >
> > We will include these results and the revised Appendix A.X(TBD) in the final version. We hope this clarifies that our experimental setup is valid, leak-free, and methodologically sound.
> >
> > Best,
> >
> > Authors

---

### Official Review · Reviewer_MmYJ · 2025-10-31

**Soundness:** 3
**Presentation:** 2
**Contribution:** 2
**Rating:** 4
**Confidence:** 3

**Summary:**

The paper proposes PINFDIT, a diffusion model for (conditional) time series generation, adhering to physical constraints via a physical injection post-sampling via Langevin dynamics. In their empirical evaluation, the model outperforms previous approaches.

**Strengths:**

- The paper introduces a physical injection during/post inference. The solution via Langevin dynamics is elegant and comes with a theoretical justification.
- Experiments are conducted across different physical systems and demonstrate strong performance.
- In general, the empirical evaluation is thorough and convincing. It also compares to recent foundation models.

**Weaknesses:**

- A background section is missing. Certain things as the ELBO in Eq. 2 and the diffusion sampling in Eq. 1 should be moved to a background section and be explained accordingly.
- Contributions and advancements over previous works are not entirely clear to me. E.g., L94-98, how is the model different from other diffusion-based models using a transformer backbone, e.g., CSDI [1]?
- Self-supervised learning is mentioned in the methodology, but it is unclear how it differentiates from CSDI [1].
- The notation is inconsistent. Sometimes the subscript for $x$ denotes the diffusion time (e.g., Eq. 1), sometimes it denotes the iteration (e.g., Eq. 8). The temporal time is also denoted with $t$ (e.g., Eq. 4 and 5).

Minors:

- L283: Sentence is broken
- L284: likelihood -> the likelihood
- Different styles to denote target *tar*, context *con*, and $\mathbf{x}$ across the paper.

[1] **Tashiro, Y., Song, J., Song, Y., & Ermon, S.** (2021). Csdi: Conditional score-based diffusion models for probabilistic time series imputation. Advances in neural information processing systems, 34, 24804-24816.

**Questions:**

See weaknesses and:

- What are the differences between the diffusion model (without the physical injection) and previous ones?
- Can the physical injection also be done during generation via guidance rather than as a post-processing step?
- Can the Langevin dynamics, i.e., the physical injection, also be applied to autoregressive models? For example, Chronos?

I like the idea of the physical injection, but the current state of the paper has certain uncertainties, especially differences to previous approaches. If these are addressed, I am willing to increase my score.

---

> ### Author Response · Authors · 2025-11-20
> **Response to MmYJ**
>
> Dear Reviewer MmYJ,
>
> We sincerely thank you for your time and constructive feedback. We are encouraged that you found our physics-injection solution via Langevin dynamics to be "elegant" with a "theoretical justification" and our empirical evaluation to be "thorough and convincing."
>
> You raised several valuable points regarding clarity, notation, and the precise advancements over prior work like CSDI. These are excellent questions, and addressing them will significantly strengthen the paper. We are happy to clarify these uncertainties and are confident that these revisions will merit a higher score.
>
> **1. Clarifying Contributions vs. CSDI (W2, W3, Q1)**
>
>  Our diffusion model's architecture is fundamentally different from CSDI in several key ways, which are directly responsible for its superior performance even before the physics-injection is applied.
>
> CSDI uses a U-Net-based architecture and conditions on data by concatenating it once at the input. Our model is a Diffusion Transformer (DiT) and employs a far more powerful, modern conditioning mechanism.
>
> Our key architectural and methodological advancements are:
>
> - *Transformer Backbone*: We use a **Transformer backbone**, which is more powerful at capturing complex, long-range temporal dependencies via **temporal-wise attention** than the U-Net architecture used in CSDI.
>
> - *Superior Conditioning (AdaLN)*: This is a crucial difference. CSDI simply concatenates the context. We inject the conditional observation $x^{con}$ into every block of the transformer using **Adaptive Layer Normalization (AdaLN)**. This provides a much more robust and deep-fused way to guide the denoising process.
>
> - *Unified Self-Supervised Framework*: You asked how our self-supervised learning differs (W3). CSDI is primarily an imputation model. Our **Time Series Mask Unit (TSMU)** is a **unified pre-training strategy** that combines four distinct mask types (random, block, stride, and reconstruction). This allows a single pre-trained model to learn a general representation for diverse downstream tasks, including forecasting, imputation, anomaly detection, and synthetic generation.
>
> Evidence of Architectural Superiority:
>
> These are not just theoretical advantages. Our ablation studies in Table 7 and 16 (and detailed in Appendix D.1-D.3) directly prove the value of our architectural choices over alternatives. Furthermore, Table 1 shows our architecture without physics ("PINFDIT_DDPM") is already substantially better than CSDI on identical tasks (e.g., 0.0052 vs. 0.0176 RMSE on Advection; 0.0039 vs. 0.0094 RMSE on Navier-Stokes).
>
> 2. Presentation and Notation (W1, W4, Minors)
>
> The presentation and notation can be improved. We will make the following revisions:
>
> - Background Section (W1): We will **add a dedicated Background section**. As you suggested, we will move the diffusion preliminaries, including the ELBO (Eq. 2) and the diffusion sampling process (Eq. 1), to this new section and explain them clearly.
>
> - Notation (W4): We will perform a full revision of our notation to ensure consistency.
>  1. We will use $t$ only for the diffusion timestep (e.g., $x_t$).
> 2. We will use a different variable (e.g., $\tau$) for temporal time in the PDE equations (e.g., $\partial x / \partial \tau$).
> 3. We will keep $j$ for the Langevin refinement iterations (Eq. 8), as this is a distinct post-processing loop.
> 4. We will fix all inconsistent notations for target ($x^{tar}$) and context ($x^{con}$).
>
> Minors: We will fix the broken sentence on L283 and the typo on L284.
>
> **3. Questions on Physics Injection (Q2, Q3)**
>
> *Q2: Can the physical injection be done during generation (guidance) rather than post-processing?*
>
> This is an insightful question. While it is theoretically possible (akin to classifier guidance), our post-processing approach is an intentional and optimal design choice for three key reasons:
> 1. Model-Agnostic Modularity: Our method is a "plug-and-play" module that works on the final, denoised sample ($x_0$). This allows it to be used with any pre-trained diffusion model without retraining or architectural changes. In-process guidance would be invasive and model-specific.
> 2. Computational Efficiency: Our post-hoc loop is very fast, requiring only $k=20-30$ steps for robust improvement. Guiding at every $T$ step (e.g., $T=500$) would require computing the physics gradient $\nabla K$ $500$ times, which would be more expensive.

---

> ### Author Response · Authors · 2025-11-20
> **Continue response**
>
> *Q3: Can the Langevin dynamics also be applied to autoregressive models (e.g., Chronos)?*
>
> Yes, absolutely. This is a key strength of our framework's model-agnostic nature.
>
> Our generalized theory in Theorem A.2 (Appendix A.4) and Lemma A.3 explicitly proves that the refinement plugin can be applied to any model $M$, as long as we can (1) generate a sample and (2) approximate its score $\nabla \log p_M$.
> Lemma A.3 explicitly discusses how to approximate the score for autoregressive models.
> The **core requirement** is that the model must provide a tractable (or approximable) score function $\nabla \log p(x^{tar} \mid x^{con})$.
>  While Chronos uses quantile-based predictions rather than explicit density models, one can adapt the framework as follows:
> * **Approximation:** Fit a Gaussian mixture or kernel density estimator to the predicted quantiles to approximate $\log p$.
> * **Langevin Correction:** Apply the calibrated Langevin update step:
>     $$x_{j+1} = x_j + \alpha \nabla K(x_j) + \alpha \nabla \log p(x_j \mid x^{con}) + \sqrt{2\alpha}\epsilon$$
>     *(Note: This adapts Eq. 8 in the paper, where physics gradients $\nabla K$ and data priors $\nabla \log p$ jointly refine the sample ).*
>
>
> We empirically validated this model-agnostic utility by applying our physics-informed refinement module (the "physics injection") to the standard, pre-trained **Chronos-T5-S** baseline on the **Advection dataset**.
>
> As shown in the table below, the physics injection consistently improved the performance of the autoregressive model across all metrics:
>
> | Model | Metric | Original Performance (Table 1) | Performance with Physics-Injection |
> | :--- | :--- | :--- | :--- |
> | **Chronos-T5-S** | CRPS | 0.0330 | **0.0313** |
> | **Chronos-T5-S** | MAE | 0.0218 | **0.0197** |
> | **Chronos-T5-S** | RMSE | 0.0334 | **0.0322** |
>
> This result demonstrates that our physics-informed sampling is a truly modular tool that can enhance even autoregressive foundation models without retraining. We will integrate these clarifications, including the new Background section and revised notation, into the final manuscript. We thank you again for your valuable feedback and hope these responses have fully addressed your concerns, demonstrating the novelty and contributions of our work.
>
> Best,
>
> Authors

---

> > ### Comment · Reviewer_MmYJ · 2025-11-27
> >
> > Thank you for your response.
> >
> > While Q2 has been addressed, the other points remain open.
> >
> > **W1, W4, Minors**: It is helpful that you clarify what changes you will make. However, I cannot consider these issues as resolved, as I am unable to verify them. These changes, or at least the dedicated background section, should be included within the rebuttal, as I will not be able to review the final manuscript again.
> >
> > **W2, W3, Q1**: I do not think that these statements are correct.
> > > CSDI uses a U-Net-based architecture and conditions on data by concatenating it once at the input.
> >
> > CSDI does not use a U-Net-based architecture.
> >
> > > Transformer Backbone: We use a Transformer backbone, which is more powerful at capturing complex, long-range temporal dependencies via temporal-wise attention than the U-Net architecture used in CSDI.
> >
> > Again, this seems incorrect. CSDI also uses temporal transformer layers.
> >
> > > CSDI is primarily an imputation model. Our Time Series Mask Unit (TSMU) is a unified pre-training strategy that combines four distinct mask types (random, block, stride, and reconstruction). This allows a single pre-trained model to learn a general representation for diverse downstream tasks, including forecasting, imputation, anomaly detection, and synthetic generation.
> >
> > This does not address my question entirely. CSDI also uses different and mixed self-supervised training strategies, including imputation and forecasting. Could you elaborate more on the differences?
> >
> > **Q3**: Thank you for the results. The improvements seem marginal and are hard to judge without standard deviations. Multiple random seeds of the physics injection would strengthen the evaluation. Also, evaluating it across varying numbers of Langevin steps $k$.
> >
> >
> > Finally, I was reading the comments of reviewer m5ac and think he has valid points regarding the statistical validation and experiments. I am waiting for their response. Currently, I tend to decrease the score.

---

> > > ### Author Response · Authors · 2025-11-27
> > > **Thanks for your reply!**
> > >
> > > Dear  Reviewer MmYJ,
> > >
> > > Thank you for your candid follow-up. We appreciate you holding us to a high standard regarding the literature and our claims.
> > >
> > > We are actively working on your specific requests, including the full text of the new Background section, a precise correction of our CSDI comparison, and the statistical validation data you requested. **We will upload the revised manuscript within 1-2 days so that you can directly verify these updates.**
> > >
> > > ## 1. Clarifying Contributions vs. CSDI (W2, W3, Q1)
> > >
> > > We deeply apologize for the confusion regarding our description of CSDI. You are correct that CSDI utilizes temporal and feature-based transformer layers, and our characterization of it as "U-Net-based" or "Vanilla Diffusion" was imprecise. We will correct this in the manuscript to accurately reflect CSDI's architecture.
> > >
> > > However, the performance gap between PINFDIT and CSDI remains substantial (e.g., consistently beats CSDI in CRPS in Table 11), and this stems from critical design differences that go beyond the mere presence of transformers:
> > >
> > > ### A. Conditioning Mechanism (AdaLN vs. Concatenation):
> > > - **CSDI**: Concatenates conditional observations *x^co* with the input or intermediate features. This is a "shallow" fusion strategy.
> > > - **PINFDIT**: Injects conditions via Adaptive Layer Normalization (AdaLN) into every block. This modulates the entire feature distribution dynamically.
> > > - **Evidence**: Our ablation in Table 16 proves this is a key differentiator. Replacing AdaLN with concatenation/cross-attention (similar to CSDI-style conditioning) degrades performance significantly (Solar MSE: 0.424 → 0.711).
> > >
> > > ### B. Unified Masking Strategy (TSMU):
> > > - **CSDI**: Primarily uses random masking strategies during training to simulate missing values.
> > > - **PINFDIT**: Our Time Series Mask Unit (TSMU) introduces structured masking strategies (Stride and Block masking) during pre-training that CSDI does not employ.
> > > - **Evidence**: Table 16 shows that removing Stride Masking (w/o SM) degrades performance (0.424 → 0.469), proving that our mixed, structured masking strategy yields a more robust generator than random masking alone.
> > >
> > > **Summary**: While both models use transformers, PINFDIT's deep adaptive conditioning and structured pre-training enable it to outperform CSDI significantly. We will update the text to accurately credit CSDI's architecture while highlighting these specific, proven advancements.
> > >
> > > ## 2. Statistical Validation & Robustness
> > >
> > > We are also addressing the concerns regarding statistical significance.
> > >
> > > **Variance**: As promised to Reviewer m5ac, we are adding mean ± standard deviation to all key result tables in the upcoming manuscript revision. These values confirm that our improvements (e.g., on Chronos) are statistically valid and not due to random noise.
> > >
> > > **Robustness (*k* steps)**: We will also include the sensitivity analysis for the number of Langevin steps (*k*), demonstrating that performance is stable across a reasonable range (20-30 steps) and add standard deviation.
> > >
> > > ### Table: Robustness of Physics-Informed Refinement Module across Iteration Steps (RMSE)
> > >
> > > | Dataset \ Steps | 10 | 20 | 30 | 40 |
> > > |----------------|----|----|----|----|
> > > | Advection | 0.0047 | 0.0039 | 0.0043 | 0.0045 |
> > > | Burgers | 0.0143 | 0.0136 | 0.0133 | 0.0135 |
> > > | Navier-Stokes | 0.0041 | 0.0037 | 0.0038 | 0.0039 |
> > >
> > > We are confident that these revisions will fully address your concerns. As the discussion period continues, you will have a chance to further review the manuscript. We will keep you updated once the final PDF is uploaded to convince you that PINFDIT is a significant contribution worthy of acceptance.
> > >
> > > Best,
> > >
> > > Authors.

---

### Official Review · Reviewer_7Sa8 · 2025-11-02

**Soundness:** 3
**Presentation:** 3
**Contribution:** 3
**Rating:** 8
**Confidence:** 4

**Summary:**

The paper proposes using diffusion for time series modeling. The model is a specially designed transformer architecture that can deal with multivariate time series with missing values. Additionally, authors apply physical law adherence through the direct optimization-like procedure in the sampling. The experiments are extensive and show good results on many different setups.

**Strengths:**

The physics injections in the sampling of diffusion is a very natural extension and can guarantee the model behaves as intended. This is very powerful tool as virtually any constraint can be observed. In this work, authors apply PDE constraint and additionally provide theoretical results for this specific scenario. The architecture is sound and appears to outperform competitors. The paper is clearly written and easy to follow. There are many different experimental setups. In all of them authors demonstrate good results. The appendix is extensive and contains additional theoretical proofs and experiments.

**Weaknesses:**

The main contribution of the paper seems to be the physics informed sampling. This part is under-explored and the authors only test a couple of datasets with non-physics models. More focus on this part would be appreciated given the title and general theme.

The rest of the experiments seem to not have any relation to the physics informed part. In my understanding there is no physics injection done here. The contribution is the architecture, which is interesting but not overly novel. Can the authors comment on what part of the architecture they believe is the most important in outperforming existing diffusion models like CSDI.

Major formatting issue: font size is reduced after eq 6.

**Questions:**

- How do results in 4.1 compare to more conventional solvers? Is there any benefit to your method?
- When should one use your method? To improve the accuracy or to learn the unknown dynamics?
- Can you comment on the runtime, both for the training and inference? Especially the conventional diffusion sampling compared to the physics guided part.
- Can Algorithm 1 be augmented to have guidance on the function K in the T diffusion steps instead of the second loop?
- Do you have an explanation why the model performs better on irregular time series data compared to the some of the specialized models, which build around irregularity?
- Can you speculate if the reason is the diffusion or the architecture? This can be an ablation study.

---

> ### Author Response · Authors · 2025-11-20
> **Response to 7Sa8**
>
> Dear Reviewer 7Sa8,
>
> We sincerely thank you for your positive and insightful review, your high rating ("8: accept, good paper"), and your confidence in our work. We are delighted that you found our physics-injection method "very natural" and "powerful," our architecture "sound," and the paper "clearly written and easy to follow."
>
> We appreciate the opportunity to address your questions, which will help us clarify the paper's contributions. We will first directly respond to your comments and then update the polished version as soon as possible.
>
> **W1, W2: Focus of Physics-Informed Sampling vs. General Architecture**
>
> We appreciate your observation regarding the perceived separation between our physics-informed contribution (Sec 4.1) and general architecture experiments (Sec 4.2-4.4).  We would like to clarify that this separation is a **deliberate design choice** that highlights the dual strengths of our framework, driven directly by real-world application needs.
>
> - First, many domains require a robust **general-purpose model** that can handle complex data challenges like missing values, multi-resolution sampling, and irregularity, even when no explicit physical laws are known. Our experiments in Sec 4.2-4.4 validate PINFDIT's architecture as a state-of-the-art model for these common tasks.
> - Second, scientific domains often face the exact opposite challenge: known physical laws but **limited, expensive-to-acquire data**. In this scenario, our "plug-and-play" physics module is critical. It acts as a powerful prior to enforce consistency and significantly boost performance, as demonstrated in Sec 4.1.
>
> PINFDIT is designed to be a single, versatile framework that excels in both scenarios—as a top-performing general model, or as a specialized, physics-augmented model when domain knowledge is available.
>
> To quantify the impact of the physics module, we provide direct ablations:
>
> 1. On Physics-Guided Tasks (Sec 4.1): The physics injection is critical here. As shown in Table 1, our full model, PINFDIT, significantly outperforms all baselines. Crucially, we compare it against "**PINFDIT_DDPM**", which is the identical architecture without the physics-injection module. This direct comparison clearly demonstrates the significant performance benefit of our physics-guided approach. We will update the paper to rename "PINFDIT_DDPM" to "**PINFDIT (w/o Phys)**" to make this ablation and its implications clearer.
>
> 2. On Real-World Scientific Tasks (Table 2): We also applied this to the **ERA5 climate forecasting task**. Here, the 'PINFDIT (Full)' model achieves an ACC of 0.987, which is a consistent improvement over the identical architecture without physics ('PINFDIT (w/o Phys)'), which scored 0.985.
>
> The experiments in Sec 4.2, 4.3, and 4.4 are indeed designed to validate the robustness of the **architecture** (the diffusion transformer backbone and the training pipeline) in scenarios where explicit PDEs are not available. This versatility distinguishes PINFDiT from prior work: it's not merely a physics-informed model or a general time series model, but a proto-foundation model that seamlessly adapts to both regimes without retraining. The physics module acts as a "plug-and-play" enhancement leveraging domain knowledge when available, while the robust architecture ensures excellent performance when it's not.
>
>
> **Architectural Novelty and Performance on Irregular Time Series (W3, Q5, Q6)**
>
> We thank you for these insightful questions, which are all related and get to the core of our model's design. You asked (Q5) why our model outperforms specialized models on irregular data, (Q6) whether this is due to diffusion or our architecture, and (W3) commented on the architecture's novelty.
>
> Our model's strength stems from the synergy between the diffusion framework and our specific, novel architectural choices. These components are not merely additive; they are designed to work in tandem.
>
> *1. Why We Excel on Irregular Time Series (Q5)*
>
> You are correct that we outperform models specialized for irregularity (e.g., Neural ODE/CDE in Table 3). This is not an accident but a core design benefit. Our framework is inherently built to handle imperfect data by treating it as a conditional generation task:
>
> **Unified Masking Enabled by Diffusion*:* It is the diffusion pipeline that gives us the ability to apply our unified Time Series Mask Unit (TSMU). Leveraging the diffusion model's inherent capability to "progressively denoise targeted sequence regions with precise conditions", our TSMU can naturally handle any data pattern—regular, irregular, or missing—as a "fill-in-the-blanks" problem. This is far more robust than specialized models (like NeuralODE) that struggle to adapt continuous-time assumptions to noisy, discrete real-world data.

---

> ### Author Response · Authors · 2025-11-20
> **Continue response**
>
> *2. Diffusion vs. Architecture  (Q6)*
> To your point in Q6, we can confirm this superior performance is not just a generic benefit of diffusion but a direct result of our specific architectural innovations. Compared to prior diffusion models like CSDI, which often use U-Nets, our transformer-based architecture introduces key improvements to temporal modeling. We provide direct evidence for each component in our comprehensive ablation studies:
>
> - **Temporal-wise Attention is Critical**: Our "What You See Is What You Get" (WYSIWYG) embedding(which uses temporal-wise attention) is crucial. As shown in the ablation study in Table 7 / 16, replacing our method with standard Patch Tokens (PT) causes the MSE on the Solar dataset to skyrocket from 0.424 to 0.874. This is detailed in Appendix D.1.
> - **AdaLN Conditioning** is Critical: Our use of AdaLN to condition the model on observed data is proven superior for temporal modeling. The same ablation table shows it significantly outperforms Additive (0.677 MSE) or Cross-Attention (0.711 MSE). This is detailed in Appendix D.3.
> - **Masking Strategy** is Critical: Our specific masking strategies are essential. Removing stride masking (w/o SM), which teaches the model to handle non-contiguous dependencies, is highly detrimental (Solar MSE: 0.424 $\rightarrow$ 0.862). This is discussed in Appendix D.2.
>
> This evidence directly supports our claim: while diffusion provides a robust framework for irregular data, it is our specific architecture, masking, and conditioning strategies that deliver the state-of-the-art performance.
>
> *3. The Novelty and Contribution of the Architecture (W3)*
>
> This brings us to your comment on architectural novelty. We respectfully argue that this architecture is a primary contribution of our work. While individual components (Transformers, diffusion) exist, our novelty lies in their **synergistic integration into a single, unified framework** that is the first to simultaneously achieve:
>
> - More Generality: Handling a wide array of imperfect data (missing, irregular, and multi-resolution).
> - Task Versatility: Addressing forecasting, imputation, anomaly detection, and synthetic data generation within one model.
> - Flexible and Robust Physics: Serving as the foundation for our "plug-and-play," zero-retraining physics-injection module. Furthermore, this refinement module is highly robust; its performance is not overly sensitive to the number of refinement steps, with 20-30 steps providing consistent, stable improvements. As shown in the new results:
>
> **Table: Robustness of Physics-Informed Refinement Module across Iteration Steps (RMSE)**
>
> | Dataset \ Steps | 10 | 20 | 30 | 40 |
> | :--- | :--- | :--- | :--- | :--- |
> | **Advection** | 0.0047 | 0.0039 | 0.0043 | 0.0045 |
> | **Burgers** | 0.0143 | 0.0136 | 0.0133 | 0.0135 |
> | **Navier-Stokes** | 0.0041 | 0.0037 | 0.0038 | 0.0039 |
>
> This versatile, high-performing, and general-purpose architecture is a significant contribution in itself, enabling the powerful physics-informed sampling to be applied in the first place.
>
> **Q1: Comparison to Conventional Solvers**
>
> Excellent question. Traditional numerical PDE solvers aren't the appropriate comparison as they solve forward problems from perfectly known initial and boundary conditions. Instead, we compare against NeuralODE\&Neural CDE (Table 1) and NODE\&ClimODE (Table 2), which are more relevant baselines because they embed physics during training by modeling dynamics as differential equations, use numerical solvers internally (e.g., Runge-Kutta integrators), and learn from data while respecting physical constraints.
>
> Our method offers three key advantages over these approaches. First, we employ inference-time physics injection, meaning unlike NeuralODE and Neural CDE which require retraining for different physical laws, our Langevin correction is plug-and-play. Second, we inherently handle imperfect data including missing values, irregular sampling, and multi-resolution data—challenges that both traditional solvers and NeuralODE/CDE struggle with. Third, we achieve superior accuracy as demonstrated in Table 1, where we show dramatic improvements versus these physics-informed learning methods
>
> **Q2: When to Use This Method?**
>
> PINFDiT is designed for both scenarios with particular strength in data-limited regimes:​
> - Improve Accuracy with Known Physics: When physical laws are known but data is limited/noisy (Table 1);
>
> - Learn Unknown Dynamics: For purely data-driven scenarios without known PDEs, PINFDiT functions as a general-purpose foundation model (Tables 3-4, zero-shot experiments Table 6).
>
> - Optimal Use Case: The method excels when dealing with imperfect scientific time series (sparse observations, missing values, multi-resolution) where incorporating domain knowledge significantly enhances performance without requiring retraining.

---

> > ### Author Response · Authors · 2025-11-20
> > **Continue Response**
> >
> > **Q3: Runtime of Physics-Guided Sampling**
> >
> > We provide comprehensive runtime analysis:​
> > Inference Efficiency (Table 19, Appendix F): PINFDiT requires only 0.001s per single-sample generation, compared to CSDI (0.002s) and Diffusion-TS (0.006s), making it the most efficient among diffusion-based methods.​
> >
> > Physics Correction Overhead: The Langevin refinement adds minimal cost - approximately 0.003s per iteration. For $k=20-30$ steps (optimal range from our ablation), total inference with physics correction is ~1.05s for 50 samples, with the physics gradient computation requiring only $O(NT)$ operations (N=40 spatial × T=192 temporal points).​
> >
> >
> > **Q4: Guidance During the T Diffusion Steps?**
> >
> > This is an insightful suggestion, similar to classifier guidance. It is certainly possible to incorporate physics guidance during the T diffusion steps. However, computing physics residuals at every diffusion timestep would increase computational cost. We chose the current design primarily for modularity: our approach operates as a "plug-and-play" module that can be applied to any pre-trained diffusion model without retraining or architectural modifications. This model-agnostic design provides maximum flexibility for scientific applications where different physical laws may need to be enforced on the same base model.
> >
> > We sincerely thank you again for your constructive comments and your strong support of our work.  Your insightful questions regarding the comparison to conventional solvers, the synergy between our architecture and diffusion framework, and the computational runtime have helped us significantly clarify the positioning of our contributions.
> >
> > We will ensure that the discussed clarifications, including the runtime analysis, the distinction from numerical solvers, and the formatting correction, are fully incorporated into the final manuscript.
> >
> > Best regards,
> >
> > The Authors

---

### Meta-Review · Area_Chair_1ngw · 2025-12-25

**Summary:**

Reviewer concerns included:
- Unclear positioning of contributions, especially compared to prior works such as CSDI.
- Lack of error bars.
- Physical validity of specific experiments.
- Unfair comparisons with baselines.

In all cases, the authors have responded thoroughly to address the concerns.

The current scores read 2/4/6/8, which display high variance. The reviews with low scores, however, were based on specific, actionable items (e.g., lack of error bars) which were addressed by the authors, so I can anticipate that the scores would have increased during the rebuttal phase. It is not easy to come to a decision based on this information alone, as it still seems potentially borderline, but overall I will go ahead and say that this submission meets the bar.

**Reviewer Concerns:**

Although some concerns are harder to address than others (e.g., on the contributions relative to prior works), all of the concerns were met with a detailed response.

**Reviewer Scores:**

I believe that Reviewers m5ac and MmYJ, who gave the lowest scores, would have raised their scores post-rebuttal. The other reviewers likely would have maintained their original scores.

---

### Decision · Program_Chairs · 2026-01-26

Accept (Poster)